# Isotonic and minimally invasive optical clearing media for live cell imaging ex vivo and in vivo

Shigenori Inagaki [1,2] ✉, Nao Nakagawa-Tamagawa [3], Nathan Zechen Huynh [4], Yuki Kambe [4], Rei Yagasaki [5], Satoshi Manita [6], Satoshi Fujimoto [1], Takahiro Noda[1], Misato Mori[5], Aki Teranishi [5], Hikari Takeshima[1], Koki Ishikawa[1], Yuki Naitou[7], Tatsushi Yokoyama [8,9], Masayuki Sakamoto [8,10], Katsuhiko Hayashi [7,11,12], Kazuo Kitamura [6], Yoshiaki Tagawa[3], Satoru Okuda [5,13], Tatsuo K. Sato[2,4] & Takeshi Imai [1,13] ✉

Tissue clearing has been widely used for fluorescence imaging of fixed tissues, but its application to live tissues has been limited by toxicity. Here we develop minimally invasive optical clearing media for fluorescence imaging of live mammalian tissues. Light scattering is minimized by adding spherical polymers with low osmolarity to the extracellular medium. A clearing medium containing bovine serum albumin (SeeDB-Live) is compatible with live cells, enabling structural and functional imaging of live tissues, such as spheroids, organoids, acute brain slices and the mouse brains in vivo. SeeDB-Live minimally affects neuronal electrophysiological properties and sensory responses in vivo, and facilitates fluorescence imaging of deep cortical layers in live animals without detectable toxicity to neurons or behavior. We further demonstrate its utility to epifluorescence voltage imaging in acute brain slices and in vivo preparations. Thus, SeeDB-Live expands both the depth and modality range of fluorescence imaging in live mammalian tissues.

Live biological tissues are dynamic by nature. Thanks to various chemical and genetically encoded fluorescent biosensors, we can image and measure dynamic biological phenomena within the live tissues and organs using fluorescence microscopy. However, the imaging depth is often limited by tissue opacity. It has been a long-standing challenge to make live and healthy biological tissues transparent to facilitate live imaging.

The opacity of the biological tissues is largely due to the inhomogeneity of refractive index within the samples. Two-photon microscopy uses a near-infrared excitation laser instead of visible light to reduce light scattering; however, the imaging depth is limited to a few hundred microns in mammalian tissues in vivo[1]. Adaptive optics uses a deformable mirror or spatial light modulator to correct aberrations caused by macroscopic refractive index distortions, but it is not effective for highly scattering samples[2]. For fixed tissues, optical clearing is a powerful approach: light refraction and scattering are minimized by removing high-index components (for example, lipids) and/or by immersing the sample in high-index solutions with refractive indices of 1.43–1.55 (refs. 3–13). However, most of the clearing agents developed for fixed tissues are toxic to live cells. Less toxic chemicals (for example, glycerol, dimethyl sulfoxide and sugars) have been tested for highly fibrous extracellular structures, such as skin and skull in vivo[14–19]; however, these chemicals interfere with cellular functions. A recent study claimed to have achieved optical clearing of the skin in live animals using a strongly absorbing dye, such as tartrazine[18]. However, the osmolality of the dye solutions used in the study was several-fold higher

than the physiological osmolality condition, precluding its application to live imaging of normal physiological functions. Therefore, live mammalian cells and tissues have not yet been rendered transparent while maintaining intact cellular functions.

Some chemicals have been proposed to be compatible with live cell imaging. Iodinated contrast agents were attractive candidates because of their low osmolarity. One of them, iodixanol, improves the transparency of bacteria and some multicellular organisms[20,21]. However, its toxicity to mammalian cells has not been fully evaluated. Another study attempted to improve transparency of the mouse brain by adding glycerol to drinking water[22]. However, it is unclear whether the marginal change in transparency was due to an increase in the refractive index in the brain, as glycerol should be easily metabolized once absorbed in the gut.

Here we developed SeeDB-Live, a tissue-clearing medium for live mammalian cells and tissues. SeeDB-Live contains bovine serum albumin (BSA), which has exceptionally low osmolarity when dissolved in water and is minimally invasive to live cells. SeeDB-Live improved the imaging depth of spheroids, organoids, acute brain slices and the mouse brain in vivo.

## Results

### Strategies for minimally invasive optical clearing of live mammalian cells

Light scattering in tissues is caused by refractive index mismatch between the light scatterer and the medium. Previously, simple immersion-based clearing agents (refractive index, 1.46–1.52) have been developed (for example, fructose, iohexol and tartrazine)[8,9,18]; however, osmolarity of these clearing agents is extremely high. To make live tissues transparent under isotonic conditions, we would have to use either (i) membrane-permeable or (ii) membrane-impermeable low-osmolarity (that is, high molecular weight) chemicals to reduce the refractive index mismatch (Fig. 1a). For (i) membrane-permeable chemicals, we do not need to change the concentration of the saline; however, when (ii) membrane-impermeable chemicals are added to the medium, we would need to subtract the concentration of the saline to keep the medium isotonic. We have listed membrane-permeable and membrane-impermeable high-molecular-weight chemicals as candidates. Candidate chemicals

also need to be highly soluble in water. These chemicals demonstrated a concentration-dependent increase in refractive index when dissolved in water (Extended Data Fig. 1a).

Next, we sought to determine the optimal refractive index for clearing live mammalian cells. For this purpose, we prepared a suspension of live or paraformaldehyde (PFA)-fixed and membrane-permeabilized HeLa cells ($4 \times 10^6$ cells per ml). The refractive indices of media used to clear fixed tissues are typically 1.43–1.55 (ref. 10). We tested a membrane-permeable chemical, glycerol, up to a refractive index of 1.43 (~66% wt/vol); however, it was not effective for live mammalian cells (Fig. 1b). We also tested a membrane-impermeable chemical, iodixanol; to keep the osmolarity of the buffer isotonic, we mixed isotonic iodixanol solution (60% wt/vol) and phosphate buffered saline (PBS) to prepare isotonic solutions with different refractive indices. PFA-fixed and membrane-permeabilized HeLa cells were most transparent at a refractive index of ~1.42. Paradoxically, however, we found that the live HeLa cells become most transparent at an extracellular refractive index of ~1.37, much lower than the optimal index for fixed cells (Fig. 1b). Moreover, the optimal range of the refractive index for live cells was relatively narrow; the transparency of live cells became lower at higher refractive indices (>1.38).

We next examined whether intracellular functions remain intact in the presence of candidate chemicals. Using the GCaMP6f calcium indicator, we evaluated the calcium responses of HEK293T cells to 50 μM ATP solution under various clearing media at a refractive index of 1.365 (Fig. 1c and Extended Data Fig. 1b,c). Calcium responses were completely abolished in the presence of membrane-permeable chemicals, glycerol (23% wt/vol), dimethyl sulfoxide (DMSO) and propylene glycol, while a lower concentration of glycerol (5%) showed weak responses. These results indicate that membrane-permeable clearing agents impair cellular functions at a refractive index of 1.365. Among the membrane-impermeable, high-molecular-weight chemicals, straight polymers abolished calcium responses (for example, polyethylene glycol and polyvinyl pyrrolidone). In contrast, intact calcium responses were observed for iodinated contrast agents (for example, iodixanol) and spherical polymers (for example, Ficoll70). These results indicate that some of the membrane-impermeable, high-molecular-weight chemicals could be useful for index matching of the extracellular medium without compromising cellular functions.

**Fig. 1 | Screening for nontoxic optical clearing agents for imaging live mammalian cells and tissues. a**, Strategies for optical clearing of live cells. **b**, Transmittance (at 600 nm) of HeLa cell suspension ($4 \times 10^6$ cells per ml) in isotonic saline solution with glycerol or iodixanol at different refractive indices. Fixed cells were treated with PFA and saponin. **c**, Calcium imaging of GCaMP6f-expressing HEK293T cells stimulated with 50 μM ATP. The refractive index of the medium was adjusted to 1.365 (except for 5% glycerol). Osmolarity was not adjusted to isotonicity. Data are the median ± interquartile range (IQR). ***$P < 0.001$; **$P < 0.01$; NS, not significant ($P \geq 0.05$; two-sided Dunnett's multiple-comparison test). **d**, The osmolality of candidate chemicals in aqueous solution (refractive index 1.365, in double-distilled water (ddH$_2$O; $n = 3$ each). Sucrose was used as a control. Spherical polymers refer to polymers with highly branched and/or higher-order structure. BSA#1 and BSA#2 represent two examples of different BSA products. The osmolality of low-salt BSA (2) was 2.7 mOsm kg$^{-1}$, consistent with its molar concentration (2.3 mM). **e**, The optimal refractive index of the extracellular medium was determined in PBS adjusted at different osmolalities. Transmittance of live HeLa cell suspensions ($4 \times 10^6$ cells per ml) was measured. Left: optimal refractive index (1.369) of iodixanol-containing PBS. Right: optimal refractive index (1.363–1.369) of BSA-containing PBS ($n = 3$ each). **f**, Phase contrast images of live HeLa cells in normal and BSA-containing medium (refractive index, 1.363). **g**,**h**, Growth of HeLa/Fucci2 cells. Cell numbers were measured by fluorescence imaging of cell nuclei ($n = 5$ wells). **g**, Proliferation curve of HeLa/Fucci2 cells in iodixanol, Ficoll70 and BSA#1-containing medium (refractive index, 1.363; 320 mOsm kg$^{-1}$). *$P < 0.05$; NS ($P \geq 0.05$; two-sided Dunnett's multiple-comparison test). $P$ values are <0.001 unless otherwise mentioned. **h**, Growth ratio in refractive index-optimized

(refractive index, 1.363; 320 mOsm kg$^{-1}$) medium compared to the control medium. *$P < 0.05$; NS ($P \geq 0.05$; two-sided Dunnett's multiple-comparison test). $P$ values are <0.001 unless otherwise mentioned. **i–k**, HeLa/Fucci2 cell spheroids cleared with SeeDB-Live. **i**, Phase contrast images of HeLa/Fucci2 cell spheroids under normal (left) and SeeDB-Live culture medium (refractive index 1.366, 320 mOsm kg$^{-1}$; right). **j**, Growth curve of HeLa/Fucci2 cell spheroids with and without treatment with SeeDB-Live for 4 h per day. NS ($P \geq 0.05$; two-sided Wilcoxon rank-sum test combined with Holm–Bonferroni correction). **k**, Three-dimensional (3D) confocal images of a HeLa/Fucci2 cell spheroid. **l**,**m**, Intestinal organoids in Matrigel treated with SeeDB-Live (refractive index, 1.363) for 4 h per day. **l**, Phase contrast image. **m**, Growth of the intestinal organoids. The sizes of the organoids (areas in the phase contrast images) were determined using Cellpose. NS ($P \geq 0.05$; two-sided Wilcoxon rank-sum test combined with Holm–Bonferroni correction). **n**, 3D confocal images of GCaMP6s-expressing EECs in intestinal organoids from ePet-Cre; Ai162 mice before and after SeeDB-Live treatment. **o**,**p**, Calcium imaging of cortical organoids (confocal). Basal fluorescence (temporal median; left) and Δ$F/F_0$ images (right) of a cortical organoid labeled with a calcium indicator, Calblyte-650AM (**o**). Spontaneous calcium transients of neurons (**p**). **q**, Principles of optical clearing of live cells with SeeDB-Live. Maximum transparency was achieved by matching the refractive index of the extracellular medium to that of the cytosol (1.363–1.366). Data with error bars represent the mean ± s.d. Images are representatives of ≥3 trials. See Supplementary Table 4 for detailed statistical data. MW, molecular weight; PG, propylene glycol; PEG, polyethylene glycol; HBCD, hyperbranched cyclic dextrin; PVP, polyvinyl pyrrolidone.

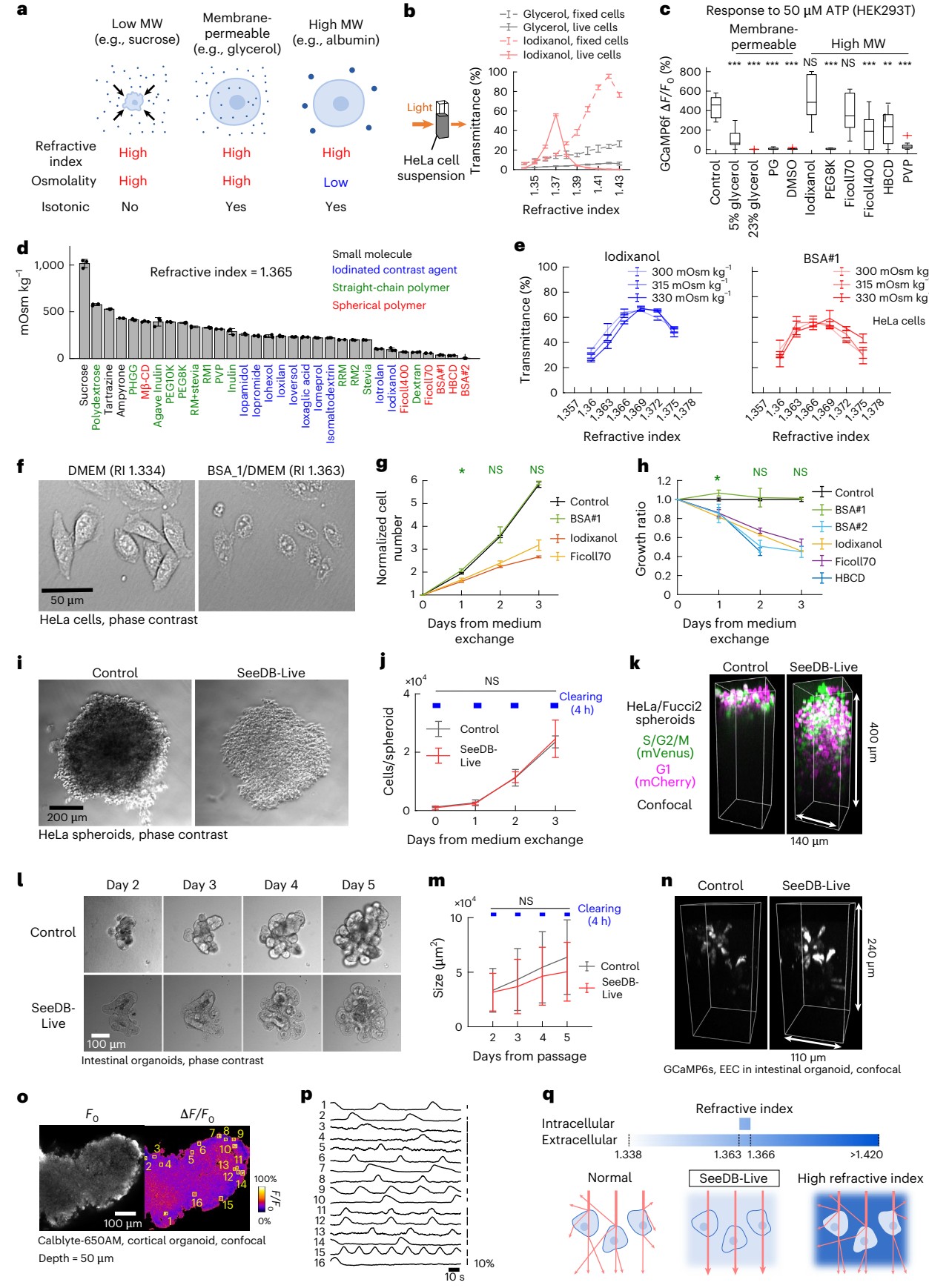

## Screening and optimization of low-osmolarity and nontoxic clearing agents

Membrane-impermeable chemicals will not directly interfere with intracellular functions but may increase osmolarity. To keep the clearing medium isotonic, we would have to reduce the concentration of the saline; however, the extracellular ionic conditions would affect membrane properties. Therefore, the ideal chemical should have a low osmolarity when dissolved in water to achieve the optimal refractive index. The increase in osmolarity can be minimized if we use high-molecular-weight chemicals (>1 kDa); however, extremely large particles (>10-nm scale) will cause Rayleigh scattering.

We measured the osmolality of the candidate media prepared at a refractive index of 1.365. Straight-chain polymers had prohibitively high osmolalities, much higher than the theoretical values based on molar concentrations[23]; the higher osmolality may explain why straight-chain polymers showed cellular toxicity (Fig. 1c and Extended Data Fig. 1c). In contrast, we found that spherical polymers (polymers with highly branched and/or higher-order structures) have much lower osmolalities (Fig. 1d). Among them, low-salt BSA (BSA#2) demonstrated exceptionally low osmolality of only 2.7 mOsm kg$^{-1}$, consistent with its molar concentration (2.3 mM; Fig. 1d and Extended Data Fig. 1d). Slightly higher osmolarity for another BSA product (BSA#1) was due to residual salts in the product (Extended Data Fig. 1e), suggesting that BSA itself has very low osmolality.

The osmolarity of the human serum is typically 280–290 mOsm l$^{-1}$. The osmolarity of the saline buffers and culture media for mammalian cells is 230–340 mOsm l$^{-1}$, but is typically 300–330 mOsm l$^{-1}$. For both iodixanol (control) and BSA, we further refined the optimal refractive index for this range. We prepared PBS at 300 mOsm kg$^{-1}$, 315 mOsm kg$^{-1}$ and 330 mOsm kg$^{-1}$ with refractive indices of 1.360–1.375 using iodixanol or BSA. The highest transparency was found at refractive indices of 1.363–1.369 when prepared at 300–330 mOsm kg$^{-1}$ (Fig. 1e and Extended Data Fig. 1f). The best refractive index was higher at higher extracellular osmolarity, likely because the cytosol is more condensed. When live HeLa cells were incubated with BSA-containing medium (refractive index, 1.363), the plasma membrane was almost invisible under the phase contrast microscopy (Fig. 1f).

Using index-optimized isotonic culture medium, we evaluated long-term toxicity using HeLa cells. Cell growth was monitored for up to 3 days. Cell growth was comparable to the control DMEM for one of the BSA products (BSA#1; Fig. 1g,h and Extended Data Fig. 1g,h,m,n). However, cell growth was lower for iodixanol, highly branched cyclic dextrin, Ficoll70 and some of the BSA products (Fig. 1g,h and Extended Data Fig. 1g,i).

BSA is also preferable in terms of lower viscosity (Extended Data Fig. 1j,k) and specific gravity (Extended Data Fig. 1l). Culture with iodixanol is difficult because the specific gravity of the cell is lower than that of the iodixanol solution, and cells easily detach and float in the medium[20]. Other proteins may be similarly useful; however, BSA has exceptional water solubility and is one of the most affordable proteins available. In addition, albumin is the most abundant protein in the serum (4–5% wt/vol) and has been widely used for mammalian cell culture, suggesting that BSA is minimally adverse to the mammalian cells.

Albumin buffers divalent cations (for example, Ca$^{2+}$ and Mg$^{2+}$)[24,25]. We, therefore, optimized the total concentration of Ca$^{2+}$ and Mg$^{2+}$ in the media to keep the concentrations of free Ca$^{2+}$ and Mg$^{2+}$ physiological; the optimal total concentration was 1.5–3-fold higher than in the conventional artificial cerebrospinal fluid (ACSF) based on the evaluation in neurons (Extended Data Fig. 2a,b). Earlier biochemical studies indicated that a half of Ca$^{2+}$ and Mg$^{2+}$ binds to BSA in this condition, consistent with our results[24,26]. Because BSA substantially contributes to the negative charge of the buffer, we also evaluated the acceptable range of Na$^+$ and Cl$^-$ concentrations (Extended Data Fig. 2c–e). Primary cultures of mouse cardiomyocytes (Extended Data Fig. 3a–c) and hippocampal neurons (Extended Data Fig. 3d–g) were maintained in the

BSA-containing culture medium for at least 3 days without any obvious signs of toxicity.

In this way, we established BSA-containing clearing media (15–17% wt/vol) for live mammalian cells, named SeeDB-Live, with optimal refractive index (1.363–1.366), osmolality (230–340 mOsm kg$^{-1}$) and total Ca$^{2+}$ and Mg$^{2+}$ concentrations (4–6 mM and 1.5–2.5 mM, respectively) with saline or culture medium (Supplementary Tables 1 and 2). The concentration of BSA in SeeDB-Live is only twice as high as the total protein concentration in the serum (typically 6–8% wt/vol).

Recently, strongly absorbing dyes (for example, tartrazine and ampyrone) have been shown to clear the mouse skin[18,27], but they work only under prohibitively high osmolality conditions. Under physiological osmolality conditions (~300 mOsm kg$^{-1}$), they have lower refractive indices and cannot effectively clear live cells (Fig. 1d and Extended Data Fig. 3h–k). Another recent study increased refractive index of the extracellular media by only 0.01 using polymer solutions (6% polyethylene glycol (PEG) and 4% dextran)[28]; however, the refractive index of 1.34–1.35 was far below the optimal range for live mammalian cells (Fig. 1b and Extended Data Fig. 3k). Thus, SeeDB-Live is currently the only method that achieves optical transparency of live, healthy mammalian cells.

## Fluorescence imaging of live spheroids and organoids with SeeDB-Live

We examined whether SeeDB-Live is useful for fluorescence imaging of multicellular structures. We cleared cultured HeLa/Fucci2 cell spheroids[29] with SeeDB-Live; the spheroids became quickly transparent without apparent shrinkage or expansion under SeeDB-Live (Fig. 1i and Supplementary Video 1). Growth of HeLa/Fucci2 spheroids was slightly slower when continuously cultured in SeeDB-Live, possibly due to lower circulation of oxygen (Extended Data Fig. 4a)[30]. However, daily clearing with SeeDB-Live for 4 h per day did not affect the growth of the spheroid culture (Fig. 1j). In the confocal microscopy, mVenus and mCherry signals were visible up to ~100 μm in depth in the control medium; in contrast, signals were visible up to ~250 μm in the SeeDB-Live medium (Fig. 1k, Extended Data Fig. 4b–d and Supplementary Video 2). The brightness of the signals was improved particularly in the deeper area of the spheroids (Extended Data Fig. 4d).

Next, we tested SeeDB-Live for imaging intestinal organoids cultured in Matrigel. Intestinal organoids were developed from ePet-Cre; Ai162 mice, in which enteroendocrine cells (EECs) express a calcium indicator, GCaMP6s. The intestinal organoids became transparent after the incubation in SeeDB-Live (Fig. 1l), and the organoid growth was not affected by daily 4-h clearing with SeeDB-Live (Fig. 1m). The luminal cavity of the organoid was less transparent, suggesting that BSA does not efficiently penetrate the tight junctions formed by the epithelial tissues. Nonetheless, the GCaMP6s-positive EECs were visible in deeper areas under SeeDB-Live using confocal microscopy (Fig. 1n and Extended Data Fig. 4e). Calcium imaging demonstrated robust responses to high potassium stimulation, indicating that their physiological functions are maintained (Extended Data Fig. 4f). We also tested SeeDB-Live for confocal calcium imaging of the neuroepithelial and cortical organoids induced from mouse embryonic stem cells (Fig. 1o,p and Extended Data Fig. 4g–j)[31]. Thus, SeeDB-Live will be useful for functional assays of organoids.

Together, our results indicate that the light scattering in live cells can be greatly reduced by index matching between the cytosol (1.363–1.366) and the extracellular medium. Index matching of the extracellular medium with isotonic medium with BSA (SeeDB-Live) is minimally invasive and powerful for optical clearing of live mammalian tissue (Fig. 1q).

## Clearing acute brain slices with SeeDB-Live

Volume imaging is in high demand for neuroscience applications. Perfusion with SeeDB-Live/ACSF cleared acute brain slices within 30 min

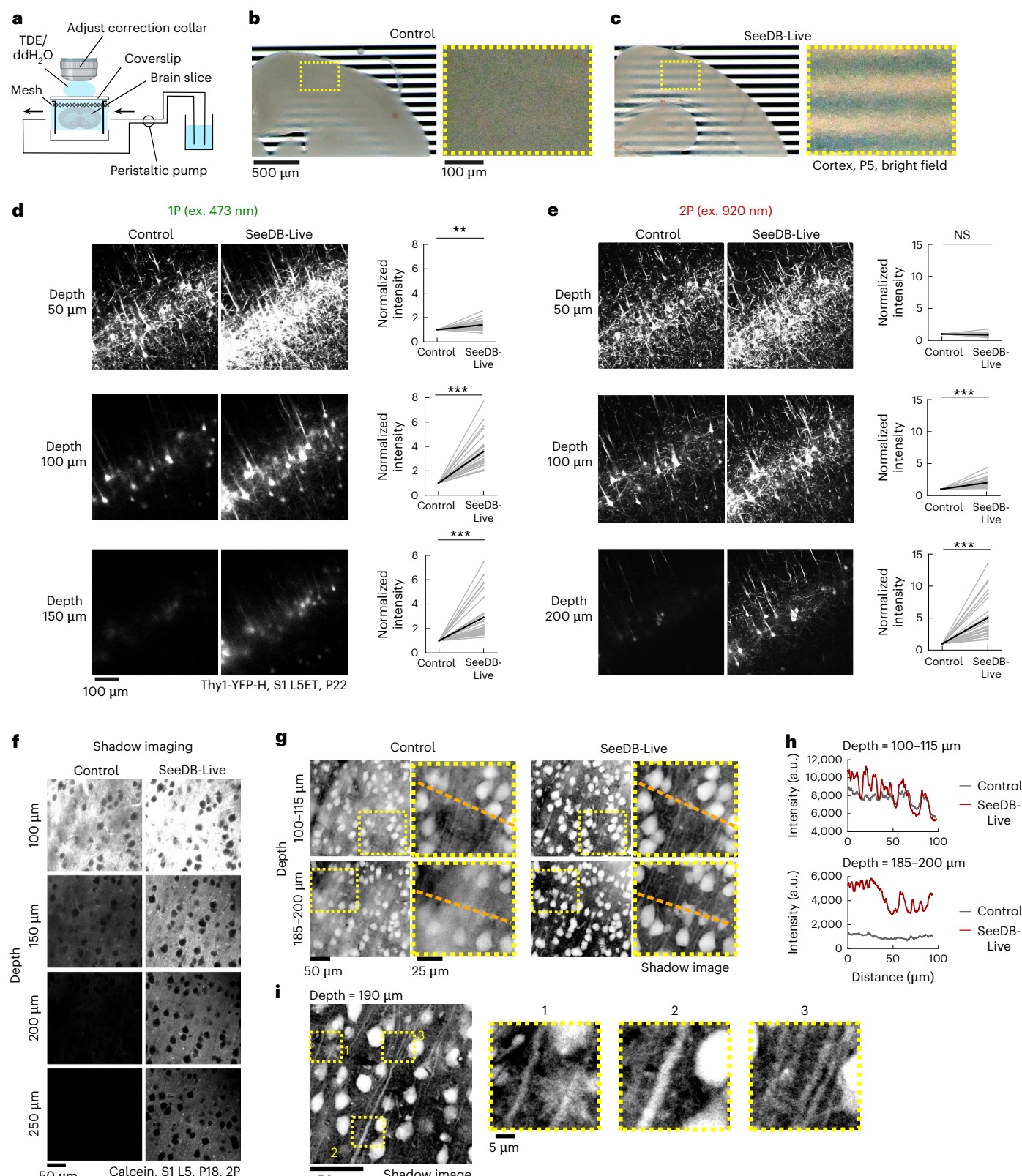

f Shadow imaging

Calcein, S1 L5, P18, 2P

h Depth = 100–115 μm

Depth = 185–200 μm

i Depth = 190 μm

Shadow image

(Fig. 2a–c and Supplementary Video 3). We evaluated the performance of SeeDB-Live using acute brain slices from Thy1-YFP-H mice. After the recovery of acute brain slices in oxygenated ACSF, confocal and two-photon images were acquired. The brain slices containing the cerebral cortex were then perfused with SeeDB-Live/ACSF for 1 h and imaged again under the same conditions. The imaging depth was increased ~2-fold for both confocal and two-photon microscopy under SeeDB-Live (Fig. 2d,e and Supplementary Videos 4 and 5). Similar results were obtained for the hippocampus (Extended Data Fig. 5a–c and Supplementary Videos 6 and 7). The optimal refractive index of SeeDB-Live was ~1.363 in acute brain slices (Extended Data Fig. 5d), consistent with our results for cultured cell data (Fig. 1h). We did not

**Fig. 2 | Improved morphological imaging of acute brain slices with SeeDB-Live.** **a**, Preparation of acute brain slices and clearing with SeeDB-Live. Acute brain slices were perfused with SeeDB-Live/ACSF (refractive index, 1.363; 320 mOsm kg$^{-1}$ in ACSF) at a flow rate of 1.5 ml min$^{-1}$ in a chamber. 15.6% (vol/vol) 2,2′-thiodiethanol (TDE) in ddH$_2$O (refractive index, 1.363) was used for immersion to minimize spherical aberration. The correction collar of the objective lens was turned to the appropriate position. **b,c**, An acute brain slice (300-µm thick; age, postnatal day 5 (P5); bright-field images) before (b) and after (c) clearing with SeeDB-Live/ACSF. Right: magnified images. **d,e**, Left: $x$–$y$ fluorescence images of an acute brain slice (300-µm thick) of L5ET neurons in S1. Confocal one-photon (1P) and two-photon (2P) images are shown. Thy1-YFP-H mice (age, P22) were used. Right: normalized fluorescence intensity from cell bodies in $x$–$y$ fluorescence images of S1 L5ET neurons are shown on the right for each depth. The same sets of neurons were compared before and after clearing. \*\*\*$P < 0.0001$, \*\*$P < 0.001$, NS ($P \geq 0.05$; Wilcoxon signed-rank test). **f**–**i**, Two-photon (2P) shadow imaging using SeeDB-Live. **f**, Two-photon shadow images in S1 L5 region of acute brain slices (age P18, 300-µm thick). Imaging was performed before and after clearing with SeeDB-Live/ACSF. Both ACSF (used for recovery and imaging) and SeeDB-Live/ACSF contain 40 µM calcein. **g**, Shadow images (left) and their magnified views (right) in the intermediate (100–115-µm stack) and deeper (185–200-µm stack) regions (inverse look-up table images). Note that the brightness and contrast were adjusted for each image because fluorescence intensities differed across the conditions. **h**, Line plots showing the raw fluorescence intensity along the orange dashed line in **g** under control and SeeDB-Live. **i**, Magnified views of the shadow images showing dense nerve fibers. Data from representative samples of ≥3 trials are shown. See Supplementary Table 4 for detailed statistical data. ex., excitation; a.u., arbitrary units. Panel **a** created in BioRender. Imai, T. (2026) https://BioRender.com/gyynf4j.

observe improved transparency with 5% glycerol ex vivo, contrary to a previous report in vivo (Extended Data Fig. 5f, g)[22].

Previously, shadow imaging of organotypic brain slice cultures with super-resolution, confocal and two-photon microscopy have been proposed for comprehensive structural imaging of the brain, including dense connectomics applications[32–34]; in these techniques, the extracellular space of the brain slices is labeled with a dye solution (for example, calcein). The shadow images visualize the structure of all components in the tissue, allowing for comprehensive structural profiling. Previously, these techniques can only access the surface of the brain slices due to the light scattering. However, the surface of the brain slices (~50 µm) is often mechanically damaged during slice preparation (Extended Data Fig. 5e), making it difficult to image 'acute' brain slices that represent native in vivo structure. Using SeeDB-Live, the imaging depth possible for the shadow imaging was much improved with two-photon microscopy (Fig. 2f and Supplementary Video 8). Inverse look-up table images demonstrated morphology of all the cells at higher signal-to-noise ratio under SeeDB-Live (Fig. 2g–i). Thus, SeeDB-Live facilitates comprehensive structural imaging of acute brain slices.

## Electrophysiological properties of neurons cleared with SeeDB-Live

For comprehensive recording of neuronal activity with SeeDB-Live, it is important to ensure that neuronal functions remain intact. Using acute mouse brain slices (age, P15–18), we examined membrane properties of layer 5 extratelencephalic-projecting (L5ET) neurons using patch-clamp recording (Fig. 3a and Extended Data Fig. 6a–e). We found that liquid junction potentials are different between ACSF (13.04 mV ± 0.20 mV) and SeeDB-Live/ACSF (9.19 mV ± 0.09 mV; Extended Data Fig. 6a). After calibration for the liquid junction potentials, there was no significant difference in the resting membrane potential of L5ET neurons (−76.3 mV ± 2.9 mV for the ACSF and −75.2 mV ± 3.1 mV for SeeDB-Live/ACSF; mean ± s.d.; $P = 0.33$, Wilcoxon rank-sum test; Fig. 3b). Some of the electrophysiological parameters were slightly affected (Fig. 3b and Extended Data Fig. 6e). However, the firing properties in the frequency–current curve were not affected, possibly because differences in some factors counteracted each other (Fig. 3c). We obtained consistent results in older animals (Extended Data Fig. 6f,g) and for fast-spiking interneurons (Extended Data Fig. 6h–l).

Patch-clamp recording under SeeDB-Live was extremely difficult because brain slices were almost transparent. We cannot exclude the possibility that unintentional sampling bias has contributed to the difference. It should also be noted that the ionic composition of SeeDB-Live is not identical to that of the control ACSF. The total amount of Ca$^{2+}$ and Mg$^{2+}$ is adjusted higher (Extended Data Fig. 2a,b). Cl$^{-}$ concentration is slightly lower because BSA substantially contributed to the net negative charge (Extended Data Fig. 2c,d). BSA may also show the Donnan effect. These factors potentially affect the electrophysiological properties, and further optimization might be needed for more specific experiments.

## Calcium imaging of brain slices with SeeDB-Live

We next evaluated population-level properties of neurons using slice calcium imaging. We used olfactory bulb slices (P11–15), in which mitral/tufted cells show spontaneous activity[35]. We used Thy1-GCaMP6f mice, in which mitral/tufted cells express GCaMP6f. We imaged mitral cells with two-photon microscopy at a depth of ~100 µm. The frequencies

**Fig. 3 | Electrophysiological recording and calcium imaging of neurons in acute brain slices. a**–**c**, Electrophysiology in acute brain slices. L5ET neurons in S1 were analyzed at P15–18. Samples were analyzed at the same time point after preparation. **a**, Changes in membrane potentials in response to square current pulses of −300 pA (blue), +100 pA (green) and +300 pA (red) in L5ET neurons. Representative neurons are shown. Data were calibrated for liquid junction potentials in control ACSF (13.04 mV) and SeeDB-Live/ACSF (9.19 mV). **b**, Resting membrane potential, action potential (AP) threshold, AP amplitude and input resistance are shown. \*\*$P < 0.01$; NS ($P \geq 0.05$; Wilcoxon rank-sum test). $n = 17$ neurons from four mice and 14 neurons from three mice for control and SeeDB-Live, respectively. **c**, AP frequency was plotted against injected current amplitude. NS (Wilcoxon rank-sum test). **d,e**, Spontaneous currents at the holding potential of −60 mV. Representative traces (**d**), amplitude and frequency of spontaneous excitatory postsynaptic currents (sEPSCs; **e**) are shown. Data are the median ± IQR. **f**–**n**, Spontaneous and evoked responses of mitral cells in the olfactory bulb (OB) cleared with SeeDB-Live and imaged with two-photon microscopy. **f**, GCaMP6f fluorescence images (temporal median) of mitral cells in acute OB slices (age, P11) imaged with two-photon microscopy. Traces for representative neurons (arrows) are shown. Spontaneous activity was imaged before (left), during (middle) and after (right) clearing with SeeDB-Live/ACSF at a depth of 100 µm from the surface of the slice. **g,h**, Amplitude and frequency of spontaneous activity in the same set of mitral cells in ACSF and SeeDB-Live/ACSF. $n = 18$ cells from three mice (age, P11–14). NS (multiple comparisons with Bonferroni correction). **i**–**k**, 100 µM NMDA and 40 µM glycine (Gly) were applied to the OB slices (Thy1-GCaMP6f, age P15) for 1.5 min. $n = 70$ and 47 cells from three mice each for control and SeeDB-Live, respectively. $\Delta F/F_0$ images (**i**), time traces (mean ± s.d.) (**j**) and response amplitudes (**k**) are shown. NS (two-tailed Welch's $t$-test). The slower decay of the response may be due to slower washout of NMDA/glycine under SeeDB-Live. **l**–**n**, Acute OB slices (age, P9–11) were imaged with two-photon microscopy at a depth of 150 µm. **l**, Basal fluorescence (temporal median) of GCaMP6f and $\Delta F/F_0$ images are shown for different time points. Basal fluorescence intensity (**m**) and $\Delta F/F_0$ (**n**) of mitral cell somata during clearing with SeeDB-Live/ACSF are shown. $n = 38$ cells from three mice. \*\*\*$P < 0.001$; NS ($P \geq 0.05$; two-sided Tukey–Kramer multiple-comparison test). **o**, Confocal images of acute OB slices (Thy1-GCaMP6f mouse, age P13) under control and SeeDB-Live conditions. The images of temporal median are shown. Representative data are from ≥3 trials. Box plots indicate the median ± IQR. Whiskers indicate 1.5 times the IQR. See Supplementary Table 4 for detailed statistical data.

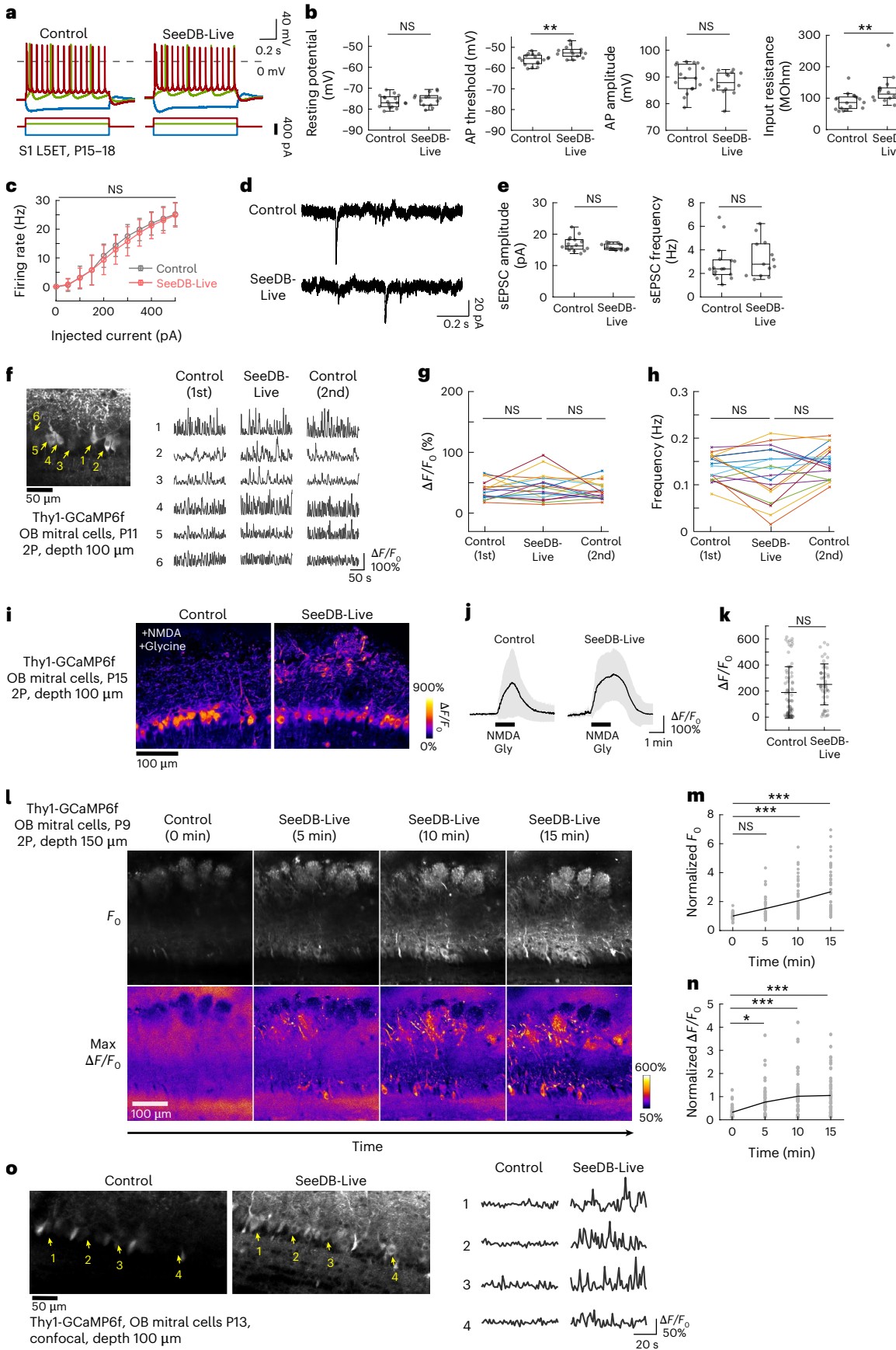

of spontaneous activity were not significantly different between control and SeeDB-Live when $Ca^{2+}/Mg^{2+}$ concentrations were optimized (Fig. 3f–h, Supplementary Tables 3 and 4 and Supplementary Video 9; but see also Extended Data Fig. 2a,b for non-optimized conditions). In contrast, spontaneous activity was no longer maintained in the glycerol or iodixanol-containing ACSF (Extended Data Fig. 6m–q). Evoked responses (to 100 μM $N$-methyl-D-aspartate (NMDA) and 40 μM glycine) of mitral cells were also comparable between control and SeeDB-Live (Fig. 3i–k). Time-lapse imaging of a deeper area (150-μm depth) demonstrated a significant improvement in brightness and $\Delta F/F_0$ by SeeDB-Live/ACSF treatment (Fig. 3l,m, Supplementary Tables 3 and 4 and Supplementary Video 10).

Notably, spontaneous activity of mitral cells was clearly visible with one-photon confocal microscopy at a depth of 100 μm under SeeDB-Live, but not with control ACSF (Fig. 3o and Supplementary Videos 11 and 12). It should be noted that the superficial ~50 μm of acute brain slices is typically damaged during sample preparation, and intact neuronal activity is only visible in deeper areas, where only two-photon microscopy can access under the normal ACSF. Thus, SeeDB-Live enables calcium imaging of healthy neuronal activity in acute brain slices using conventional confocal microscopy, without using two-photon microscopy systems.

## Optical clearing of cerebral cortex in vivo

SeeDB-Live is based on index matching with a membrane-impermeable molecule, BSA. Therefore, the performance of clearing is limited by the accessibility of BSA to the tissues. To clear the mouse brain in vivo in live animals, we performed a craniotomy at the primary somatosensory cortex (S1) and removed the dura mater (durotomy) under anesthesia. We then exposed the brain surface to SeeDB-Live/ACSF with gentle perfusion, allowing BSA to permeate into the CSF of the brain (Fig. 4a). We confirmed that fluorescently tagged BSA was infused to a depth of ~500 μm from the surface of the cerebral cortex (Fig. 4b,c). We used a transgenic line, Thy-YFP-H, in which L5ET neurons are labeled with EYFP. Under two-photon microscopy, overall brightness of EYFP signals was increased in the deeper area after incubation with SeeDB-Live for 1 h (Fig. 4d–f and Supplementary Video 13). Their basal dendrites, including dendritic spines, were better visualized with SeeDB-Live (Fig. 4g,h). The brightness of L5ET somata, located at a depth of 600–800 μm,

was increased ~3-fold by SeeDB-Live treatment (Fig. 4i,j and Supplementary Video 14). The cleared part returned opaque once SeeDB-Live is diluted by the CSF circulation and/or by active washout with ACSF (Fig. 4k). Thus, SeeDB-Live is a powerful tool for in vivo imaging of live neurons in the brain.

We examined possible toxicity of SeeDB-Live in vivo using a large cranial window on the right cortical surface (Fig. 4l). Acute SeeDB-Live treatment of the right cortex, including motor cortices, in awake animals did not affect locomotor activity on a treadmill (Fig. 4m). Moreover, SeeDB-Live treatment did not affect locomotor activity, motor function (wire hanging test) and food intake on consecutive days (Fig. 4n–p). We observed no obvious sign of the inflammatory responses in the brain (for example, the number and morphology of neurons and microglia) after clearing with SeeDB-Live (Extended Data Fig. 7). Thus, SeeDB-Live treatment does not induce acute or chronic toxicity in animals.

## In vivo functional imaging of neurons with SeeDB-Live and two-photon microscopy

Next, we investigated whether sensory responses are preserved after SeeDB-Live treatment in vivo. We performed a durotomy in the primary visual cortex (V1) and compared the visual responses of layer 4 neurons before and after SeeDB-Live treatment (Fig. 5a), assuming that layer 4 is adequately infused with SeeDB-Live (Fig. 4b,c). Using calcium indicators jGCaMP8m and Cal-520, we recorded the calcium responses of the same sets of neurons to visual grating stimuli of different orientations under anesthesia (Fig. 5a–e and Extended Data Fig. 8a–e). We found that the preferred orientation, response amplitude ($\Delta F/F_0$), orientation selective index (OSI) and tuning width of layer 4 neurons were largely preserved (Fig. 5c–e). Thus, SeeDB-Live preserves physiological sensory responses.

In the olfactory bulb, we were able to better visualize odor responses in mitral cell somata located at a depth of ~400 μm using GCaMP6f (Fig. 5f). Due to the improved brightness, inhibitory responses, represented by a reduction in basal GCaMP6f fluorescence, were better detected using SeeDB-Live (Fig. 5g,h and Supplementary Video 15).

Voltage imaging is more challenging than calcium imaging in vivo due to the lower signal-to-noise ratio of the signals. However, using SeeDB-Live, we were able to reliably detect action potentials from the

---

**Fig. 4 | Optical clearing of the mouse brain in vivo with minimal toxicity.**
**a**–**k**, Optical clearing and fluorescence imaging of the cortex in live mice under anesthesia. **a**, Schematic diagram of surgery and clearing of the mouse cortex with SeeDB-Live/ACSF-HEPES (refractive index, 1.363; 300 mOsm kg⁻¹). Craniotomy and durotomy were made on the right hemisphere. The brain surface was perfused with SeeDB-Live/ACSF-HEPES and perfused for 1 h under anesthesia. The objective lens was directly immersed in SeeDB-Live/ACSF-HEPES. The correction collar of the objective lens was turned to the best position. **b,c**, The diffusion of fluorescently labeled BSA (1% BSA-CF597 dissolved in SeeDB-Live) into the cortex in anesthetized mice (age, 2–4 months). The mice were euthanized either immediately (0 h) or 24 h after treatment. Frozen sections of non-perfused and unfixed brains were analyzed (**b**). The relative fluorescence intensity across cortical depth is shown (**c**). $n = 3$ mice for each time point (0 h and 24 h after treatment). **d**–**k**, S1 of a Thy1-EYFP-H mouse was imaged before and after clearing with SeeDB-Live/ACSF-HEPES (1 h after clearing) with two-photon microscopy. L5ET neurons are labeled. **d**, 3D-rendered images of L5ET neurons (Thy1-YFP-H; age, 6 months). Laser power and photomultiplier tube gain were kept constant across the depths. Depths were 0–700 μm. **e**, $x$–$y$ images at different depths. **f**, Fluorescence intensity at different depths. $n = 3$ mice. **g,h**, Somata and basal dendrites of L5ET neurons (age, 4 months). Basal dendrites and their dendritic spines could only be clearly visualized after clearing with SeeDB-Live/ACSF-HEPES. Depth was 495 μm. **i,j**, Time-lapse images of L5ET neurons in S1 during in vivo clearing with SeeDB-Live/ACSF-HEPES (**i**). **j**, Quantification of fluorescence for the same sets of neurons. Depth was 590 μm. **k**, S1 L5ET neurons of a 4-month-old Thy1-EYFP-H mouse were imaged using two-photon microscopy before, during and after 1 h of clearing with

SeeDB-Live/ACSF-HEPES. **l**–**p**, Toxicity assay using animal behavior. **l**, A large cranial window encompassing motor and somatosensory areas was made for the right hemisphere. After craniotomy and durotomy, an optical window was made using a PVDC wrapping film, silicone sealant and a coverslip (center; day −7). SeeDB-Live treatment was performed 7 days after the initial surgery (day 0). In the acute behavioral experiments, SeeDB-Live/ACSF-HEPES was maintained on the brain surface during the behavioral test. The cranial window was replaced with a new one after SeeDB-Live/ACSF-HEPES treatment at day 0 for chronic behavioral assays (**n**–**p**). **m**, Mouse locomotor activity on a treadmill was measured for 10 min during clearing with SeeDB-Live/ACSF-HEPES in head-fixed awake animals. The total distance traveled and the maximum speed of mice treated with control ACSF-HEPES and SeeDB-Live/ACSF-HEPES were compared. $n = 5$ mice. NS (Wilcoxon signed-rank test). **n**, Locomotion assay. Total distances traveled by mice in an open chamber at 1, 4 and 7 days after treatment with control ACSF-HEPES and SeeDB-Live/ACSF-HEPES are shown. NS ($P \geq 0.05$; two-sided Wilcoxon rank-sum test). $n = 4$ mice per group. **o**, Motor function was examined with the wire hanging test[55]. We used a unilateral cortical ischemia model as a control. Fall time of mice in the wire hanging test at 1, 4 and 7 days after treatment with ACSF-HEPES, Rose Bengal and SeeDB-Live/ACSF-HEPES. $n = 4$ mice per group. ***$P < 0.0001$; NS ($P \geq 0.05$; two-sided Tukey–Kramer multiple-comparison test). **p**, Food consumption of mice treated with control ACSF-HEPES, unilateral ischemia and SeeDB-Live/ACSF-HEPES. $n = 4$ mice per group. ***$P < 0.001$; **$P < 0.01$; NS ($P \geq 0.05$; two-sided Tukey–Kramer multiple-comparison test). Graphs show the mean ± s.d. or median ± IQR. Images show representatives of ≥2 trials except for **k** (single trial). See Supplementary Table 4 for detailed statistical data. Panels **a** and **m** created in BioRender. Imai, T. (2026) https://BioRender.com/gyynf4j.

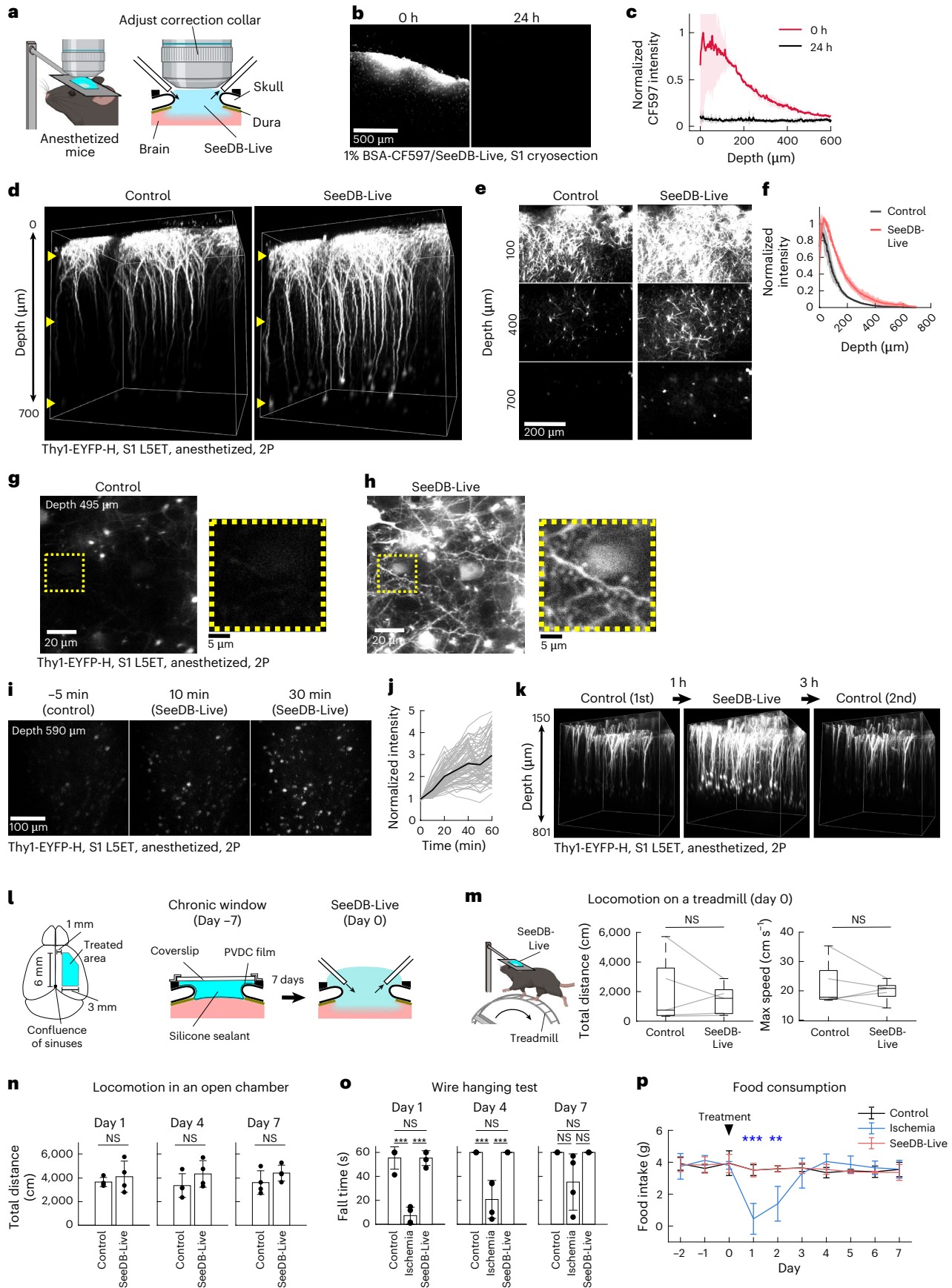

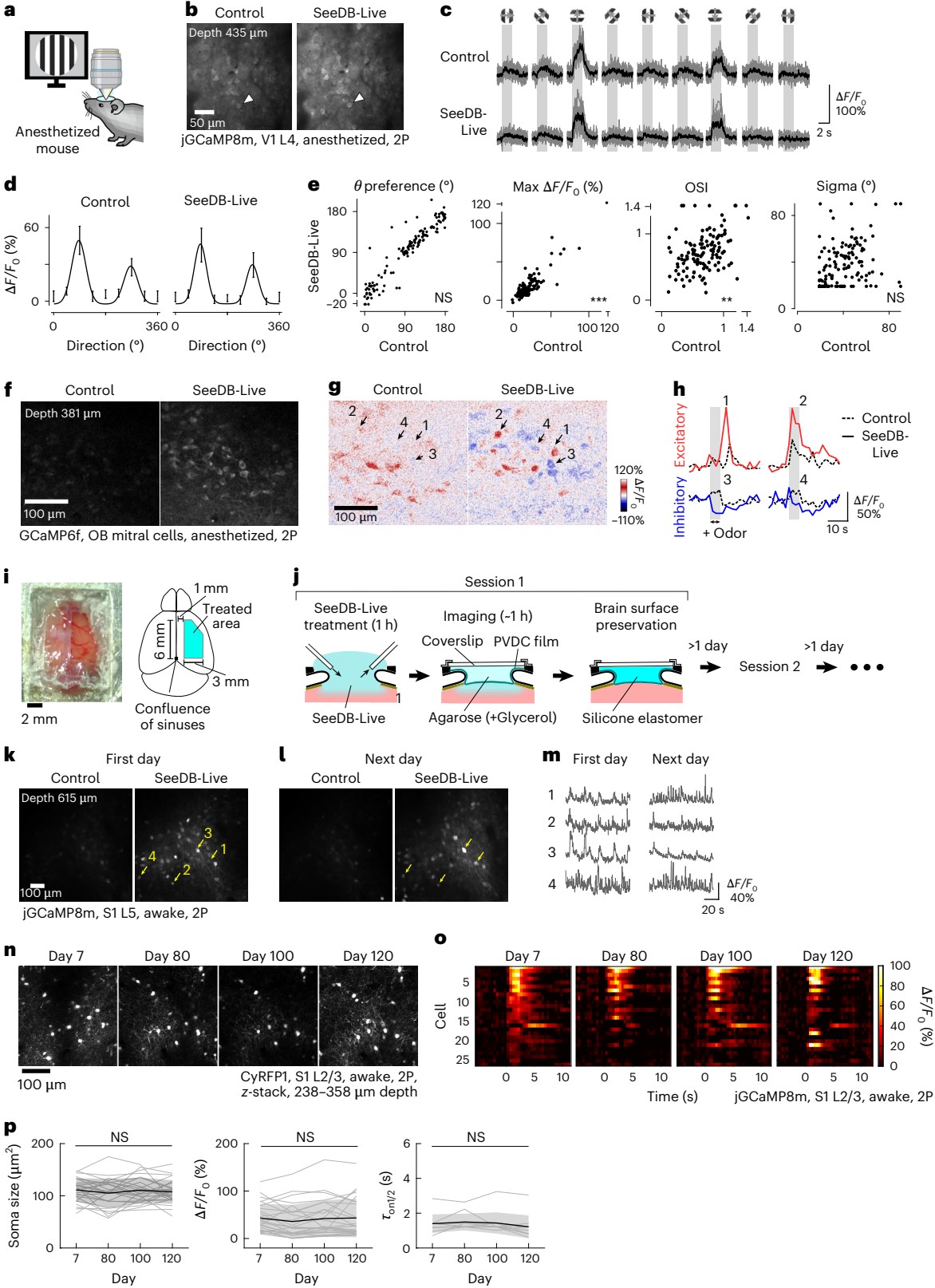

somata of layer 5 pyramidal neurons located at a depth of ~560 μm using JEDI-2P indicator (Extended Data Fig. 8f–i)[36].

Optical clearing with SeeDB-Live is transient in vivo (~1 h) as BSA is gradually washed out (Fig. 4k). To perform chronic calcium imaging with SeeDB-Live in awake animals, we used easily removable plastic films for the cranial window (Fig. 5i,j). A large cranial window

(6 × 3 mm²) was made in one hemisphere and SeeDB-Live was applied for 1 h. A polyvinylidene chloride (PVDC) wrapping film was attached onto the window[37]. Refractive index–matched agarose was placed between the PVDC film and a coverslip for imaging. We obtained stable responses of GCaMP8m in S1 (615-μm depth). Since the window is easily detached, we could repeat the same procedures on consecutive days

**Fig. 5 | Improved in vivo imaging of neuronal activity with SeeDB-Live.**
**a**–**e**, Two-photon calcium imaging of L4 neurons expressing jGCaMP8m (AAV-DJ-Syn-jGCaMP8m-WPRE) in V1 before and after clearing with SeeDB-Live/ACSF-HEPES. Anesthetized animals were used. **a**, Experimental setup. Drifting gratings of various orientations were presented to anesthetized mice. **b**, Basal fluorescence of jGCaMP8m without visual stimulation. L4 neurons at a depth of 435 µm. **c**,**d**, Responses of a representative L4 neuron (indicated by arrowheads in **b**) to visual grating stimuli before and after clearing with SeeDB-Live/ACSF-HEPES (**c**). The tuning curve was fitted with the sum of two Gaussian curves (**d**). **e**, Preferred orientation, maximum responses ($\Delta F/F_0$), OSI and tuning width (Sigma) for the same set of L4 neurons (136 neurons from three mice) before (*x* axis) and after (*y* axis) clearing with SeeDB-Live/ACSF-HEPES. The comparison was performed as described previously[56]. ***$P < 0.001$; **$P < 0.01$; NS (two-sided Wilcoxon signed-rank test). **f**–**h**, Odor responses of mitral cells in the olfactory bulb in anesthetized mice. **f**, *x*–*y* images of mitral cells in the olfactory bulb of a Thy1-GCaMP6f mouse (4-month-old, anesthetized) was imaged before and after clearing with SeeDB-Live/ACSF-HEPES (1 h after clearing) with two-photon microscopy. The depth was 381 µm. The correction collar of the objective lens was turned to the best position. **g**, Odor responses of mitral cells upon 1% valeraldehyde. The odor was delivered to a mouse nose for 5 s at 1 l min⁻¹. Arrows indicate the same sets of neurons. **h**, Representative excitatory/inhibitory responses of mitral cells indicated in **g**. **i**–**p**, Chronic imaging in awake animals using repeated SeeDB-Live treatment. **i**, A photo (left) and schematic diagram (right) of a large cranial window with a PVDC wrapping film. **j**, After clearing with SeeDB-Live/ACSF-HEPES for 1 h, the brain surface was covered with the PVDC film. Between the film and the glass coverslip, 1.5% (wt/vol) agarose was

applied with and without 19.3% (wt/vol) glycerol (refractive index, 1.363) for the SeeDB-Live and control conditions, respectively. For objective lens immersion, 15.6% (vol/vol) TDE/ddH₂O (refractive index, 1.363) and ddH₂O were used for SeeDB-Live and the control, respectively. The correction collar of objective lens was turned to the best position. Imaging was performed within 1 h after SeeDB-Live treatment. For chronic imaging, a silicone elastomer was filled between the coverslip and the plastic film until the next imaging session. **k**, Basal fluorescence (temporal median) of jGCaMP8m-expressing L5 neurons at day 0. Awake animals were imaged. The depth was 615 µm. L5 neurons in S1 were labeled with AAV-jGCaMP8m-P2A-CyRFP1. **l**, Basal fluorescence of the same set of L5 neurons on the next day. **m**, Representative Ca²⁺ responses of L5 neurons indicated in **k** and **l** during repeated whisker stimulations with air puffs. **n**–**p**, Long-term monitoring of neuronal morphology and physiology in awake mice. Data are from a representative animal of four trials. L2/3 neurons in S1 were labeled with AAV-jGCaMP8m-P2A-CyRFP1. *z*-stack images (imaging depth: 238–358 µm) of the CyRFP1 fluorescence of L2/3 neurons after SeeDB-Live treatment on days 7, 80, 100 and 120 (**n**). Mean calcium responses of jGCaMP8m-expressing L2/3 neurons to whisker stimulations (five times) with air puffs (**o**). Soma size, $\Delta F/F_0$ and half-rise time of neurons to whisker stimulation after SeeDB-Live treatment on days 7, 80, 100 and 120 (**p**). $\Delta F/F_0$ and half-rise (τ) time were calculated from the mean responses to five whisker stimulations. Half-rise time was analyzed only for cells whose maximum $\Delta F/F_0$ was greater than the mean + 5 s.d. of $F_0$ on all time points. NS ($P ≥ 0.05$; two-sided repeated-measures analysis of variance). Data in **c**, **d** and **p** indicate the mean ± s.d. Images show representative samples of 2–4 trials. See Supplementary Table 4 for detailed statistical data. Panel **a** reated in BioRender. Imai, T. (2026) https://BioRender.com/gyynf4j.

without compromising the quality of the window (Fig. 5k–m). We did not find any changes in cytoarchitecture and sensory responses (amplitude and frequency) over 4 months, suggesting that normal neuronal functions are maintained during repeated clearing (Fig. 5n–p and Extended Data Fig. 7a). Moreover, we observed minimal inflammatory responses after repeated clearing (Extended Data Fig. 7l). This approach could be powerful for chronic imaging of deep cortical regions.

## Epifluorescence voltage imaging of acute brain slices with SeeDB-Live

Genetically encoded voltage indicators with high signal-to-noise ratios have been developed in recent years. To image fast voltage

changes, high-speed epifluorescence imaging is advantageous over point-scanning two-photon microscopy. Here we cleared acute olfactory bulb slices with SeeDB-Live and imaged calcium and voltage signals using GCaMP6f and a fast and sensitive chemigenetic voltage indicator, Voltron2, sparsely introduced to mitral/tufted cells by in utero electroporation (Fig. 6a)[38]; Voltron2 was visualized with JF₅₄₉–HaloTag ligand applied to the medium (Voltron2₅₄₉). After the clearing with SeeDB-Live, the epifluorescence signals of Voltron2₅₄₉ were clearly visualized at a depth of >150 µm (Fig. 6b). Using a high-speed CMOS camera (2 kHz), we recorded voltage changes in different compartments of mitral cell dendrites. We could visualize the backpropagation of action potentials from somata to dendritic tips in single-shot

**Fig. 6 | Epifluorescence voltage imaging with SeeDB-Live ex vivo and in vivo.** **a**–**f**, Voltage imaging of a mitral cells in acute brain slices ex vivo using epifluorescence microscopy. **a**, Mitral cells in olfactory bulb slices. OSN, olfactory sensory neuron. **b**, Mitral cells labeled with Voltron2₅₄₉ at different depths under control and SeeDB-Live conditions (acquired with a high-speed CMOS camera, temporal median). Voltron2 was introduced to mitral cells by in utero electroporation and analyzed at P11. Voltron2 was labeled with Janelia Fluor HaloTag Ligand 549 before the imaging. **c**, Representative traces of Voltron2₅₄₉ signals at a depth of 150 µm. Ticks indicate the detected action potentials. **d**, Two-photon image identified a labeled mitral cell (*z*-stacked, left). Epifluorescence of Voltron2₅₄₉ (temporal median) is shown on the right. Voltron2 and GCaMP6f were introduced to mitral cells by in utero electroporation. **e**, Backpropagation of action potentials were imaged at 2 kHz (single-shot images) using a high-speed CMOS camera. Representative −$\Delta F/F_0$ images are shown. ROI was manually cropped based on 2P and epifluorescence images shown in **d**. The arrow indicates the initiation of the action potential. **f**, Spatiotemporal pattern of backpropagating action potentials, averaged from 65 events. The half-rise time of action potentials at soma was defined as 0 ms. Median filtering (4 × 4 pixels) was applied to the images. **g**–**p**, Epifluorescence voltage imaging in the mouse olfactory bulb in vivo. **g**, Schematic diagram of the epifluorescence voltage imaging of mitral/tufted cells. AAV-syn-FLEX-Volton2 was injected into the olfactory bulb of *Pcdh21*-Cre mice before the imaging experiments. After durotomy, the olfactory bulb was immersed with ACSF-HEPES containing 50 nM Janelia Fluor HaloTag Ligand 549 for 1 h, followed by a 1-h washout in ACSF-HEPES. SeeDB-Live treatment was then performed for 1 h. We imaged a deeper part of the glomerular layer (90 µm), where mitral/tufted cells form dendritic branches. A focal plane is shown as a yellow line (arrows). We used

a ×25 objective (NA 1.05) with a short focal depth (1.36 µm) to minimize out-of-focus signals. **h**,**i**, Epifluorescence images (temporal median) of dendrites (and some somata) of mitral/tufted cells labeled with Voltron2₅₄₉ in an anesthetized mouse (4-month-old) before (**h**) and after (**i**) clearing with SeeDB-Live (1 h after clearing). $F_0$ images at a depth of 90 µm. Magnified images are shown on the right. **j**, ROIs were semiautomatically detected from the $F_0$ image using ilastik. Representative traces (−$\Delta F/F_0$) from highlighted ROIs (dendrites) are shown on the right. Ticks indicate the detected action potentials. Note that subthreshold activities were also correlated between ROIs within the same glomerulus. **k**, Cross-correlation matrix for voltage traces. ROIs were clustered using *k*-means clustering. The cluster number *k* was defined based on the number of glomeruli. **l**, Spatial distribution of ROIs in each cluster. Each color represents a different cluster. **m**,**n**, Subclusters based on spike synchronicity between ROIs within a glomerulus (cluster 2; **m**) and representative traces from indicated ROIs (**n**). The black trace on the top (mean) shows the averaged −$\Delta F/F_0$ of all glomeruli indicating sniff-coupled theta waves in the olfactory bulb. **o**,**p**, Comparison of synchronicity between the subclusters in the same or different glomeruli (clusters 2 and 11; **o**) and representative traces of the indicated subclusters (**p**). The red ticks indicate the synchronous events that coincided with spikes in the C2-2 subcluster. The synchronicity index indicates the proportion of synchronous events normalized by the spike frequency. Odor (1% amyl acetate) was delivered to a mouse nose for 5 s (shaded) at 1 l min⁻¹. The black trace on the top (mean) shows the averaged −$\Delta F/F_0$ of all glomeruli indicating sniff-coupled theta waves in the olfactory bulb. Data are from representative samples of two trials each. See Supplementary Table 4 for detailed statistical data. Panel **g** created in BioRender. Imai, T. (2026) https://BioRender.com/gyynf4j.

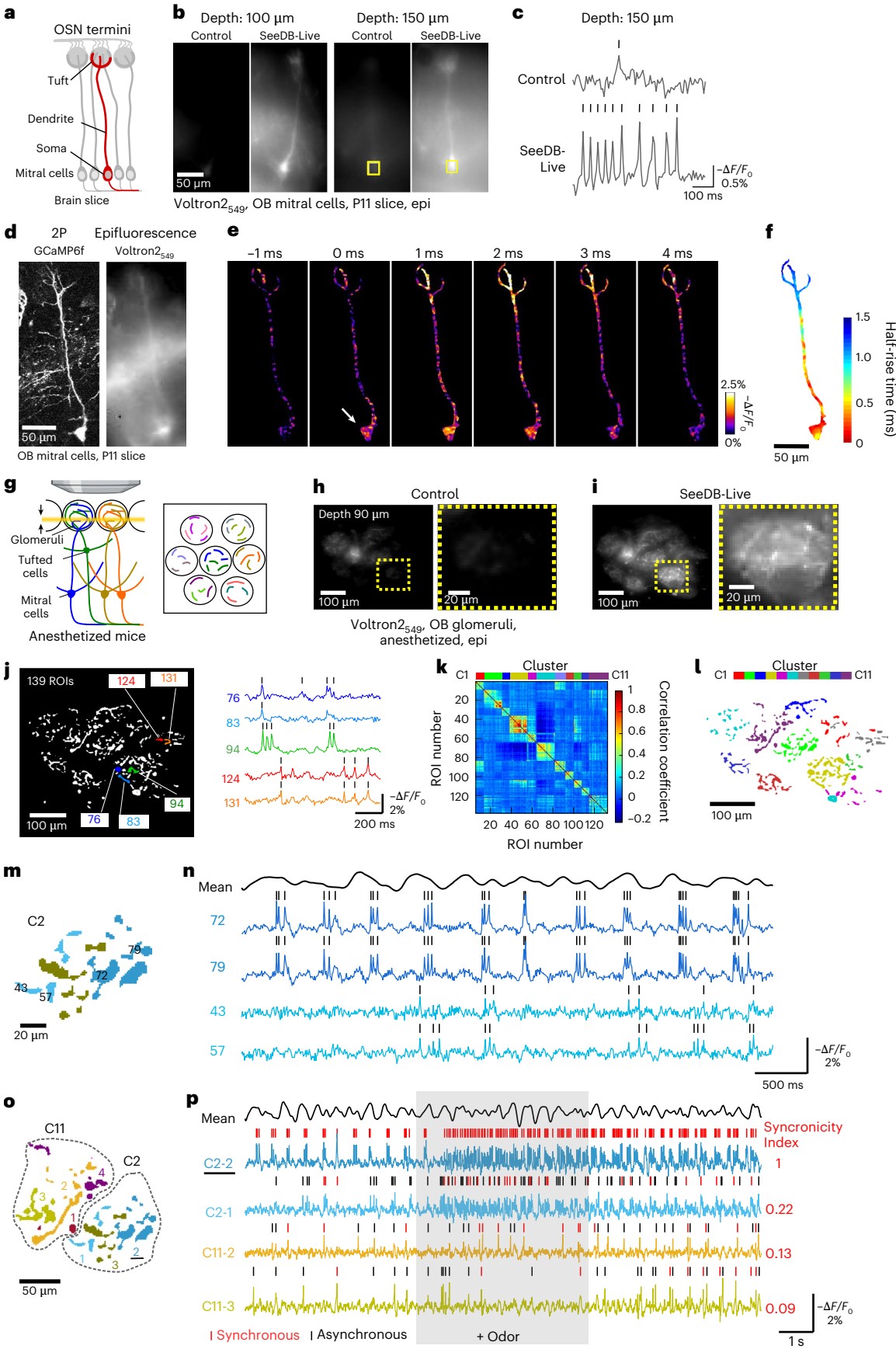

imaging, without averaging (Fig. 6c–f, Extended Data Fig. 9a–c and Supplementary Video 16). We observed a ~1.5-ms delay in responses at the tip of the primary dendrites (Fig. 6f). Thus, the combination of SeeDB-Live and epifluorescence imaging will be a powerful tool for studying subcellular dynamics of voltage signals in acute brain slices.

### Epifluorescence voltage imaging of backpropagating action potentials in vivo

Previously, it has been challenging to image genetically encoded voltage indicator signals at deeper parts of the brain using epifluorescence imaging in vivo[38–40]. We expressed Voltron2-ST specifically in layer 2/3 (L2/3) pyramidal neurons in S1 using in utero electroporation, visualized with $JF_{549}$–HaloTag ligand. After the clearing with SeeDB-Live, L2/3 neurons located at a depth of 120–150 µm were better visualized, allowing for reliable detection of spontaneous action potentials in their somata (Extended Data Fig. 9d–f). We also detected backpropagating action potentials from dendrites of Voltron2-expressing L2/3 neurons in awake mice (Extended Data Fig. 9g–i).

Next, we performed voltage imaging of mitral/tufted cells in vivo. In the mitral/tufted cells, the odor information is encoded not only by the spike frequencies, but also by the timing[41]. Epifluorescence voltage imaging has been performed in the olfactory bulb, but not at the single-neuron resolution[42,43].

We expressed Voltron2 specifically and sparsely in mitral/tufted cells using the *Pcdh21*-Cre driver and a Cre-dependent adeno-associated virus (AAV) vector, labeled with $JF_{549}$–HaloTag ligand ($Voltron2_{549}$). $Voltron2_{549}$ signals were found not only in somata, but also in dendrites (Fig. 6g–i). As a result, we were able to detect backpropagating action potentials from ~140 neurites (regions of interest or ROIs) at a depth of ~90 µm using SeeDB-Live and epifluorescence imaging (Fig. 6j). In this imaging setup, the theoretical focal depth was ~1.36 µm (Fig. 6g). Of course, out-of-focus signals will contribute substantially to the total fluorescence, $F_0$. However, as the fluorescence changes caused by the spikes were all-or-none and up to 2–3% $\Delta F/F_0$ (much smaller than calcium imaging), it is unlikely that scattered out-of-focus signals interfere with or contaminate spike detection (that is, $-\Delta F/F_0$) in each of the ROIs.

Correlation of the voltage traces across ROIs revealed ~11 discrete clusters (Fig. 6k). The ROIs within each cluster (11 clusters by *k*-means) were also spatially clustered (Fig. 6l), demonstrating that neurites within the same glomerulus have similar voltage dynamics (including subthreshold changes; Fig. 6j). This makes sense because neurons connecting to the same glomerulus ('sister' mitral/tufted cells) receive similar synaptic inputs and are electrically coupled within the glomerulus[44]. When we looked at individual ROIs within the same glomerulus, some pairs, but not all, demonstrated highly correlated backpropagating action potentials, suggesting that these dendritic branches originated from the same neuron (Fig. 6m,n). We, therefore, grouped ROIs with highly synchronized backpropagating action potentials into a subcluster. In this way, we obtained 21 subclusters, each of which most likely represents a single neuron (Extended Data Fig. 10a,b). Subclusters that belong to the same glomerulus ('sister' mitral/tufted cells) tend to show more synchronized events than those in different glomeruli (Fig. 6o,p)[44]. We also observed odor-evoked phase shifts in action potentials relative to the sniff-coupled theta oscillations (Extended Data Fig. 10c,d), consistent with previous studies[45,46]. Thus, epifluorescence imaging of dendrites combined with SeeDB-Live provides a powerful approach for studying population voltage dynamics in vivo.

## Discussion

To date, several studies have achieved optical clearing of live tissues, but only under unhealthy conditions for live cells. In this study, we identified the optimal refractive index and achieved optical clearing of live tissues without affecting osmolarity using BSA. Furthermore, the extracellular ionic condition was largely preserved, which is critical for studying the normal physiology of neurons. This is an advantage of SeeDB-Live over existing methods (Supplementary Table 5). Notably, the SeeDB-Live treatment demonstrated an undetectable level of toxicity to neuronal physiology and animal behavior, providing a powerful new option for imaging-based neurophysiology. Combined with wider field-of-view two-photon microscopy[47–50], targeted one-photon imaging approaches[39,40,51] and red-shifted indicators[52], SeeDB-Live expands the imaging scale for biological phenomena at the tissue and organ scale both ex vivo and in vivo.

In this study, we demonstrated that SeeDB-Live is particularly useful for epifluorescence voltage imaging. Previously, large-scale imaging of voltage changes has been difficult due to the slow scanning speed of two-photon microscopy and light scattering with one-photon microscopy. Epifluorescence imaging of dendritic voltage changes with SeeDB-Live could be a powerful strategy for studying subcellular and/or population-scale voltage dynamics both ex vivo and in vivo.

We have demonstrated the utility of SeeDB-Live for acute functional assays of organoids. However, some of the induction experiments (for example, optic cup formation[31]) were unsuccessful for unknown reasons. For the improved culture of organoids, microfluidic culture systems may be useful to improve circulation and/or to exchange the medium[30]. For the in vivo applications, accessibility of BSA to the target tissues may be the major issue. For the chronic in vivo imaging of the mouse brain, we demonstrate the utility of easily removable cranial windows[37]. In the future, BSA-permeable membrane may be more useful for the optical window. Alternatively, infusion into the CSF circulation system may be useful for efficient permeation of SeeDB-Live for more extensive clearing of the entire brain in the future studies[53]. Other organs may be more difficult to clear, and future in vivo applications would require strategies to overcome the accessibility issue.

The improved transparency with SeeDB-Live also expands the modality of the imaging methods. With SeeDB-Live, we can now use confocal microscopy for deep imaging, enabling high-resolution multicolor imaging. The combination of fluorescence imaging and photostimulation will be easier with one-photon and SeeDB-Live than with a multi-photon setup. As optical aberration is minimized, SeeDB-Live should also be very useful for super-resolution imaging of large volume in live tissues[9,54]. While we have demonstrated shadow imaging with two-photon microscopy, STED microscopy in combination with SeeDB-Live may enable saturated connectomics in acute brain slices, rather than in cultured brain slices[33,34]. For the best performance, it should be important to use objective lenses optimized for SeeDB-Live (refractive index, ~1.363). Together with ongoing efforts to develop microscopy techniques, our live tissue-clearing approach facilitates our understanding of the tissue-scale and organ-scale dynamics of biological phenomena.

## Online content

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

[1]Department of Developmental Neurophysiology, Graduate School of Medical Sciences, Kyushu University, Higashi-ku, Fukuoka, Japan. [2]FOREST, Japan Science and Technology Agency, Saitama, Japan. [3]Department of Physiology, Graduate School of Medical and Dental Sciences, Kagoshima University, Kagoshima, Japan. [4]Department of Pharmacology, Graduate School of Medical and Dental Sciences, Kagoshima University, Kagoshima, Japan. [5]Nano Life Science Institute, Kanazawa University, Kanazawa, Japan. [6]Department of Neurophysiology, Faculty of Medicine, University of Yamanashi, Chuo, Japan. [7]Department of Stem Cell Biology and Medicine, Graduate School of Medical Sciences, Kyushu University, Higashi-ku, Fukuoka, Japan. [8]Department of Optical Neural and Molecular Physiology, Graduate School of Biostudies, Kyoto University, Kyoto, Japan. [9]Laboratory of Optical Biomedical Science, Institute for Life and Medical Sciences, Kyoto University, Kyoto, Japan. [10]Center for Living Systems Information Science, Graduate School of Biostudies, Kyoto University, Kyoto, Japan. [11]Department of Genome Biology, Graduate School of Medicine, Osaka University, Suita, Japan. [12]Division of Reproductive Systems, Premium Research Institute for Human Metaverse Medicine (WPI-PRIMe), Osaka University, Suita, Japan. [13]CREST, Japan Science and Technology Agency, Saitama, Japan. ✉e-mail: inagaki.shigenori.570@m.kyushu-u.ac.jp; imai.takeshi.457@m.kyushu-u.ac.jp

## Methods

### Mice

All animal experiments were approved by the Institutional Animal Care and Use Committee (IACUC) of Kyushu University, Kagoshima University and Yamanashi University. *Thy1-GCaMP6f* (line GP5.11; JAX, 024339)[57], *Thy1-YFP-H* (JAX, 003782)[58] and *Pcdh21-Cre* (RIKEN BRC, RBRC02189)[59] mice have been described previously. ICR and C57BL/6N mice were purchased from Japan SLC. *Thy1-GCaMP6f* (line GP5.11; hemizygotes), *Thy1-YFP-H* (homozygotes) and *Pcdh21-Cre* (hemizygotes) mice were on the C57BL/6N background. Both males and females were used for our experiments. Mice were kept under a consistent 12-h light–12-h dark cycle (lights on at 8:00 and off at 20:00), with an ambient temperature of 20–26 °C and humidity of 40–70%.

### Plasmids

To construct pCAG-GCaMP6f, GCaMP6f gene was PCR amplified from pGP-CMV-GCaMP6f (Addgene, 40755) with Q5 High-Fidelity 2X Master Mix (M0492S, NEB). The cDNA was flanked by *EcoR*I and *Not*I sites. The GCaMP6f cDNA was subcloned into pCAG vector with a ligation kit (6023, Takara). To make pAAV-CAG-jGCaMP8f-WPRE, jGCaMP8f gene was amplified from pGP-AAV-syn-jGCaMP8f-WPRE (Addgene, 162376) with Q5 High-Fidelity 2X Master Mix. The cDNA contained an extra 20–30-bp overlap regions with pAAV-CAG-tdTomato (Addgene, 59462). The tdTomato was removed from pAAV-CAG-tdTomato by digestion with *Kpn*I and *Hind*III, and jGCaMP8f cDNA with the extra sequence was subcloned into the vector with NEBuilder HiFi DNA Assembly (E2621S, NEB). pCAG-GCaMP6f plasmid has been deposited to Addgene (no. 249680).

### Preparation of SeeDB-Live

Control ACSF comprised 125 mM NaCl, 3 mM KCl, 1.25 mM NaH$_2$PO$_4$, 2 mM CaCl$_2$, 1 mM MgCl$_2$, 25 mM NaHCO$_3$ and 25 mM glucose. In some experiments, we used ACSF-HEPES (145 mM NaCl, 5 mM KCl, 2 mM CaCl$_2$, 1 mM MgCl$_2$, 10 mM HEPES, pH 7.3).

To prepare SeeDB-Live/ACSF, crystallized BSA from bioWORLD (22070004; BSA#1) was used. BSA was dissolved with gentle shaking at 15.6% wt/vol. We found that BSA#1 contained residual salts (~30 mM Na$^+$ and ~1 mM Ca$^{2+}$ when dissolved at 15% wt/vol; Extended Data Fig. 1e), and this was taken into account. pH was adjusted with sodium hydroxide. BSA is known to chelate Ca$^{2+}$ and Mg$^{2+}$. To maintain the free Ca$^{2+}$ and Mg$^{2+}$ concentrations the same as ACSF, additional CaCl$_2$ (2 mM) and MgCl$_2$ (1 mM) were supplemented after BSA was fully dissolved in the medium (except for Fig. 1a–h and Extended Data Fig. 1). SeeDB-Live/ACSF contained 151.0 mM Na$^+$, 111.3 mM Cl$^-$, 3.0 mM K$^+$, 1.0 mM H$_2$PO$_4$, 6.1 mM Ca$^{2+}$, 2.9 mM Mg$^{2+}$, 20.1 mM HCO$_3^-$ and 15 mM glucose (pH 7.4 under 5% CO$_2$; refractive index, 1.363; Supplementary Table 2). SeeDB-Live/ACSF-HEPES contained 157.4 mM Na$^+$, 131.9 mM Cl$^-$, 4.0 mM K$^+$, 6.1 mM Ca$^{2+}$, 2.5 mM Mg$^{2+}$ and 7.8 mM HEPES (pH 7.3; refractive index, 1.363). To oxygenate SeeDB-Live/ACSF and SeeDB-Live/ACSF-HEPES, 95% O$_2$/5% CO$_2$ gas was filled in the bottle containing the medium for ~2 h before the experiments. Saturation of O$_2$ was checked with an O$_2$ sensor (9521, Horiba). Bubbling is not recommended because it produces a lot of foam, and BSA may be denatured. As for the culture medium, BSA#1 was dissolved in ×0.8 culture medium to adjust the osmolarity and CaCl$_2$ (2 mM) and MgCl$_2$ (1 mM) were supplemented. In this way, the concentrations of salts and osmolality of the BSA-containing medium/ saline were adjusted to be the same as that of the original medium/ saline (Supplementary Table 1). The osmolarity of the BSA solution was measured with a vapor pressure osmometer (VAPRO 5600, Xylem ELITech). The refractive index was measured with an Abbe refractometer (ER-2S, Erma) with a white LED light source.

We tested the following BSA products: BSA#1, BSA crystal (22070004, bioWORLD); BSA#2, BSA Low Salt (015-15125, Fujifilm), BSA#3, BSA crystal (012-15093, Fujifilm); BSA#4, BSA pH 5.2 (017-21273, Fujifilm); BSA#5, BSA pH 7.0 (019-27051, Fujifilm); BSA#6, BSA Globulin

Free (016-15111, Fujifilm); and BSA#7, BSA Protease Free (019-28391, Fujifilm). Salts contained in BSA powder were analyzed using Inductively Coupled Plasma Mass Spectrometry (Agilent Technologies, ICP-MS 7700x). See Extended Data Fig. 1e for the data.

We tested the following chemicals during the screening process: glycerol (17018-25, Nacalai), DMSO (043-07216, Fujifilm), propylene glycol (164-04996, Fujifilm), iodixanol (VISIPAQUE 320 INJECTION 50 ml, GE HealthCare), iodixanol (D1556-250ML, Optiprep), iotrolan (Isovist Injection 300, Bayer Pharma Japan), iopamidol (OYPALOMIN, FujiPharma), iopromide (iopromide 370 Injection (FRI), Fujifilm), iohexol (OMNIPAQUE350 INJECTION, GE healthcare Pharma), ioxilan (Imagenil350 Injection, Guerbet Japan), ioversol (Optiray350 Injection, Mallinckrodt), ioxagilic acid (Hexabrix320 Injection, Guerbet Japan), iomeprol (Iomeron400 Bracco-Eisai), Ficoll70 (17031050, Cytiva), Ficoll400 (17030010, Cytiva), HBCD (307-84601, Glico), PVP (P0471, TCI), sucrose (193-00025, Fujifilm), tartrazine (T0388, Sigma-Aldrich), ampyron (017-02272, Fujifilm), polydextrose (polydex300, Nichiga), partially hydrolyzed guar gum (2021092403, Nichiga), methyl-β-cyclodextrin (M1356, TCI), agave inulin (agabe500, Nichiga), PEG8000 (V3011, Promega), PEG10000 (81280, Sigma-Aldrich), RM + stevia (dex-5-500m, Nichiga), resistant maltodextrin from corn (RM1, MK-H108-6T6I, Nichiga), resistant maltodextrin from wheat (RM2, dekisutorin-komugi-400, Nichiga), inulin (inurinn500, Nichiga), isomaltodextrin (Fibryxa, Hayashibara), reduced resistant maltodextrin (kg-nandeki-400, Nichiga), dextran (D1662, Sigma-Aldrich) and stevia (sutebiasw5-150m, Nichiga).

See Supplementary Tables 1 and 2 for more detailed composition of SeeDB-Live and other clearing media. A step-by-step protocol and technical tips are available at SeeDB Resources (https://sites.google.com/site/seedbresources/).

### Transmittance measurement in cell suspension

HeLa S3 cells (JCRB9010, JCRB) were cultured in Dulbecco's Modified Eagle Medium (DMEM high glucose; 043-300085, Fujifilm) supplemented with 1% penicillin–streptomycin and 10% fetal bovine serum (FBS) at 37 °C, 5% CO$_2$. After trypsinization, cells were collected at $2 \times 10^5$ cells per tube. After centrifugation, the medium was replaced with 50 µl of index-adjusted PBS. Transmittance of the cell suspension in 400–1,100 nm was measured with a ratio beam spectrophotometer (U-5100, Hitachi High-Tech). This measurement was performed quickly because unhealthy cells have a nonoptimal refractive index, which results in reduced transmittance.

### Ca$^{2+}$ measurement in cultured cells

HEK293T cells (AAVpro 293T, 632273, Takara) were cultured in DMEM (high glucose) supplemented with 1% penicillin–streptomycin and 10% FBS at 37 °C and 5% CO$_2$. Cells seeded in 35-mm glass-bottom dishes (60% confluent) were transfected with pGP-CMV-GCaMP6f (Addgene, 40755) using PEI Max (Polysciences, 24765-1). Twenty-four hours after transfection, the medium was replaced with an index-adjusted medium (refractive index, 1.365). The osmolality of the medium was not adjusted to isotonicity. Two hours after the medium exchange, cells were imaged with an inverted microscope (DMI600B, Leica) equipped with a ×10 NA 0.4 dry objective lens and controlled by LAS AF software (Leica). A final concentration of 50 µM ATP was added to the medium during imaging. The maximum values after stimulation were used for the data analysis. For the calcium measurement with a plate reader (TriStar LB941, Berthold), GCaMP6f-expressing HEK293T cells were transferred to a 96-well plate at $4 \times 10^5$ cells per ml per well. A total of 50 µM ATP was added to the medium during the time-series measurement. Mean values during 1–10 s after stimulation were analyzed.

### Viscosity measurement

To measure the viscosity of the solutions, we measured the time taken for 20 ml of solutions to flow out from a 50-ml syringe (TERMO,

SS-50ESZ) with an internal tip diameter of ~2 mm. The point at which the solution flow was stopped was considered as the end of the flow. The viscosity of sucrose solutions at room temperature (21 °C)[60] was used as the standard. The plots were fitted by single-exponential fitting. The viscosity of the solutions was calculated based on the calibration curve.

## Cell growth measurement

HeLa/Fucci2 cells were seeded on a clear-bottom 384-well plate at 700 cells per well. The cells were cultured in DMEM (high glucose), without phenol red, and glutamine (040-30095, Fujifilm) supplemented with 1% penicillin–streptomycin, 10% FBS and 1% glutamine at 37 °C and 5% $CO_2$. Twenty-four hours after seeding, the medium was replaced with an index-adjusted medium (refractive index, 1.363; 310–320 mOsm $kg^{-1}$). Fucci2 fluorescence (mVenus and mCherry) was imaged with an inverted fluorescence microscope (DMI600B, Leica) equipped with a ×5 NA 0.1 dry objective lens and controlled by LAS AF software (Leica). Cells were counted based on the nucleus images of Fucci2 fluorescence with ImageJ software (https://imagej.net/ij/). First, the green and red channels were summed. The speckle noise was removed with 'Despeckle'. The overlapped nuclei were separated with 'Watershed'. The intensity threshold was determined manually. Finally, the cell number in each well was counted with 'Analyze particles'. For manual counting, the cells were seeded on 35-mm dishes at $1 \times 10^5$ cells per dish. Twenty-four hours after plating, the medium was replaced with an index-adjusted medium (refractive index, 1.363; 320 mOsm $kg^{-1}$). After trypsinization, the cell number was counted with a hemocytometer.

For spheroid formation, HeLa/Fucci2 cells were seeded on an ultralow-attachment 96-well plate (7007, Corning) at 1,000 cells per well. At 24–48 h after plating, the spheroid was incubated in an index-adjusted medium (SeeDB-Live; refractive index, 1.366; 320 mOsm $kg^{-1}$) for 4 h per day or for all the time. The half volume of the culture medium was replaced with the fresh one every day in Fig. 1i–k. For manual counting, the cells were incubated in a mixture of 50 μl DMEM and 200 μl Trypsin-EDTA for 30 min. The suspension was then centrifuged at 1,000 rpm for 5 min. The pellet was resuspended with the culture medium for cell counting with a hemocytometer.

## Spheroid imaging

The HeLa/Fucci2 spheroid was incubated in SeeDB-Live (refractive index, 1.366; 320 mOsm $kg^{-1}$) for 1 h. Then, the spheroid was mounted on a glass slide and sealed with a 1-mm-thickness silicone rubber spacer (Togawa rubber) and a coverslip (Matsunami). Imaging was performed using an FV1000MPE microscope (Olympus/Evident) with Fluoview FV10-ASW software (Olympus/Evident, RRID: SCR_014215) and a ×25 NA 1.05 objective lens (Olympus/Evident, XLPLN25XWMP). Immersion was performed with water and 17.2% (vol/vol) TDE/ddH₂O (refractive index 1.366) for control and SeeDB-Live samples, respectively. The correction collar was turned to the appropriate position. For the confocal imaging, 473-nm and 569-nm lasers were used to excite mVenus and mCherry, respectively. For the two-photon imaging, a femtosecond laser (Insight DeepSee, SpectraPhysics) was tuned to 920 nm for mVenus excitation. A 1,040-nm laser was used for mCherry excitation.

For cell detection, flat areas of the images were cropped. The green and red channels were merged to make reference images. A median filter was applied (2 × 2 pixels). ROIs for each cell in a spheroid were created with Cellpose[61,62]. Cell numbers and the fluorescence intensity were calculated based on the ROIs using MATLAB (MathWorks). 3D-rendered images were made by Imaris Viewer (Oxford Instruments).

## Imaging of intestinal organoids

Intestinal organoids were created following the manufacturer's protocol (VERITAS). Briefly, ePET-Cre; Ai162 mice were euthanized with an overdose of pentobarbital (intraperitoneal (i.p.) injection, 100–150 mg per kg body weight). A small intestine was taken out and cut to expose the lumen side. The lumen was gently washed with cold PBS (−) several

times. The small intestine was cut into 2-mm pieces in 10 ml of cold PBS (−). After pipetting three times, the supernatant was replaced with new cold PBS (−). This procedure was repeated >15 times until the supernatant became clear. The supernatant was replaced with 25 ml of Gentle Cell Dissociation Reagent (ST-100-0485, STEMCELL Technologies). The pieces were gently shaken for 15 min at room temperature. The supernatant was replaced with 10 ml of cold 0.1% BSA/PBS. After pipetting three times, the suspension was passed through a 70-μm cell strainer (352350, Corning). This step was repeated to obtain the fraction containing more crypts. After centrifugation at 300g for 5 min at 4 °C, the supernatant was replaced with 10 ml of cold 0.1% BSA/PBS. After centrifugation at 200g for 3 min at 4 °C, the supernatant was replaced with 10 ml of DMEM/F-12 (11039-021, Thermo Fisher). After centrifugation at 200g, for 5 min at 4 °C, the supernatant was replaced with 150 μl of IntestiCult Organoid Growth Medium (ST-06005, STEM-CELL Technologies). Matrigel (150 μl; 356237, Corning) was added to the suspension. After pipetting ten times, 50 μl of the mixture was mounted on the well of a 24-well plate. The plate was incubated for 10 min at 37 °C to gelatinize Matrigel. IntestiCult Organoid Growth Medium (750 μl) was added to the wells carefully.

For imaging, the organoids were dissociated from a gel by pipetting with Gentle Cell Dissociation Reagent and transferred to a 15-ml tube. The tube was gently shaken for 10 min at room temperature. After centrifugation at 300g for 5 min at 4 °C, the supernatant was replaced with 10 ml of cold DMEM-F-12. After centrifugation at 300g for 5 min at 4 °C, the supernatant was replaced with 150 μl of Intesti-Cult Organoid Growth Medium (osmolality, 270 mOsm $kg^{-1}$). Matrigel (150 μl) was added to the suspension, and 50 μl of the mixture was mounted and spread on the glass region of a 35-mm glass-bottom dish. The mixture was gelatinized by incubation for 10 min at 37 °C. IntestiCult Organoid Growth Medium (1 ml) was added to the dish carefully. For clearing, the culture medium was replaced with SeeDB-Live/ IntestiCult Organoid Growth Medium (refractive index, 1.363) 2–3 h before imaging. Phase contrast images were taken with an inverted microscope (DMI600B, Leica) equipped with a ×10 NA 0.4 dry objective lens and controlled by LAS AF software (Leica). Fluorescence of EECs in an organoid was imaged with an inverted confocal microscopy (TCS SP8, Leica) equipped with ×20 NA 0.75 multi-immersion lens and LASX software (Leica Microsystems). Immersion was performed using water and 17.2% (vol/vol) TDE/ddH₂O (refractive index 1.366) for controls and SeeDB-Live samples, respectively. The correction collar was turned to the appropriate position. A 488-nm laser was used to excite GCaMP6s expressed in EECs. To measure the $Ca^{2+}$ responses of EECs, KCl (+30 mM at final concentrations) was added to the medium during imaging.

## Imaging of embryonic stem cell-derived organoids

Mouse embryonic stem cells were maintained as described in the previous study[31]. The cell line used in this study is a subline of the mouse embryonic stem cell line EB5 (129/Ola), in which the GFP gene was knocked in under the *Rax* promoter and Lifeact-mCherry gene was knocked in to the *Rosa26* locus. The cell line was provided by M. Eiraku at Kyoto University.

Cells were maintained as described in the previous study[31]. For maintenance, cells were cultured in a gelatin-coated 100-mm dish. The dish contained maintenance medium, to which 20 μl of $10^6$ units per ml leukemia inhibitory factor (Sigma-Aldrich) and 20 μl of 10 mg $ml^{-1}$ blasticidin (14499, Cayman) were added. Cells were incubated at 37 °C in 5% $CO_2$. The maintenance medium consisted of Glasgow's Modified Eagle Medium (G-MEM; 078-05525, Wako) supplemented with 10% Knockout Serum Replacement (KSR; 10828028-028, GIBCO), 1% FBS (GIBCO), 1% Non-essential Amino Acids (NEAA; 139-15651, Wako), 1 mM pyruvate (190-14881, Wako) and 0.1 mM 2-mercaptoethanol (2-ME; M6250, Sigma-Aldrich). The solution was filtered through a 0.2-μm filter bottle, stored at 4 °C, used within 1 month.

For organoid induction, the serum-free floating culture of embryoid body-like aggregates with quick reaggregation (SFEBq) culture method was performed as described in a previous study[31]. In this method, 3,000 cells were suspended in 100 µl of differentiation medium in each well of a 96-well plate on day 0. On day 1, Matrigel (354230, Corning) was mixed with the differentiation medium and added to each well to reach a final concentration of 2.0%. This plate was incubated at 37 °C in a 5% $CO_2$ environment. The differentiation medium consisted of G-MEM supplemented with 1.5% KSR, 1% NEAA, 1% pyruvate and 0.1% 0.1 M 2-ME. This solution was filtered through a 0.2-µm filter bottle, stored at 4 °C and used within 1 month.

For imaging, Matrigel surrounding the organoids was reduced by pipetting gently in advance. Then the organoids were transferred from the 96-well plate to a 35-mm glass-bottom dish (D11130H, Matsunami) coated with 0.1% (wt/vol) poly-L-lysine solution in $H_2O$ (P8920, Sigma-Aldrich) and 2.5 mg ml$^{-1}$ Cell-Tak (354240, Corning). The organoids were attached to the bottom by removing the medium as much as possible and incubating for 20 min at 37 °C in a 5% $CO_2$ environment. Images were captured using an LSM 800 (Zeiss) equipped with a ×25 NA 0.8 multi-immersion lens and controlled by Zen software (Zeiss, RRID: SCR_013672). Immersion was performed with water and 17.2% (vol/vol) TDE/dd$H_2O$ (refractive index, 1.366) for controls and SeeDB-Live samples, respectively. On day 9 in SFEBq culture, 145 images were taken for each organoid at different z-positions with 3-µm intervals within 432 µm. Organoids were incubated for 2 h in SeeDB-Live medium adjusted at 270 mOsm kg$^{-1}$. Small incisions were made in the organoid by randomly inserting a glass capillary five times to facilitate penetration of SeeDB-Live into the internal vesicle.

Cultures for cortical organoid induction were performed as described in the study[63]. In this method, the cortical organoid differentiation medium consisted of G-MEM supplemented with 10% KSR, 1% NEAA, 1% pyruvate and 0.1% 0.1 M 2-ME. The solution was filtered through a 0.2-µm filter bottle, stored at 4 °C and used within 1 month. On day 0, 3,000 cells were suspended in 100 µl of differentiation medium and placed in each well of a 96-well plate. The plates were incubated at 37 °C in a 5% $CO_2$ environment. On day 7, the aggregates were transferred to a 35-mm bacterial-grade dish containing DMEM/F-12 with Glutamax (10565, Invitrogen) supplemented with N2 (17502-048, Invitrogen) and incubated in a 5% $CO_2$, 40% $O_2$ environment at 37 °C. The medium was changed every 3 days. For $Ca^{2+}$ imaging, the organoids were incubated with 5 µM Calbryte 630 (20721, AAT Bioquest) for 1 h, transferred to a 35-mm glass-bottom dish and covered with cover glass (Matsunami). Images were captured using an LSM 800 (Zeiss) equipped with a ×25 NA 0.8 multi-immersion lens and controlled by Zen software (Zeiss, RRID: SCR_013672). Immersion was performed using water and 15.6% (vol/vol) TDE/dd$H_2O$ (refractive index, 1.363) for controls and SeeDB-Live samples, respectively. On day 36 in SFEBq culture, 145 images were taken for each organoid at different z-positions with 3-µm intervals within 432 µm. Organoids were incubated for 1 h in SeeDB-Live medium.

## Production of AAV
AAV-DJ-syn-jGCaMP8m-WPRE vector was generated using pGP-AAV-syn-jGCaMP8m-WPRE (Addgene, 162375), pHelper (AAVpro Helper-free system, Takara), pAAV-DJ (Cell Biolabs) and the AAVpro 293T cell line (632273, Takara) following the manufacturers' instructions. Transfection was performed with PEI Max (24765-1, PSI). AAV vectors were purified using the AAVpro Purification Kit All Serotypes (6666, Takara). AAV.PHP.S-CAG-jGCaMP8f-WPRE vector was generated using pAAV-CAG-jGCaMP8f-WPRE, pHelper, pUCmini-iCAP-PHP.S (Addgene, 103006) and the AAVpro 293T cell line as described previously[64]. Briefly, the conditioned medium containing AAV vectors was filtered with a syringe filter to remove cell debris at 6 days after transfection. The filtered medium was concentrated and formulated with D-PBS (−) using the Vivaspin 20 column pretreated with 1% BSA

in PBS. Viral titers were measured using AAVpro Titration Kit (6233, Takara) or THUNDERBIRD SYBR qPCR Mix (QPS-201, TOYOBO) with StepOnePlus system (Thermo Fisher) or QuantStudio3 real-time PCR system (Applied Biosystems).

## Imaging of primary cardiomyocytes
Primary cultures of cardiomyocytes were prepared from P0 ICR mice as previously described[65]. The pups were anesthetized on ice and decapitated. The hearts were dissected and washed in PBS (−) containing 20 mM 2,3-butanedione monoxime (B0753, Sigma). The hearts were cut into 0.50–1-mm pieces in Hanks' Balanced Salt solution (HBSS (−); 084-08345, Fujifilm) containing 0.08% Trypsin-EDTA and 20 mM 2,3-butanedione monoxime and shaking at 4 °C for 2 h. L15 medium (128-06075, Fujifilm) containing 1.5 mg ml$^{-1}$ collagenase/Dispase mix (10269638001, Roche) and 20 mM 2,3-butanedione monoxime was added. Thirty minutes after shaking at 37 °C, the suspension was filtered through a 70-µm cell strainer. The trapped heart tissues were transferred to an L15 medium containing 1.5 mg ml$^{-1}$ collagenase/Dispase mix and 20 mM 2,3-butanedione monoxime and incubated for 10 min at 37 °C. The suspension was filtered through the cell strainer again. After centrifugation at 100g for 5 min, the pellet was resuspended with DMEM (high glucose) supplemented with 1% penicillin–streptomycin and 10% FBS. The suspension was mounted on a cell culture dish for 2 h. This helped the removal of highly adhesive cells. After gentle pipetting, the suspension was collected and plated on 35-mm dishes at $1.2 \times 10^5$ cells per cm$^2$. The cells were cultured in DMEM (high glucose), without phenol red and glutamine (040-30095, Fujifilm) supplemented with 1% penicillin–streptomycin, 10% FBS and 1% glutamine at 37 °C, 5% $CO_2$. On day 1 in vitro (DIV-1), AAV.PHP.S-CAG-jGCaMP8f-WPRE was added at $2 \times 10^{10}$ genome copies (GCs) per ml. On DIV-2, the culture medium was exchanged. The spontaneous activity of cardiomyocyte aggregates was measured with a Leica TCS SP8 equipped with a ×20 NA 0.75 multi-immersion lens and LASX software (Leica Microsystems) at DIV-3 to DIV-5. Phase contrast images were taken with an inverted microscope (DMI600B, Leica) equipped with a ×10 NA 0.4 objective lens and controlled by LAS AF software (Leica).

## Imaging of primary hippocampal neuron culture
Primary cultures of hippocampal neurons were prepared from embryonic day 16 ICR mice. The embryos were taken from the uterus and decapitated in cold HBSS (−). The brain was extracted and put into a cold dissection medium consisting of HBSS (−) supplemented with 20 mM HEPES and 1% penicillin–streptomycin solution. The hippocampus was extracted from the brain and transferred to the dissection medium in a 15-ml tube. Papain (2 mg ml$^{-1}$; LS003119, Worthington)/HBSS (−) was activated for 5 min at 37 °C. After filtration, the hippocampi were transferred to papain/HBSS (−) and incubated for 20 min at 37 °C. A total of 1 ml of 150 mg ml$^{-1}$ DNase I (11284932001, Roche)/HBSS (−) was added to the papain/HBSS (−) containing the hippocampi. The hippocampi were incubated for 5 min at 37 °C. The hippocampi were washed twice with 2 ml of HBSS (−). The supernatant was replaced with 2 ml of Neurobasal medium (21103-049, Thermo Fisher) supplemented with 2% B27 (17504-044, Thermo Fisher), 1% GlutaMax (35050-061, Thermo Fisher) and 1% penicillin–streptomycin solution. The cells were dissociated with gentle pipetting using a Pasteur pipette (Iwaki). The cells were then plated on a 35-mm glass-bottom dish coated with poly-D-lysine (P7886, Sigma) at $1.5 \times 10^5$ cells on a 12-mm-diameter coverslip and cultured in 5% $CO_2$ at 37 °C. On DIV-2, AAV-DJ-hsyn-jGCaMP8m-WPRE was added at $7 \times 10^{10}$ GCs per ml after half of the culture medium in the dishes was transferred to a 50-ml tube. Twenty-four hours after infection, the medium in the dishes was replaced with the culture medium kept in the 50-ml tube together with the same amount of fresh medium. On DIV-7, half of the culture medium was transferred to a 50-ml tube. SeeDB-Live (refractive index, 1.363) was made from this culture medium together with the same amount of fresh medium. The culture medium in the

dishes was then replaced with SeeDB-Live. The spontaneous activity was measured with a Leica TCS SP8 equipped with a ×20 NA 0.75 multi-immersion lens and LASX software (Leica Microsystems) on DIV-8 to DIV-10. Phase contrast images were taken with an inverted microscope (DMI600B, Leica) equipped with a ×20 NA 0.7 objective lens and controlled by LAS AF software (Leica).

### Transmission imaging of acute brain slices

ICR mice (P5) were anesthetized on ice and euthanized by decapitation. The brain was immediately taken and placed in cold and $O_2$-saturated ACSF (125 mM NaCl, 3 mM KCl, 1.25 mM $NaH_2PO_4$, 2 mM $CaCl_2$, 1 mM $MgCl_2$, 25 mM $NaHCO_3$ and 25 mM glucose). The brain was mounted on a silicone rubber block (Togawa Rubber) and sliced at 300-μm thickness using a microslicer (Dosaka EM). The slices were placed on a line target (Thorlabs, R1L3S6P), enclosed in a 1-mm-thick rectangular silicone chamber, and secured with a slice anchor. Slices were recovered in $O_2$-saturated ACSF for 1 h at room temperature and then cleared with SeeDB-Live. An upright microscope (Leica, S9E) equipped with a USB camera (Swift, EC5R) was used for image acquisition.

### Confocal and two-photon imaging of acute brain slices

Thy1-GCaMP6f mice (P11–15) were used for $Ca^{2+}$ imaging of the acute olfactory bulb slices. Thy1-YFP-H mice (P17–22) were used for morphological analyses. Mice were euthanized with an overdose of pentobarbital (i.p. injection, 100–150 mg per kg body weight) and decapitated. The brain was immediately dissected and placed in cold and $O_2$-saturated ACSF (125 mM NaCl, 3 mM KCl, 1.25 mM $NaH_2PO_4$, 2 mM CaCl2, 1 mM $MgCl_2$, 25 mM $NaHCO_3$ and 25 mM glucose). The brain was mounted on a silicone rubber block (Togawa rubber) and sliced using a microslicer (Dosaka EM) at 300-μm thickness. The slices were placed on a custom-made silicone chamber for imaging using an upright microscope as previously described[35,66]. The slices were recovered under the perfusion of $O_2$-saturated ACSF for 1 h at room temperature. For clearing, the brain slices were perfused with SeeDB-Live for 1 h. To remove SeeDB-Live from tissue, ACSF was perfused for >1.5 h. We could record spontaneous activity of the olfactory bulb up to 5 h.

The custom-made silicone chamber was set under an FV1000MPE microscope (Olympus/EVIDENT) with Fluoview FV10-ASW software (Olympus/Evident, RRID: SCR_014215) and a ×25 NA 1.05 water-immersion objective lens (Olympus/Evident, XLPLN25XWMP). FV5000 (Evident) with a ×25 NA 0.85 multi-immersion objective lens (EVIDENT, LUPLAPO25XS) was used only for Supplementary Video 12 and controlled by FLUOVIEW Smart software (Evident). A perfusion chamber (Warner Instruments, JG-23W/HP, PM-1, SHD-26GH/10) was used for inverted imaging. Immersion was performed using water and 15.6% (vol/vol) TDE/ddH2O (refractive index, 1.363) for controls and SeeDB-Live samples, respectively. The correction collar was turned to the appropriate positions (refractive index -1.34 for control and -1.363 for SeeDB-Live). For one-photon confocal imaging, a 473-nm laser was used. For two-photon imaging, a femtosecond laser (InSight DeepSee, SpectraPhysics) was tuned to 920 nm. For stimulation, 100 μM NMDA (Nacalai, 22034-1) and 40 μM glycine (Sigma, G7126-100G) in ACSF or SeeDB-Live/ACSF was applied during $Ca^{2+}$ imaging. Imaging data were analyzed with ImageJ. Briefly, small drifts were corrected by the Image Stabilizer plugin for ImageJ (https://imagej.net/plugins/image-stabilizer) when necessary. ROIs were created manually. After fluorescence intensity was obtained, the data were analyzed with MATLAB software (MathWorks). The $F_0$ was calculated by temporal median filtering (ten-frame window). After the signal was filtered with temporal median filtering (three-frame window), the 'findpeaks' function was applied for peak detection.

### Two-photon shadow imaging of acute brain slices

Acute brain slices were prepared from wild-type C57BL/6N mice (P18, male and female). The slice was mounted on a custom-made silicone chamber for imaging using an upright microscope as previously described[35,66]. The slice was perfused with ACSF at room temperature for 30 min for recovery. Calcein (40 μM) was added to the ACSF. The slice was perfused with ACSF containing 40 μM calcein for 1 h. For clearing, the slice was perfused with SeeDB-Live containing 40 μM calcein for 1 h. For imaging, two-photon microscopy (MM201, Thorlabs) equipped with a 25x NA 1.05 water-immersion objective lens (Olympus/Evident, XLPLN25XWMP) and ThorImageLS software (Thorlabs) was used. A 920-nm femtosecond laser (ALCOR 920-4 Xsight, SPARK LASERS) was used. The images were averaged five times during the imaging.

### Electrophysiology

Liquid junction potential (LJP) was determined for ACSF and SeeDB-Live/ACSF as described previously (Extended Data Fig. 6a)[67]. We filled the recording electrode with internal solution and the reference electrode with 3 M KCl. The two electrodes were sequentially inserted into internal solution, ACSF and SeeDB-Live/ACSF while recording under current-clamp mode to measure potentials in each solution ($V_{IN}$, $V_{control-ACSF}$, $V_{SeeDB-Live/ACSF}$). Potential differences between internal solution and each external solution ($V_{IN} - V_{control-ACSF}$ and $V_{IN} - V_{SeeDB-Live/ACSF}$) were measured 12 times, and their average values were taken as LJPs for control ACSF and SeeDB-Live/ACSF.

C57BL/6J mice (male and female) were purchased from Japan SLC. Recordings were performed at P14–18 (L5ET neurons and fast-spiking interneurons) and P28–29 (L5ET neurons). Mice were deeply anesthetized with isoflurane. For P14–18 mice, brains were quickly removed from mice and put into ice-cold ACSF bubbled with 95% $O_2$ and 5% $CO_2$. For P28–29 mice, 15 ml of ice-cold cutting solution (210 mM sucrose, 2.5 mM KCl, 1.25 mM $NaH_2PO_4$, 8 mM $MgCl_2$, 1 mM $CaCl_2$, 25 mM $NaHCO_3$ and 25 mM glucose) bubbled with 95% $O_2$ and 5% $CO_2$ were intracardially perfused, and the brains were dissected and put into the ice-cold cutting solution[68]. Acute coronal slices (300-μm thick) containing S1 were prepared using a vibratome (VT1200S, Leica). The slices were recovered in ACSF for at least 1 h at room temperature (23–24 °C) before recording for the control condition. For the SeeDB-Live condition, following the recovery in ACSF for 1 h, the slices were transferred into SeeDB-Live solution saturated with 95% $O_2$ and 5% $CO_2$ for 1 h at room temperature. For the P15–18 L5ET neurons, all the recordings were performed within 6 h after recovery (4 h after clearing). To minimize the sampling bias, recording was first performed under the control ACSF for the half of the experiments; for the remaining half, the recording was first performed under the SeeDB-Live/ACSF. For the P14–18 fast-spiking interneurons and P28–29 L5ET neurons, all the recordings were performed within 6 h after recovery. Recording was extremely difficult when cleared with SeeDB-Live, even using infrared differential interference contrast. We tried to minimize the sampling bias by limiting the recording period after sample preparation. Consistent sampling (for example, cell type and depth) was confirmed.

The slices were perfused with ACSF or SeeDB-Live at room temperature during the recording. Neurons were visualized by an infrared differential interference contrast video microscope with a ×60, NA 1.0 water-immersion lens. Patch pipettes (3.9–9.9 MΩ) were filled with 130 mM potassium gluconate, 8 mM KCl, 1 mM $MgCl_2$, 0.6 mM EGTA, 10 mM HEPES, 3 mM $Na_2ATP$, 0.5 mM $Na_2GTP$, 10 mM $Tris_2$-phosphocreatine and 0.2% biocytin (pH was adjusted to 7.35 with KOH and osmolality was adjusted to 295 mOsm kg⁻¹). Whole-cell recording was performed in S1. L5ET neurons (thick-tufted L5 neurons with large cell bodies) and L5 fast-spiking interneurons with high-frequency spiking and little adaptation were analyzed. Neurons whose cell bodies were located deeper than 35 μm from the slice surface were recorded. Neurons were almost invisible under SeeDB-Live. Therefore, we ejected the internal solution with the lower refractive index from the pipette to better visualize target neurons. Recordings were performed using MultiClamp700B amplifiers (Molecular Devices), filtered at 10 kHz

using a Bessel filter and digitized at 20 kHz with Digidata 1440 A digitizer (Molecular Devices), and stored using pClamp10 (Molecular Devices). Membrane potentials were corrected for LJPs (13.04 mV for ACSF and 9.19 mV for SeeDB-Live experiments; Extended Data Fig. 6a). A series resistance compensation was not used for recordings. When the series resistance exceeded 35 MΩ, the data were discarded. Data were analyzed using MATLAB.

To characterize firing properties, hyperpolarizing and depolarizing square current pulses were injected under current-clamp mode (+50-pA increment, 1 s). In characterizing membrane properties, the membrane potential was clamped at −60 mV, and square pulses (−5 mV, 50 ms) were applied in voltage-clamp mode. sEPSCs were recorded at the holding potential of −60 mV. Transient negative current responses with a peak amplitude of <−10 pA were detected as sEPSCs. The morphologies of the recorded neurons were visualized by staining biocytin with streptavidin-Cy3 (1:1,000 dilution, S6402; Sigma-Aldrich) after recording. Fluorescence images were obtained using confocal microscopy (LSM900, Zeiss) equipped with a ×10 objective lens and controlled by Zen software (Zeiss, RRID: SCR_013672).

### In utero electroporation
In utero electroporation was performed as described previously[35]. To label mitral cells at embryonic day 12 (E12), 1 µg each of pCAG-GCaMP6f and pGP-pcDNA3.1 Puro-CAG-Voltron2 (Addgene, 172909) plasmids were injected into the lateral ventricle. Electric pulses (a single 10-ms poration pulse at 72 V, followed by five 50-ms driving pulses at 40 V with 950-ms intervals) were delivered along the anterior–posterior axis of the brain with forceps-type electrodes (3-mm diameter, LF650P3, BEX) and a CUY21EX electroporator (BEX).

For imaging of the soma of L2/3 neurons in S1, 1 µg of pCAG-GCaMP6f and pGP-pcDNA3.1 Puro-CAG-Voltron2-ST (Addgene, 172910) plasmids were injected into the lateral ventricle at E15. For imaging of the dendrite of L2/3 neurons in S1, 0.2 µg of pCAG-tTA2, 0.5 µg of pBS-TRE-mTQ2 and 1 µg of pGP-CAG-Voltron2 plasmids were injected. Electric pulses (a single 10-ms poration pulse at 72 V, followed by five 50-ms driving pulses at 42 V with 950-ms intervals) were delivered toward the mediolateral axis of the brain with forceps-type electrodes (5-mm diameter, LF650P5, BEX) and an electroporator (CUY21EX, BEX).

### Voltage imaging of acute brain slices
Voltage imaging with Voltron2 was performed in acute brain slices of P4–11 ICR mice (male and female) subjected to in utero electroporation at E12. Mice were anesthetized in ice and decapitated. The brain was immediately collected and placed in cold and O₂-saturated ACSF. The brain was mounted on a silicone rubber block (Togawa rubber) and sliced using a microslicer (Dosaka EM) at 300-µm thickness. The slices were incubated in O₂-saturated ACSF containing 50 nM Janelia Fluor HaloTag Ligand 549 (GA1110, Promega) at room temperature for 1 h. After placing on a custom-made silicone chamber, the slices were then washed under the perfusion of O₂-saturated ACSF for 1 h (2 h for control experiment) at room temperature. For clearing, the brain slices were perfused with SeeDB-Live for 1 h.

The chamber was set under an FV1000MPE microscope (Olympus/Evident) with Fluoview FV10-ASW software (Olympus/Evident, RRID: SCR_014215) and a ×25 NA 1.05 water-immersion objective lens (Olympus/Evident, XLPLN25XWMP). Immersion was performed with water and 15.6% (vol/vol) TDE/ddH₂O (refractive index 1.363) for controls and SeeDB-Live samples, respectively. The correction collar was turned to the appropriate position. For epifluorescence imaging, a mercury arc lamp was used. GFP (U-MNIBA3: Ex 470–495 nm, Di 505 nm, Em 510–550 nm) and RFP filter set (U-MWIG3: Ex 530–550 nm, Di 570 nm, Em > 575 nm) were used for GCaMP and Voltron2₅₄₉, respectively. Epifluorescence was imaged with a high-speed CCD camera (MiCAM02-HR or MC03-N256, BrainVision) at 0.3–7 ms per frame. For two-photon imaging of GCaMP6f, a femtosecond laser (InSight DeepSee, SpectraPhysics)

was tuned to 920 nm. Image data were analyzed with ImageJ. Small drifts were corrected by the Image Stabilizer plugin. ROIs were created manually. After fluorescence intensity was obtained, the data were analyzed with MATLAB software (MathWorks). The $F_0$ was calculated as temporal median filtering (100-frame window). For peak detection, the 'findpeaks' function was applied. The threshold was set as the mean plus three times the standard deviation. Voltage changes within a mitral cell (single-trial data) were visualized with BV Workbench (BrainVision). A two-dimensional mean filter (3 × 3 pixels), drift removal (polyfit, degree 3) and a Savitzky–Golay filter (32 points, window size of five frames) were applied. For the spike-triggered averaged video, 65 events of backpropagating action potentials were averaged based on the spike timing at the soma.

### Evaluation of BSA permeability in the brain in vivo
C57BL/6N mice (2- to 4-month-old, male and female) were used. Surgery and imaging were performed using ketamine (Daiichi-Sankyo) and xylazine (Bayer; 80 mg per kg body weight and 16 mg per kg body weight, respectively) anesthesia. During surgery, the depth of anesthesia was assessed by the toe-pinch reflexes, and supplemental doses were added when necessary. A head holder for mice (SGM-4, Narishige) was used for surgery. Body temperature was maintained with a heating pad (Akizuki, M-08908). For the imaging, a craniotomy (~5 mm in diameter) was made over S1 using a dental drill with a 0.5-mm drill tip. A silicone sealant, Kwik-Sil (KWIK-SIL, WPI), was mounted surrounding the craniotomy. The dura matter was carefully removed by a micro hook (durotomy; 10065-15, Muromachi Kikai). ACSF-HEPES (145 mM NaCl, 5 mM KCl, 2 mM CaCl₂, 1 mM MgCl₂, 10 mM HEPES, pH 7.3) was used to prepare the control and SeeDB-Live/ACSF-HEPES (refractive index 1.366). BSA-CF594 (1 mg; 20290, biotium) was dissolved in 100 µl SeeDB-Live/ACSF-HEPES. SeeDB-Live/ACSF-HEPES (50–100 µl; 1% BSA-CF594) was mounted onto the brain surface. The solution was stirred for 1 h using a gelatin sponge (Spongel, LTL Pharma). Mice were euthanized with an overdose of pentobarbital (100–150 mg per kg body weight, i.p.) and decapitated. The brain was embedded in OTC compound (4583, Sakura Finetek) and frozen with liquid nitrogen. The frozen sections were made with a cryostat (CM3050S, Leica) and immediately imaged with an inverted microscope (DMI600B, Leica) equipped with a ×5 NA 0.1 dry objective lens and controlled by LAS AF software (Leica).

### In vivo imaging of the mouse brain
Adult Thy1-YFP-H mice (1- to 6-month-old, male and female) were anesthetized with a mixture of ketamine (Daiichi-Sankyo; 80 mg kg⁻¹ body weight) and xylazine (Bayer; 16 mg kg⁻¹ body weight); 15% mannitol solution (3 ml per 100 g body weight; Sigma-Aldrich, M4125) was then administered intraperitoneally. We made a 3 × 6-mm² cranial window with a dental drill over the right cortical hemisphere. After the durotomy, bleeding was stopped with a gelatin sponge (LTL Pharma, Spongel). The brain surface was perfused with ACSF-HEPES, then switched to SeeDB-Live/ACSF-HEPES and perfused for 1 h at a flow rate of 1.5 ml min⁻¹. The fluorescence of L5ET neurons was imaged using a two-photon microscope (MM201, Thorlabs) equipped with a ×25 NA 1.05 water-immersion objective lens (Olympus/Evident, XLPLN25X-WMP) and ThorImageLS software (Thorlabs). The correction collar was turned to the appropriate positions (refractive index ~1.34 for control and ~1.363 for SeeDB-Live). A 920-nm femtosecond laser (ALCOR 920-4 Xsight, SPARK LASERS) was used.

### In vivo calcium imaging from mouse V1
Anesthetized adult C57BL6/J mice (~7-week-old, male and female) were used for in vivo calcium imaging of V1. Under isoflurane anesthesia (2%) together with calprofen (10 mg per kg body weight), atropine (0.3 mg per kg body weight) and dexamethasone (2 mg per kg body weight), a craniotomy and durotomy was performed over the V1 (2.2-mm square).

Half of the exposed cortex was covered with a piece of square coverslip (2 × 1 mm), and the other half was directly in contact with an extracellular solution for a subsequent pipette insertion and SeeDB-Live/ACSF-HEPES application. Anesthesia was continued at a lower concentration of isoflurane (0.125–0.5%), supplemented with chlorprothixene (1 mg per kg body weight, Sigma)[69]. Body temperature was maintained at 37 °C using a feedback-controlled heating pad. Then, the mice were placed under a two-photon microscope (Bermago II, Thorlabs).

For in vivo two-photon calcium imaging, we used jGCaMP8m or a synthetic calcium sensor Cal-520 AM. jGCaMP8m expression was achieved by AAV-DJ-Syn-jGCaMP8m-WPRE ($4.1 × 10^{14}$ GCs per ml, 150 nl at a depth of 350 μm, three sites 200 μm apart), and jGCaMP8m was excited at 980 nm (MaiTai DeepSee eHP, SpectraPhysics). Cal-520 was introduced by bolus loading[70]. Cal-520 AM (AAT Bioquest) was dissolved in 4 μl DMSO containing 20% pluronic-127, and was diluted with HEPES-based ACSF (145 mM NaCl, 5 mM KCl, 2 mM $CaCl_2$, 1 mM $MgCl_2$, 10 mM HEPES) to a final concentration of 570 μM. The Cal-520 AM solution also included 50 μM Alexa Fluor 594 (Sigma) to visualize dye-loading pipettes (at 1,070 nm excitation by Fidelity-2, Coherent). The pipette was inserted and advanced obliquely under the two-photon microscope, to target the depth of ~420 μm under the coverslip. The same pipettes were also used to record visual response of local field potential at multiple sites along the pipette track, to ensure that the retinotopic position of the dye-loading site was in the monocular zone of V1 (ref. [69]). Cal-520 AM solution was pressure injected (150 mbar, 2 min), and after an incubation time of 1 h, two-photon imaging experiments were performed. Two-photon excitation of Cal-520 was at 920 nm (MaiTai DeepSee eHP, SpectraPhysics). A water-immersion objective lens without a collar was used for V1 imaging.

Visual stimulation was presented at an LCD monitor (iPad 3/4 Retina Display, Adafruit, refresh rate of 60 Hz, gamma corrected), which was placed contralateral to the craniotomy side, covering an angle of 100° horizontal and 80° vertical in the visual hemifield. The monitor was moved and placed at a position that could evoke maximal full-field calcium response. The stimulation was controlled by MATLAB programs originally developed by Cortex Lab at University College London[69]. Full-screen sinusoidal drifting gratings were presented randomly at one of the eight directions (0–315°, 45° step) together with a blank condition.

After obtaining a visual response to the drifting gratings in the control condition of ACSF-HEPES (more than 15 repeats), solution under the objective lens (~1 ml) was replaced with SeeDB-Live/ACSF-HEPES. After an additional 1-h incubation, we confirmed that baseline brightness was increased for jGCaMP8m and visual response to the same set of drifting gratings was obtained in the presence of SeeDB-Live/ACSF-HEPES.

Calcium traces for each cell were computed offline using a Python version of Suite2p[62]. ROIs were drawn over cells, and fluorescence signal for each ROI was extracted. Neuropil contamination was not corrected, considering correction factor could be different in the presence and absence of SeeDB-Live. Baseline fluorescence ($F_0$) was defined as the 25% percentile of the fluorescence signals in the sliding window of 30–0 s before each specific time point.

Visual response for each cell was further analyzed using MATLAB. First, visual responsiveness was evaluated by Kruskal–Wallis test using calcium response across eight directions plus the blank conditions. Orientation selectivity was judged by Kruskal–Wallis test using the response at eight-direction conditions.

For orientation selective cells, preferred orientation, OSI and tuning width were computed as follows. Orientation response was fitted with the sum of two Gaussian curves. The preferred direction was defined as an angle that has the peak of the fitted Gaussian curve. The OSI was defined as the depth of modulation from the preferred orientation to its orthogonal orientation[71].

$$\theta_{ortho} = \theta_{pref} + \pi/2, \text{ as } (R_{pref} - R_{ortho})/(R_{pref} + R_{ortho})$$

The tuning width was measured as full width at half maximum height of the fitted curve. The maximum response was average calcium signal at an orientation closest to the preferred orientation.

## Surgery for the mouse olfactory bulb

AAV1-Syn-FLEX-Volton2-ST-WPRE (Addgene, 172907-AAV1) was injected into 2-month-old *Pcdh21*-Cre mice (male and female) for in vivo voltage imaging. Surgery was performed under ketamine (Daiichi-Sankyo) and xylazine (Bayer; 80 mg and 16 mg per kg body weight, respectively) anesthesia. A small circle (~1 mm in diameter) was made with a dental drill over the right olfactory bulb. The AAV1 vector was injected into the center of the dorsal olfactory bulb (200-μm depth), over a 12-min period, using a Nanoject III injector (Drummond Scientific Company). The total volume of the AAV solution was 120 nl. A custom-made head-post was attached with dental cement (GC Dental Products Corporation, UNIFAST II).

Before imaging, mice were subjected to surgery under ketamine and xylazine (80 mg and 16 mg per kg body weight, respectively) anesthesia. After a craniotomy and durotomy over the right olfactory bulb, a silicone sealant rim was created around the window to maintain fluid over the brain surface. Around 50–100 μl of ACSF-HEPES or SeeDB-Live/ACSF-HEPES was applied onto the brain surface for 1 h. After removing a silicone sealant rim, a circular coverslip (2-mm diameter) was mounted on the brain surface and fixed with a superglue and silicone sealant.

## In vivo calcium imaging in the mouse olfactory bulb

Anesthetized adult Thy1-GCaMP6f mice (2- to 4-month-old, male and female) were used for imaging. The fluorescence of mitral cells was imaged using an FV1000MPE microscope (Olympus/Evident) with Fluoview FV10-ASW software (Olympus/Evident, RRID: SCR_014215) and a ×25 NA 1.05 objective lens (Olympus/Evident, XLPLN25XWMP). Immersion was performed using water or 15.6% (vol/vol) TDE/ddH$_2$O (refractive index 1.363) for control or SeeDB-Live samples, respectively. The correction collar was turned to the appropriate positions (refractive index ~1.33 for control and ~1.363 for SeeDB-Live). Small drifts were corrected by the Image Stabilizer plugin. ROIs were created manually. After fluorescence intensity was obtained, the data were analyzed with MATLAB software (MathWorks). The ΔF was normalized to the mean intensity for 10 s before stimulus onset ($F_0$), and the response amplitude was defined as the averaged ΔF/$F_0$ during the first 10 s after stimulus onset.

## In vivo voltage imaging in the mouse olfactory bulb

AAV1-Syn-FLEX-Volton2-ST-WPRE (Addgene, 172907-AAV1) was injected into the olfactory bulb of *Pcdh21*-Cre mice before the imaging experiments. The surface of the olfactory bulb was immersed with ACSF-HEPES containing 50 nM Janelia Fluor HaloTag Ligand 549 to label Voltron2 for 1 h, followed by a 1-h washout in normal ACSF-HEPES. SeeDB-Live treatment was then performed for 1 h. Mice were placed on an FV1000MPE microscope (Olympus/Evident) and a ×25 NA 1.05 water-immersion objective lens (Olympus/Evident, XLPLN25XWMP). Immersion was performed using water and 15.6% (vol/vol) TDE/ddH$_2$O (refractive index, 1.363) for controls and SeeDB-Live samples, respectively. The correction collar was turned to the appropriate position. For excitation, a mercury arc lamp was used. RFP filter set (U-MWIG3: Ex = 530–550 nm, Di = 570 nm, Em > 575 nm) was used for Voltron2$_{549}$-ST, respectively. Epifluorescence was imaged with a high-speed CCD camera (MC03-N256, BrainVision) at 1–7 ms per frame.

Imaging data were analyzed using ImageJ. Small drifts were corrected using the Image Stabilizer plugin. ROIs were generated using a machine learning-based image segmentation tool (ilastik)[72]. Further analysis was performed using MATLAB software (MathWorks). ROIs smaller than five pixels and/or outside the glomeruli were excluded. $F_0$ was calculated by temporal median filtering (25 frames window). The cross-correlation matrix was made from the time courses of each ROI.

$k$-means clustering was applied with $k = 11$ (the number of glomeruli). To define subclusters, the spike synchronicity index between ROIs was calculated. Spikes were detected using the findpeaks function. Thresholds were set at mean + 1.5 times the s.d. and mean + 3 times the s.d. for the clustering and final spike detection, respectively. A synchronous event was defined when the spike in one ROI was detected within ±7 ms (21-ms window) of the spike in another ROI. The synchronicity index was calculated by dividing the number of synchrony events by half of the total number of spikes in two ROIs. Hierarchical clustering was used to define subclusters. The threshold was set at 10% of the maximum distance in the dendrogram. To make a reference for the phase analysis, the averaged $\Delta F/F_0$ was calculated from all glomeruli. The ROIs for the glomeruli were created manually.

### Olfactometry

Odor stimulation using an olfactometer was described previously[46]. The olfactometer consists of an air pump (AS ONE, 1-7482-11), activated charcoal filter (Advantec, TCC-A1-S0C0 and 1TS-B) and flowmeters (Kofloc, RK-1250). Valeraldehyde (TCI, V0001) and amyl acetate (FUJIFILM-Wako, 018-03623) were diluted at 1% (vol/vol) in 1 ml mineral oil in a 50-ml centrifuge tube. Saturated odor vapor in the centrifuge tube was delivered to a mouse nose with a Teflon tube. The tip of the Teflon tube was located 2 cm from the nose of the animals. Diluted odors were delivered for 5 s at 1 l min$^{-1}$.

### In vivo chronic imaging through a large cranial window

We made a large cranial window with a PVDC wrapping film, silicone plug and a coverslip as described previously[37]. We created a $3 \times 6$-mm$^2$ cranial window over the right cortex using a dental drill. After durotomy, bleeding was stopped with a gelatin sponge (Spongel, LTL Pharma). The brain surface was perfused with ACSF-HEPES, then switched to SeeDB-Live/ACSF-HEPES for 1 h at 1.5 ml min$^{-1}$. A commercially available PVDC wrapping film (Asahi Kasei, Asahi Wrap or Saran Wrap, ~11-μm thick) with a ~1-mm margin was applied to the cranial window and firmly attached to the skull with superglue. Agarose (1.5% wt/vol) with or without 19.3% wt/vol glycerol (refractive index, 1.363) was filled between the film and a glass coverslip (Matsunami $18 \times 18$ no.1). After imaging, the agarose was replaced with a transparent silicone elastomer (GC, Exaclear) and a glass coverslip was applied on top. The coverslip was then sealed with waterproof film. To minimize inflammation after the surgery, dexamethasone sodium phosphate (4 mg ml$^{-1}$) was administered intramuscularly into the quadriceps muscle at a dose of 2 μg per gram of body weight immediately before surgery. Additionally, carprofen (0.50 mg ml$^{-1}$) was administered via subcutaneous injection at a dosage of 5 mg per kg body weight once daily for 3 consecutive days following the surgery to manage postoperative pain and inflammation. jGCaMP8m was introduced by the combination of the injection of 300 nl AAV-DJ-CAG-FLEX-jGCaMP8m-P2A-CyRFP ($1.8 \times 10^{11}$ GCs per ml) to S1 and intravenous injection of 100 μl AAV.PHPeB-mscRE4-minBGpromoter-iCre-WPRE-hGHpA ($7.4 \times 10^{10}$ GCs per ml; Addgene, 163476)[73] to the retro-orbital venous sinus. A two-photon microscope (MM201, Thorlabs) equipped with a ×25 NA 1.05 water-immersion objective lens (Olympus/Evident, XLPLN25XWMP) and a 920-nm femtosecond laser (ALCOR 920-4 Xsight, SPARK LASERS) was used. Immersion was performed using ddH$_2$O for control and 15.6% (vol/vol) TDE/ddH$_2$O (refractive index, 1.363) for SeeDB-Live/ACSF-HEPES. The correction collar was turned to the appropriate position. The cranial window with a plastic wrap was carefully removed with a scalpel for repeated clearing experiments. Whisker stimulation was performed with air puffs produced by Picospritzer (Parker). Inflammation should be minimized during the procedure as this can potentially trigger the immune response to BSA. Image data were analyzed with MATLAB (MathWorks). Small drifts were corrected by the rigid motion-correction program (https://github.com/flatironinstitute/NoRMCorre/). ROIs were created manually. The median image was used to calculate the $F_0$.

### In vivo epifluorescence voltage imaging from the mouse S1

For voltage imaging of L2/3 neuron somata from P17–30 ICR mice (male and female) subjected to in utero electroporation at E15, surgery and imaging were performed under urethane (Sigma, U2500) anesthesia (1.9 g per kg body weight). For imaging of the dendrites from the 2-month-old ICR mice, surgery was performed under isoflurane anesthesia (2.5% for induction, 1–1.3% for surgery, flow rate of 0.3–0.6 l min$^{-1}$). A custom-made headpost was attached with a dental cement (GC Dental Products Corporation, UNIFAST II). After a craniotomy and durotomy (3–5 mm in diameter) over S1, a silicone sealant rim was created around the window to maintain fluid over the brain surface. In total, 50–100 μl of ACSF-HEPES containing 50 nM Janelia Fluor HaloTag Ligand 549 (GA1110, Promega) was applied onto the brain surface for 1 h. After removing the mounted solution, 50–100 μl of ACSF-HEPES was mounted for 1 h (2 h for control experiment). For clearing, SeeDB-Live/ACSF-HEPES was applied onto the brain surface for 1 h. After removing a silicone sealant rim, a circular coverslip (3 mm in diameter, thickness of 0.17 mm for P17–30, 0.34 mm for 2-month-old mice) was fixed with superglue.

The mice were placed on an FV1000MPE microscope (Olympus/Evident) and a 25x NA 1.05 water-immersion objective lens (Olympus/Evident, XLPLN25XWMP). Immersion was performed using water and 15.6% (vol/vol) TDE/ddH$_2$O (refractive index, 1.363) for controls and SeeDB-Live samples, respectively. The correction collar was turned to the appropriate position. For excitation, a mercury arc lamp was used. An RFP filter set (U-MWIG3: Ex 530–550 nm, Di 570 nm, Em >575 nm) was used for Voltron2$_{549}$. Epifluorescence was conducted with a high-speed CCD camera (MC03-N256, BrainVision) at 1–3 ms per frame. Image data were analyzed with MATLAB (MathWorks). Small drifts were corrected by the rigid motion-correction program (https://github.com/flatironinstitute/NoRMCorre/). ROIs were created manually. The $F_0$ was calculated by temporal median filtering (100-frame window). For video, a 3D median filter (1 pixel radius) was applied.

### In vivo two-photon voltage imaging from the mouse S1

AAV1:pAAV-EF1a-DIO-JEDI-2P-Kv-WPRE (Addgene viral prep no. 179459-AAV1) and AAV1:pENN.AAV.CamKII 0.4.Cre.SV40 (Addgene viral prep no. 105558-AAV1) were injected at a 3:1 ratio into 2-month-old C57BL/6N mice (male and female). Surgery was performed under ketamine (Daiichi-Sankyo) and xylazine (Bayer; 80 mg and 16 mg per kg body weight, respectively) anesthesia. A small circle (~1 mm in diameter) was made with a dental drill over the right S1. The AAV1 vectors were injected into the right S1 (3 mm lateral to the midline, 1 mm caudal to bregma at a depth of 300 μm), over a 12-min period, using a Nanoject III injector (Drummond Scientific Company). The total volume of the AAV solution was 200 nl. A custom-made headpost was attached with a dental cement (GC Dental Products Corporation, UNIFAST II).

Before imaging, mice were subjected to surgery under isoflurane anesthesia (2.5% for induction, 1–1.3% for surgery, flow rate of 0.3–0.6 l min$^{-1}$). After a craniotomy and durotomy over the right S1, 50–100 μl of SeeDB-Live/ACSF-HEPES was applied onto the brain surface for 1–2 h. A circular coverslip (3 mm in diameter, thickness of 0.34 for 2-month-old ICR mice) was mounted on the brain surface and fixed with a superglue.

The mice were placed under a resonant two-photon microscope (MM201, Thorlabs) equipped with a ×25 NA 1.05 water-immersion objective lens (Olympus/Evident, XLPLN25XWMP) and ThorImageLS software (Thorlabs). Immersion was performed using 15.6% (vol/vol) TDE/ddH$_2$O (refractive index, 1.363). The correction collar was turned to the appropriate positions (refractive index ~1.33 for control and ~1.363 for SeeDB-Live). A 920-nm femtosecond laser (ALCOR 920-4 Xsight, SPARK LASERS) was used. The frame rate was >113 Hz. Image data were analyzed with MATLAB (MathWorks). Small drifts were corrected by the rigid motion-correction program (https://github.com/flatironinstitute/NoRMCorre/). ROIs were created manually. The $F_0$ was calculated by temporal median filtering (15-frame window).

## Behavioral assays

Behavioral experiments were performed to evaluate the acute and chronic toxicity of SeeDB-Live treatment. C57BL/6N mice (2-month-old, male and female) were used. We made a large cranial window with a plastic film window ($3 \times 6$ mm$^2$) for the optical clearing of the right cortex as described above. At ≥3 days after surgery, mice were anesthetized with isoflurane (2% for induction, 0.7–1.0% during surgery) and injected with 15% mannitol solution (3 ml per 100 g body weight; Sigma-Aldrich, M4135). The plastic film and silicone were carefully removed. Then, SeeDB-Live/ACSF-HEPES or ACSF-HEPES was perfused onto the brain surface at a rate of 1.5 ml min$^{-1}$ for 1 h.

For acute behavioral experiments, head-fixed mice were placed on a custom-built treadmill 1 h after recovery from anesthesia. The treadmill consisted of a freely rotating roller with eight embedded magnets, which were detected by a sensor to monitor locomotion. Locomotion was recorded at 10 kHz for 10 min using PowerLab (AD Instruments). After the experiment, the film was reapplied to the cranial window and firmly secured to the skull with superglue. A transparent silicone elastomer (GC, Exaclear) and a glass coverslip (Matsunami 18 × 18 no.1) were then applied on top of the film.

For long-term behavioral experiments, the film was reapplied to the cranial window and firmly secured to the skull with superglue after the application of SeeDB-Live/ACSF-HEPES or ACSF-HEPES for 1 h. A transparent silicone elastomer (GC, Exaclear) and a glass coverslip (Matsunami, 18 × 18 mm, no. 1) were then placed on top of the wrap. A cortical ischemia model was used for control experiments. Photosensitive dye (150 µg per gram of body weight; Rose Bengal) was i.p. injected, and the right cortical surface was irradiated through the skull with a white LED (Leica, KL300LED, 50 W) for 10 min.

For locomotion measurements, mouse locomotion in an open chamber (30 × 25 cm) was recorded for 10 min using a USB camera (eMeet, SmartCam C960). The camera was placed above the chamber. Locomotion was tracked and analyzed with ezTrack[74] under uniform ambient lighting.

To assess changes in motor function, a wire hanging test was performed as described[55]. Mice were placed on a wire mesh (1-mm diameter wire, 10 × 10-mm grid) and allowed to grip the mesh. The wire mesh was then flipped, and the hanging time (the time of the fall) was measured up to 60 s. A soft bedding was placed under the wire mesh. The test was recorded with a video camera, and the hanging time was analyzed from the footage. After the test, the mice were returned to their home cages. Wire hanging requires motor cortex but does not require pretraining.

To monitor the food intake, mice were individually housed in cages with ad libitum access to food and water. A pre-weighed portion of standard chow was provided at the start of the test, and the remaining food was weighed every 24 h to determine the amount consumed using a feeder box (MF-1, SHINFACTORY). Mice were kept under a 12-h light–dark cycle.

## Immunohistochemistry

Immunohistochemistry was performed to evaluate the inflammatory response to SeeDB-Live treatment in vivo. C57BL/6N mice (2-month-old to 4-month-old, male and female) were used. In addition, a 10-month-old mouse was used for the long-term SeeDB-Live treatment experiment. In this mouse, jGCaMP8m was introduced by a 300-nl injection of AAV-DJ-CAG-FLEX-jGCaMP8m-P2A-CyRFP ($1.8 \times 10^{11}$ GCs per ml) into S1 and an intravenous retro-orbital injection of 100 µl AAV. PHPeB-mscRE4-minBGpromoter-iCre-WPRE-hGHpA ($7.4 \times 10^{10}$ GCs per ml; Addgene, 163476). We made a large cranial window with a plastic film window ($3 \times 6$ mm$^2$) on the right cortex as described above. Ten days after the surgery, mice were anesthetized with isoflurane (2% for induction, 0.7–1.0% during surgery) and injected with 15% mannitol solution (3 ml per 100 g body weight; Sigma-Aldrich, M4135). The film and silicone were carefully removed. Then, either SeeDB-Live/ACSF-HEPES or ACSF-HEPES was perfused onto the brain surface at

a rate of 1.5 ml min$^{-1}$ for 1 h. Then, the cranial window was sealed with plastic film, silicone and the coverslip, as described above until the next day. The timeline was determined based on the standard chronic imaging protocol with open skull surgery[75]. In the mouse with repeated SeeDB-Live treatments, the treatment was initiated immediately after the window surgery and was repeatedly performed over a period of 7 months (day 0, 3, 7, 80, 100, 120 and 218).

Mice were deeply anesthetized by an overdose of pentobarbital (i.p. injection; Dainippon Sumitomo Pharma). Anesthetized mice were perfused with 4% PFA in PBS. Excised brain samples were then fixed with 4% PFA in PBS at 4 °C overnight. The samples were cryoprotected with 30% sucrose at 4 °C overnight and embedded in OCT compound (Sakura).

Frozen brains were cut into 16-µm-thick coronal sections using a cryostat (Leica). Sections were blocked with 10% normal donkey serum or 1% BSA in PBS for 1 h and then incubated with primary antibodies in 10% normal donkey serum or 1% BSA in PBS at 4 °C overnight. After washing three times in PBS with 0.05% Tween20, sections were incubated with secondary antibodies for 2 h at room temperature. Rabbit anti-Iba1 (1:500 dilution; Wako, 019-19741), mouse anti-NeuN (1:500 dilution; Millipore, MAB377), rabbit anti-Sox9 (1:500 dilution; MilliporeSigma, AB5535), rat anti-CD16/32 (1:250 dilution; BD Pharmingen, 553142), mouse anti-glial fibrillary acidic protein (1:500 dilution; MilliporeSigma, G3893) and rabbit cleaved Caspase-3 (1:250 dilution; Cell Signaling Technology, 9664S) were used as primary antibodies. Alexa Fluor 488-conjugated donkey anti-mouse IgG (1:500 dilution; Thermo Fisher, A21202), Alexa Fluor 647-conjugated donkey anti-rabbit IgG (1:500 dilution; Thermo Fisher, A31573) and Alexa Fluor 647-conjugated donkey anti-rat IgG (1:500 dilution; Thermo Fisher, A48272) were used as secondary antibodies. Nuclei were stained with 0.1% DAPI. Images were acquired using a spinning disk confocal microscope (Andor, Dragonfly 200) mounted on an inverted microscope (Nikon, Eclipse Ti2) and controlled by Fusion software (Andor).

Microglia in the S1 region were analyzed. The cells were manually counted to calculate the density. Microglia morphology was manually traced, and 3D analysis (soma volume, soma roundness, branch number, maximum branch length and total branch length) was performed using Neurolucida software (MBF Bioscience). Roundness was defined as $(4 \times a / \pi) / b^2$, where $a$ is the soma area and $b$ is the maximum diameter of the soma in each optical section. Roundness values were calculated at all $z$-levels per cell, and the median was used for analysis. Cells were selected from the S1 L2/3 region in a blinded manner.

## Statistical analysis

MATLAB (2023b) was used for statistical analysis. Sample sizes were not predetermined. The number of animals is indicated in figure legends. All statistical tests were performed using two-sided tests. Welch's $t$-test was used in Fig. 3k. Wilcoxon's rank-sum test was used in Figs. 3b,c,e and 4n and Extended Data Figs. 3f,g, 6e–g,k,l,p,q and 7e. Wilcoxon's signed-rank test was used in Figs. 2d,e, 4m and 5e and Extended Data Figs. 5f,g, 7d and 8e. Wilcoxon's rank-sum test combined with Holm–Bonferroni correction was used in Fig. 1j,m and Extended Data Fig. 1h. A multiple-comparison test with Bonferroni correction was used in Fig. 3g,h and Extended Data Figs. 2c,d and 6m,n. Dunnett's multiple-comparison test was used in Fig. 1c,g,h and Extended Data Figs. 1b,i and 2a,b. A Tukey–Kramer multiple-comparison test was used in Figs. 3m,n and 4o,p and Extended Data Fig. 7g–k. Repeated-measures analysis of variance was used in Fig. 5p. In the box plots, the middle bands indicate the median; boxes indicate the first and third quartiles; and the whiskers indicate the minimum and maximum values. Data inclusion/exclusion criteria are described in figure legends. Numerical data are summarized in Supplementary Table 3. Due to space limitations of the figures and figure legends, all the statistical details (sample size, $P$ values and statistical tests used) are summarized in Supplementary Table 4.

**Materials availability**

A new plasmid generated in this study (pCAG-GCaMP6f) has been deposited to Addgene (no. 249680).

**Reporting summary**

Further information on research design is available in the Nature Portfolio Reporting Summary linked to this article.

## Data availability

Raw image data used in this study has been deposited to the SSBD:repository (https://doi.org/10.24631/ssbd.repos.2025.11.484)[76]. Detailed protocols and technical tips are described in SeeDB Resources (https://sites.google.com/site/seedbresources/). Requests for additional program codes and data generated and/or analyzed during the current study should be directed to and will be fulfilled upon reasonable request by the corresponding author.

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

## Acknowledgements

We thank H. Zeng (Allen Institute, *Ai162*), J. Sanes (Harvard University, Thy1-YFP-H), K. Svoboda (Allen Institute, Thy1-GCaMP6f) and E. Deneris (Case Western Reserve University, *ePet-Cre*) for mouse strains; A. Miyawaki (RIKEN) for cell lines (HeLa/Fucci2); M. Eiraku (Kyoto University) for embryonic stem cell lines (EB5); D. Kim & GENIE Project (Janelia Research Campus) for pGP-CMV-GCaMP6f (Addgene plasmid no. 40755); F. St-Pierre (Baylor College of Medicine) for pAAV-EF1a-DIO-JEDI-2P-Kv-WPRE (Addgene viral prep no. 179459-AAV1); J. Wilson (University of Pennsylvania) for pENN.AAV.CamKII 0.4.Cre.SV40 (Addgene viral prep no. 105558-AAV1) and AAV1-hSyn-Cre.WPRE.hGH (Addgene viral prep no. 105553-AAV1); GENIE Project (Janelia Research Campus) for pGP-AAV-syn-jGCaMP8m-WPRE, pGP-AAV-syn-jGCaMP8f-WPRE, pGP-pcDNA3.1 Puro-CAG-Voltron2 and pGP-pcDNA3.1 Puro-CAG-Voltron2-ST, pGP-AAV-syn-FLEX-Volton2-ST-WPRE (Addgene plasmid nos. 162375, 162376, 172909, 172910 and 172907 and viral prep no. 172907-AAV1); V. Gradinaru (California Institute of Technology) for pUCmini-iCAP-PHP.S (Addgene plasmid no. 103006); B Tasic (Allen Institute) for AiP1010-pAAV-mscR E4-minBGpromoter-iCre-WPRE-hGHpA (Addgene plasmid no. 163476); M -T. Ke and M. Morimoto (RIKEN) for evaluating our earlier versions of the clearing medium; S. Uchida and K. Miyamichi (RIKEN) for sharing reagents; M. Nishihara, E. Nozoe, S. Hamatake, K. Yashiro and S. -H. Chou for technical assistance. We also thank The Research Support Center, Research Center for Human Disease Modeling, Kyushu University Graduate School of Medical Sciences, which was in part supported by Mitsuaki Shiraishi Fund for Basic Medical Research. We are grateful to EVIDENT for generously providing a chance to test the FV5000 system. This work was supported by grants from CREST program (JPMJCR2021 to T.I.), FOREST Program (JPMJFR230P to S.I.) of the Japan Science and Technology Agency (JST), AMED (JP23wm0525012 to T.I., JP25wm0625128 to T.I. and S.I., JP19dm0207080 to K.K., JP19dm0207079 to S.M., JP23gm6510022 and JP24wm0625119 to M.S.), the JSPS KAKENHI (JP21H00205, JP21H05696, JP23H02577, JP23H04236, JP23K18165, JP24H02308 and JP24H02312 to T.I., JP21H02140, JP22K18373 and JP24K01702 to

S.I., JP19K06886 and JP24K02132 to S.F., JP22K06446, JP22H05094, JP24H01289 and JP25K02560 to N.N.-T., JP22H05161, JP22H00460 and JP23K18161 to K.K., JP22H02718 to S.M., JP24K18240 and JP25H02500 to T.Y., JP23K06151 to Y.K., JP20K23378 to T.K.S., JP24H00861 and JP25K21772 to M.S., JP25KJ1906 to H.T.), Kagoshima University Megumikai Medical Research Promotion Fund (to Y.T.), World Premier International Research Center Initiative (WPI-PRIMe; to K.H.), the Mochida Memorial Foundation for Medical and Pharmaceutical Research and the Uehara Memorial Foundation (to T.I.).

## Author contributions

T.I. conceived the project. S.I. and T.I. designed the experiments. S.I. performed all the experiments for screening and optimizing SeeDB-Live. N.N.-T. and Y.T. performed electrophysiology experiments. N.Z.H., Y.K., S.I., T.N., H.T., T.Y., M.S. and T.K.S. performed in vivo imaging. S.I., S.M. and K.K. performed chronic awake in vivo imaging. S.F. performed in utero electroporation. S.I., R.Y., M.M., A.T., Y.N., K.H. and S.O. performed organoid experiments. S.I. and T.N. performed behavioral assays. H.T., K.I. and S.I. performed immunohistochemistry experiments. T.I. supervised the project. S.I. and T.I. wrote the paper with input and feedback from all the authors.

## Competing interests

S.I. and T.I. have filed a patent application related to SeeDB-Live. The other authors declare no competing interests.

## Additional information

**Extended data** is available for this paper at https://doi.org/10.1038/s41592-026-03023-y.

**Correspondence and requests for materials** should be addressed to Shigenori Inagaki or Takeshi Imai.

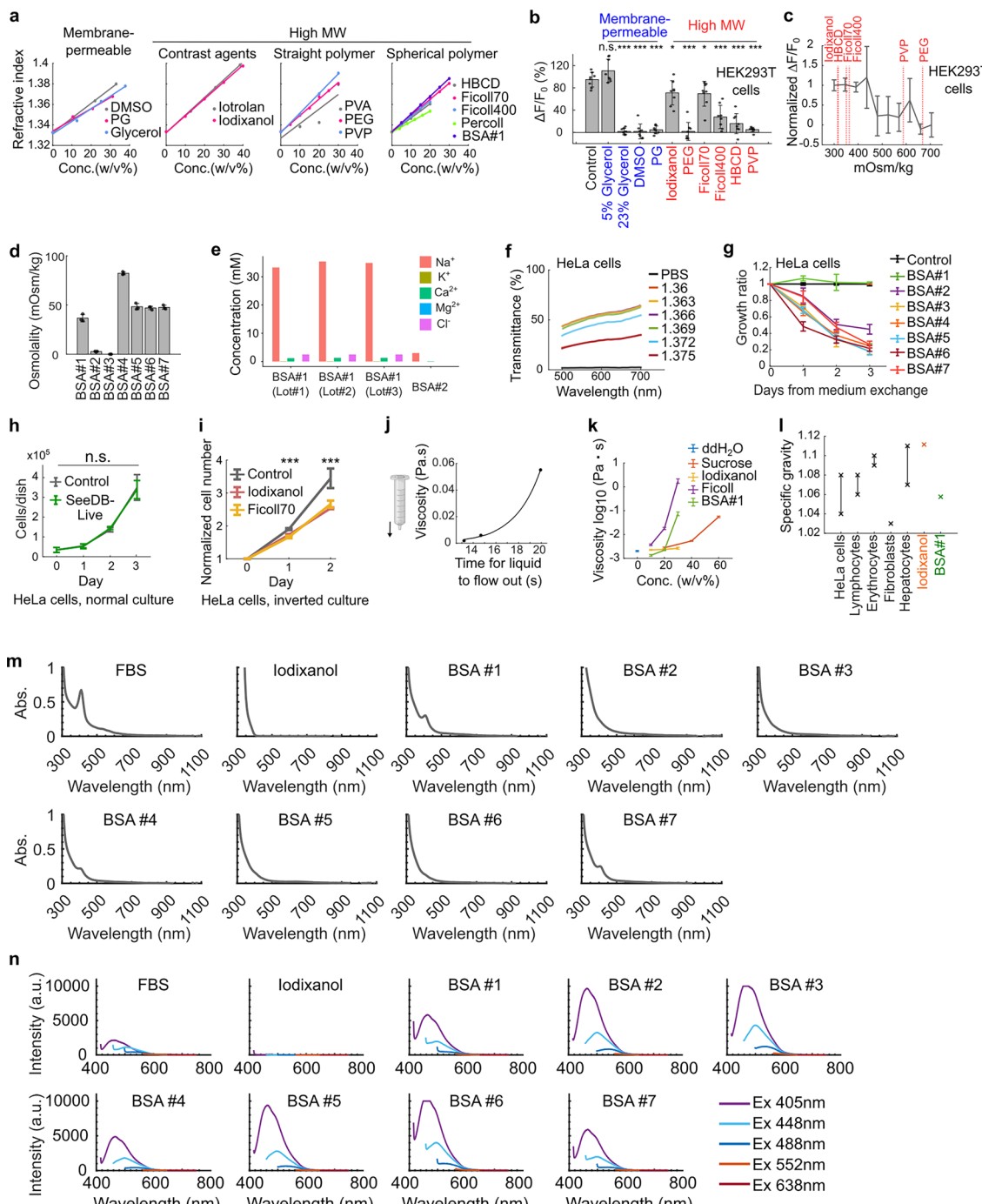

**Extended Data Fig. 1 | See next page for caption.**

**Extended Data Fig. 1 | Evaluation of candidate chemicals.** (**a**) Concentration-dependent increase in refractive indices of candidate chemicals in water. (**b**) Calcium responses of GCaMP6f-expressing HEK293T cells to 50 μM ATP, measured using a plate reader. The refractive index was adjusted to 1.365 except for 5% glycerol (refractive index 1.348). Osmolarity was not adjusted to isotonicity (hypertonic, 300-700 mOsm/kg see also **c**) in this experiment. After incubation with the clearing media for 4 hours, GCaMP6f responses were measured. n = 8 wells each. ***$p < 0.001$; **$p < 0.01$; *$p < 0.05$; n.s., not significant ($p \geq 0.05$) (two-sided Dunnett's multiple comparison test). (**c**) Calcium responses of GCaMP6f-expressing HEK293T cells to 50 μM ATP in the media with variable osmolality (300-700 mOsm/kg). The osmolality was adjusted with 10x PBS. The osmolalities of high MW media are also indicated. n = 8 wells each. (**d**) The osmolality of different BSA products in aqueous solution (refractive index 1.365, in ddH$_2$O). n = 3 each. There are several products of BSA with different grades of purity are available from different companies. The osmolality was slightly different among these products, suggesting that salts remain in some of the products. (**e**) Residual salts contained in BSA products #1 and #2. The amounts of Na$^+$, K$^+$, Ca$^{2+}$ and Mg$^{2+}$ in BSA powder were measured using Inductively Coupled Plasma Mass Spectrometry (ICP-MS). The Cl$^-$ amount was obtained from the product's certificate of analysis, which was measured by ion chromatography. Ionic concentrations of BSA solution at a refractive index of 1.365 (17% w/v) are shown. In the preparation of SeeDB-Live, residual salts contained in the BSA powder were taken into account. (**f**) Transmittance of live HeLa cell suspension (4 × 10$^6$ cells/mL, transmittance at 600 nm) in isotonic BSA#1/PBS at refractive index 1.365-1.375. n = 3 each. (**g**) Growth ratio of HeLa/Fucci2 cells in refractive index-optimized (refractive index 1.363, 320 mOsm/kg) medium of different BSA products. Cell numbers were counted based on the cell nuclei in the fluorescence

images, and the number was normalized by the control data at each stage (n = 5 wells). Purification methods are not disclosed for most BSA products. (**h**) Growth curve of HeLa cells cultured in a 35 mm dish. Cell numbers in suspension was measured with a hemocytometer after trypsinization. The medium was isotonic, and their refractive indices were adjusted to 1.363. n = 3 dishes each. n.s.; not significant ($p \geq 0.05$) (Wilcoxon rank sum test corrected with Holm-Bonferroni correction). (**i**) Growth curve of HeLa/Fucci2 cells cultured in the inverted culture in 384-well plates. Cell number was counted based on fluorescence images of the nuclei. n = 5 wells. ***$p < 0.001$ (two-sided Dunnett's multiple comparison test). (**j**) Standard curve for the viscosity measurement using sucrose solution. Flow speed was determined using 50 mL syringes. Plot was fitted with a single-exponential curve. n = 5 each. The viscosity of sucrose solution was obtained from the previous literature[60]. Created in BioRender. Imai, T. (2026) https://BioRender.com/gyynf4j. (**k**) Viscosity of candidate solutions (in ddH$_2$O). The viscosity was calculated based on the flow rate in (**j**). n = 5 each. (**l**) Specific gravity of the candidate solutions in PBS. The refractive indices of iodixanol and BSA#1 were 1.366. Specific gravity of cells are cited from previous studies[77]. (**m, n**) Some of the BSA products have autofluorescence signals of unknown origin, especially for UV to blue range; therefore, we carefully selected low autofluorescence and low toxicity products from multiple suppliers. Absorbance (**m**) and fluorescence (**n**) of fetal bovine albumin (FBS), iodixanol, and BSA from different manufacturers are shown. Except for FBS, they are adjusted to 15% (w/v) in ddH$_2$O. FBS was non-diluted. Data with error bars indicate mean ± SD. See Supplementary Table 4 for detailed statistical data. PG: propylene glycol, BSA#1: BSA crystal (bioWORLD), BSA#2: BSA Low Salt (Fujifilm), BSA#3: BSA crystal (Fujifilm), BSA#4: BSA pH 5.2 (Fujifilm), BSA#5: BSA pH 7.0 (Fujifilm), BSA#6: BSA Globulin Free (Fujifilm), BSA#7: BSA Protease Free (Fujifilm).

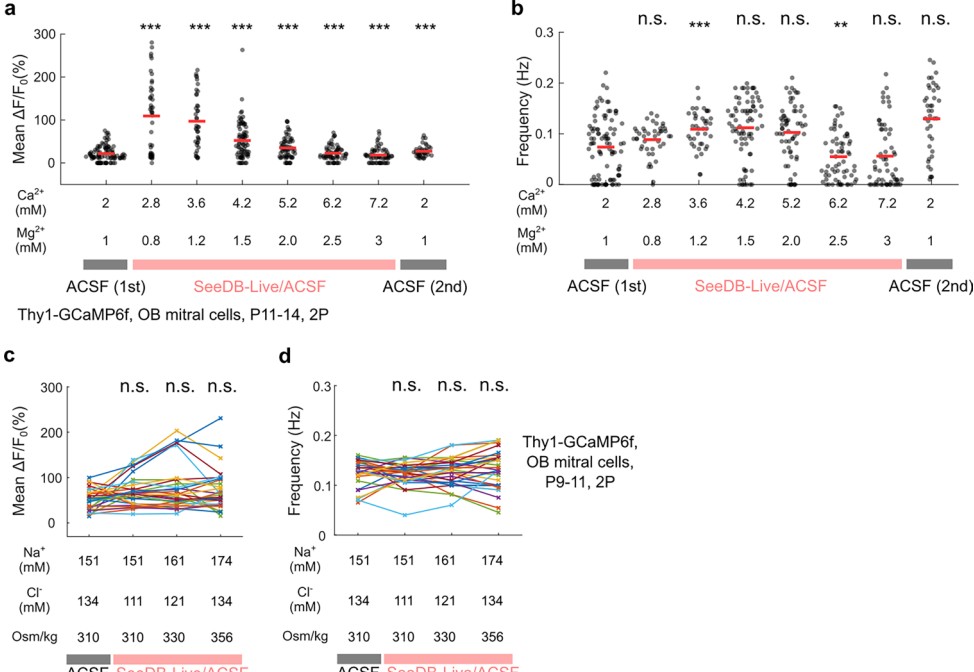

**Extended Data Fig. 2 | Optimization of ionic composition of SeeDB-Live/ACSF.**
(**a**, **b**) Spontaneous activity of mitral cells in the olfactory bulb was measured under different concentrations of divalent cations ex vivo. Amplitude (**a**) and frequency (**b**) were analyzed. Thy1-GCaMP6f mice (age, P11-14) were used for two-photon imaging of acute olfactory bulb slices. The ratio of added $Ca^{2+}$ and $Mg^{2+}$ was 2:1, following the formula of the ACSF (2 mM $Ca^{2+}$ and 1 mM $Mg^{2+}$). Note that the amounts of $Ca^{2+}$ and $Mg^{2+}$ derived from BSA products are also considered. n = 95, 43, 40, 76, 65, 64, 63 and 40 cells from 3 mice. \*\*\* $p < 0.001$; \*\* $p < 0.01$; n.s., non-significant ($p \geq 0.05$) (two-sided Dunnett's multiple comparison test). (**c**, **d**) Spontaneous activity of mitral cells in the olfactory bulb was measured under different concentrations of $Na^+$ and $Cl^-$ ex vivo. Amplitude (**c**) and frequency (**d**) were analyzed. Thy1-GCaMP6f mice (age, P9-11) were used for two-photon imaging of acute olfactory bulb slices. A NaCl stock solution was added to adjust the concentration of $Na^+$ and $Cl^-$. Note that the amounts

of $Na^+$ and $Cl^-$ derived from BSA products are also considered. There are no statistical differences in the amplitude or frequency across all conditions. However, abnormal firing (putative cortical spreading depression) was found at higher $Na^+$ conditions (174 mM) in one trial, where the $Cl^-$ concentration was matched to ACSF (134 mM). This slice was excluded in the analysis. Therefore, the isotonic condition (that is, lower $Cl^-$ condition) may be preferable. n = 31 cells from 3 mice. n.s., non-significant ($p \geq 0.05$) (Bonferroni-corrected pairwise comparisons).The formulation of the ACSF and the isotonic SeeDB-Live/ACSF (in mM) is as follows. ACSF: 151.3 $Na^+$, 134.0 $Cl^-$, 3.0 $K^+$, 1.3 $H_2PO_4^-$, 2.0 $Ca^{2+}$, 1.0 $Mg^{2+}$, 25.0 $HCO_3^-$, 25.0 glucose, 310 mOsm/kg, pH 7.4, refractive index 1.338. SeeDB-Live/ACSF: 151.0 $Na^+$, 111.3 $Cl^-$, 3.0 $K^+$, 1.0 $H_2PO_4^-$, 6.1 $Ca^{2+}$, 2.9 $Mg^{2+}$, 20.1 $HCO_3^-$, 15.0 glucose, 310 mOsm/kg, pH 7.4, refractive index 1.363. Note that the amount ions derived from the BSA products are also considered. See Supplementary Table 4 for detailed statistical data.

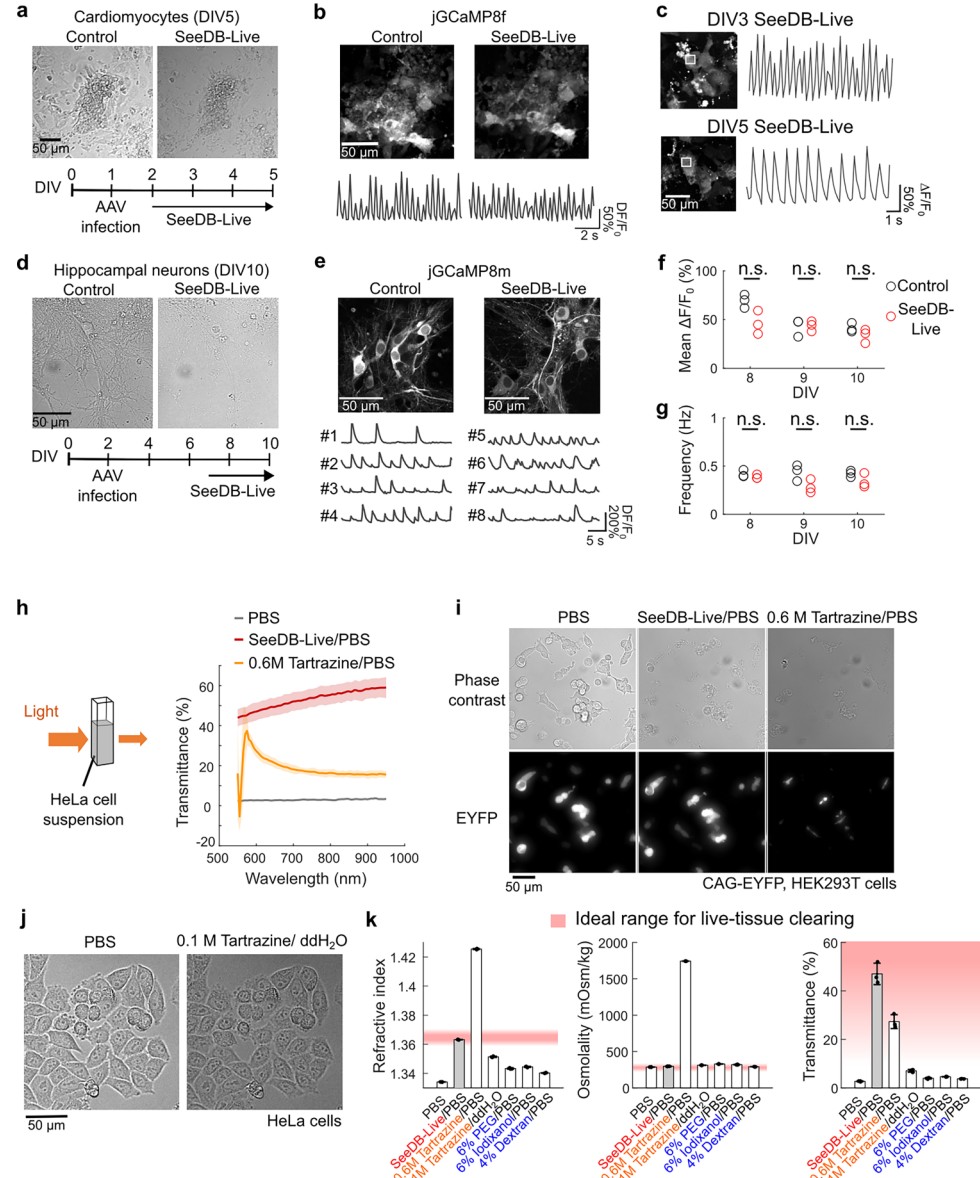

**Extended Data Fig. 3 | Long-term culture and comparison with other clearing media.** (a-g) Long-term culture in SeeDB-Live. (**a**) Primary culture of cardiomyocytes. Phase contrast image of cardiomyocytes at DIV5 (top) and the timeline of the culture (bottom). AAV.PHP.S-CAG-jGCaMP8f-WPRE ($2 \times 10^{10}$ gc/mL) was introduced at DIV1 and the medium was replaced at DIV2. Cells were cultured in SeeDB-Live medium (refractive index 1.366, 320 mOsm/kg) from DIV2 to DIV5. (**b**) Primary culture of cardiomyocytes before and after incubation in SeeDB-Live medium. Confocal image (top) and spontaneous calcium signals (bottom) at DIV2. (**c**) jGCaMP8f signals of cultured cardiomyocytes at DIV3 and DIV5. (**d**) Primary culture of hippocampal neurons. Phase contrast image (left) and the timeline of the culture (right). AAV-DJ-hSyn-jGCaMP8m-WPRE ($7 \times 10^{10}$ gc/mL) was introduced at DIV2. Neurons were cultured in SeeDB-Live medium (refractive index 1.363, 230 mOsm/kg) from DIV7 to 10. (**e**) jGCaMP8m signals of cultured hippocampal neurons before and after incubation in SeeDB-Live medium. (**f, g**) Amplitude and frequency of spontaneous calcium signals for the control (black) and SeeDB-Live (red). n = 3 dishes each. n.s., non-significant

(two-sided Wilcoxon rank sum test). (**h-k**) Comparison with other methods. (**h**) Transmittance of live HeLa cell suspension ($4 \times 10^6$ cells/mL) at different wavelengths was compared among PBS, SeeDB-Live/PBS, and 0.6 M tartrazine/PBS[18]. Due to high absorption of tartrazine, we could not compare transmission below 550 nm. (**i**) Phase contrast and fluorescence images of GFP-expressing HEK293T cells. The cells were immersed in Control (PBS), SeeDB-Live/PBS (refractive index 1.363, 296 mOsm/kg), and then in 0.6 M tartrazine/PBS (refractive index 1.43, 1407 mOsm/kg). Scale bar, 50 μm. (**j**) Phase contrast image of HeLa cells before and after immersion in 0.1 M tartrazine solution (RI 1.351), adjusted to isotonic condition (310 mOsm/kg). Scale bar, 50 μm. (**k**) Refractive indices, osmolalities, and transmittances (at 600 nm) were compared among clearing agents reported for live tissue [18,28]. Red indicates ideal ranges for non-invasive optical clearing of live cells. Data with error bars indicate mean ± SD. (**a**)-(**g**) show data from representative samples out of 3 trials. (**i**) and (**j**) show single-trial data. See Supplementary Table 4 for detailed statistical data.

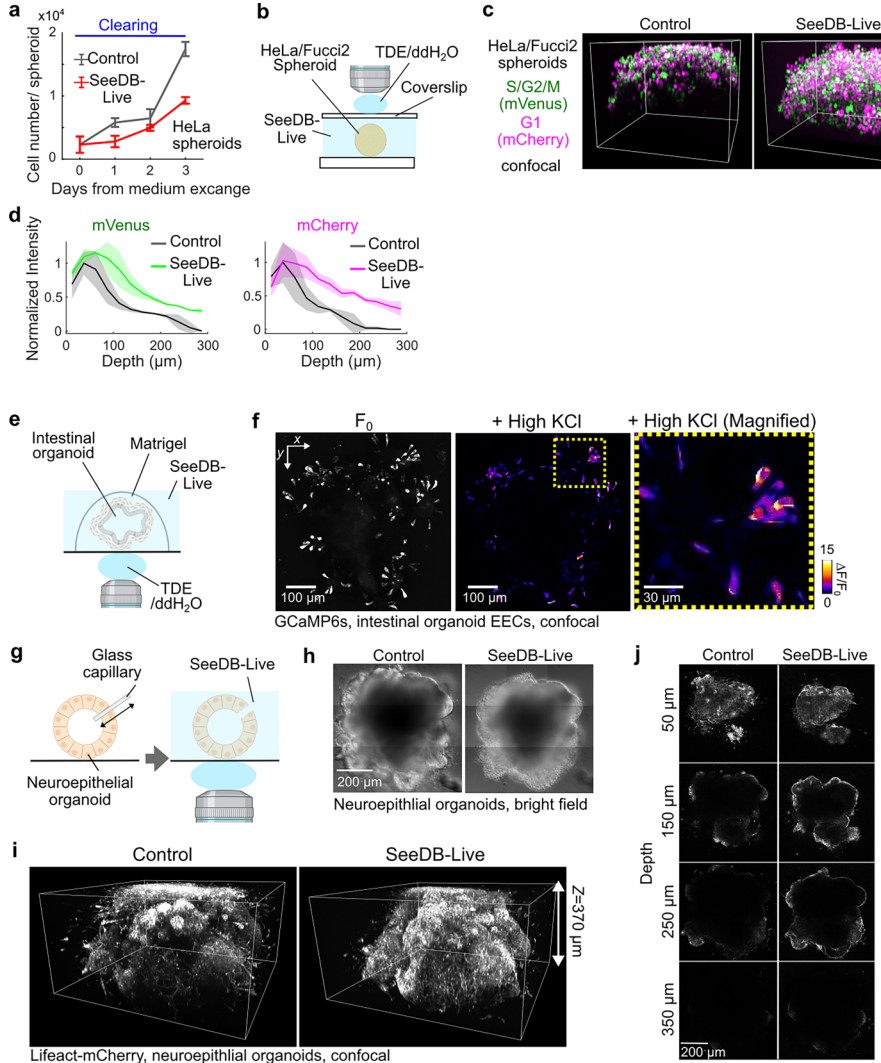

**Extended Data Fig. 4 | Clearing live spheroids and organoids with SeeDB-Live. (a-d)** HeLa/Fucci2 spheroids cleared with SeeDB-Live. **(a)** Growth of HeLa/Fucci2 spheroids cultured continuously in SeeDB-Live medium (refractive index 1.366, 320 mOsm/kg). Cell number in suspension was calculated with a hemocytometer after trypsinization. n = 3 spheroids each. Half of the medium was replaced daily. **(b)** Schematic diagram of fluorescence imaging of spheroids. 17.2% (v/v) 2,2'-thiodiethanol (TDE) in ddH$_2$O (refractive index 1.366) was used for immersion to minimize spherical aberration. The correction collar of objective lens was turned to the appropriate position. **(c)** Three-dimensional fluorescence images of a HeLa/Fucci2 cell spheroid in the control and SeeDB-Live media (4 hour clearing per day as shown in Fig. 1j). **(d)** Depth-dependent fluorescence intensity of cell nuclei in the central part of the spheroids. Fluorescence intensity indicate the mean intensity of all the cell nuclei in each z plane. n = 3 spheroids. **(e, f)** Intestinal organoid culture. **(e)** Schematic diagram of fluorescence imaging of intestinal organoids in Matrigel. 17.2% (v/v) TDE/ddH$_2$O (refractive index 1.366) was used as immersion. The correction collar was in the optimal position. **(f)** Responses of enteroendocrine cells to high potassium stimulation (30 mM at final concentrations). GCaMP6s signals are shown for the intestinal organoids derived from ePet-Cre; Ai162 mice (EEC-GCaMP6s). F$_0$ (left) and ΔF/F$_0$ (right)

images (z stack: 0-186 μm) are shown. Magnified image of the inset is shown on the right. **(g-j)** ES cell-derived neuroepithelial organoid culture. **(g)** Schematic diagram of neuroepithelial organoid sample preparations. The epithelial tissue was broken with a glass capillary to facilitate clearing of the organoid with SeeDB-Live medium. 17.2% (v/v) TDE/ddH$_2$O (refractive index 1.366) was used for immersion. The correction collar was in the optimal position. The organoids were fixed on the glass surface coated with poly-L-lysine and Cell-Tak. **(h)** ES cell-derived neuroepithelial organoids (day 9). The bright field images before and after SeeDB-Live treatment. **(i)** 3D rendered fluorescence images of Lifeact-mCherry-expressing neuroepithelial organoid before and after SeeDB-Live treatment. A representative sample out of three with similar results. Normal (left) and SeeDB-Live medium (refractive index 1.363; right). Small incision was made in the organoid before SeeDB-Live treatment. **(j)** Fluorescence images of the Lifeact-mCherry-expressing neuroepithelial organoid at different depths before and after SeeDB-Live treatment. Data with error bars indicate mean ± SD. Images show representative samples out of 2-3 trials. See Supplementary Table 4 for detailed statistical data. Panels **b**, **e** and **g** created in BioRender. Imai, T. (2026) https://BioRender.com/gyynf4j.

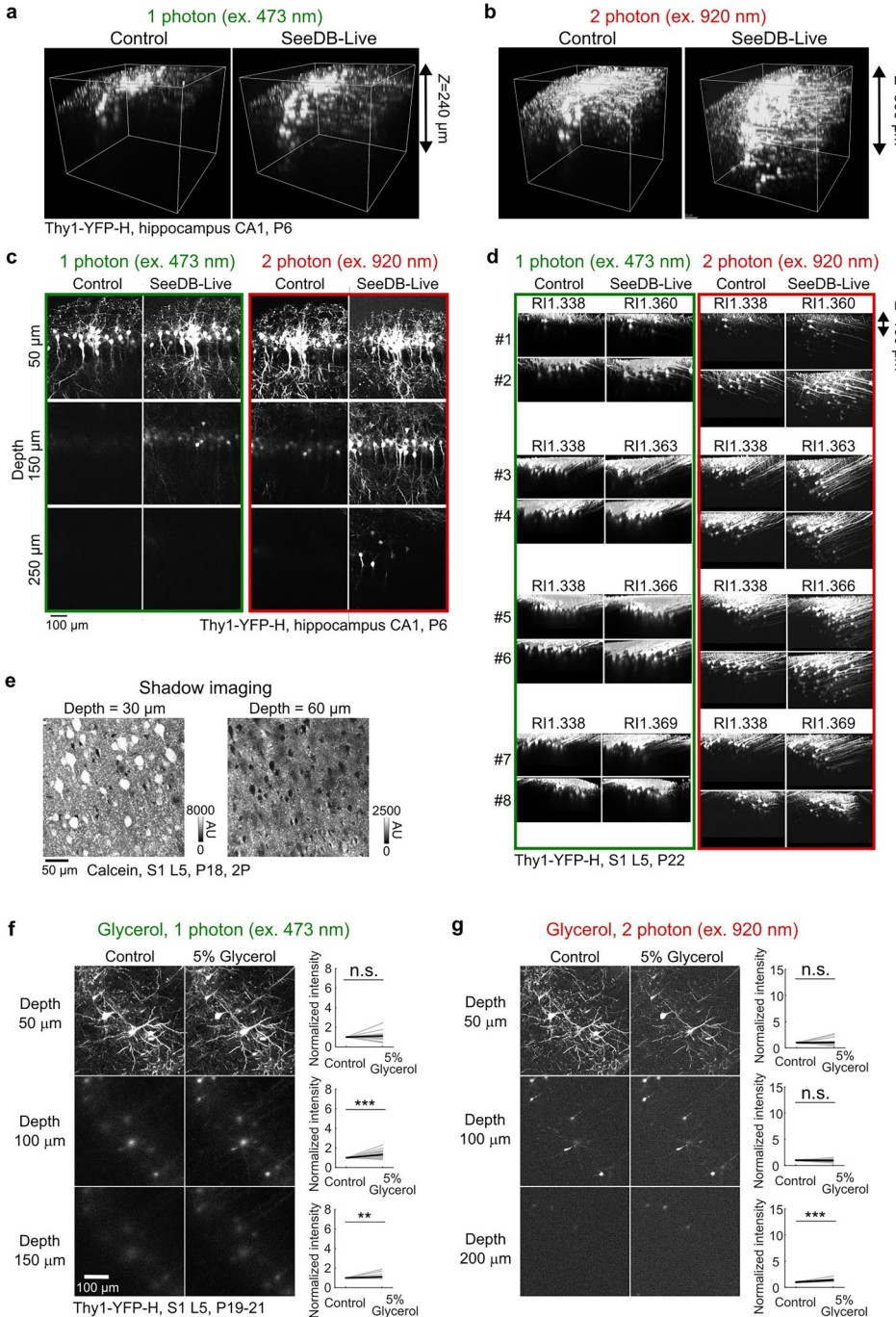

**Extended Data Fig. 5 | Clearing acute brain slices with SeeDB-Live. (a-c)** Acute hippocampal slices cleared with SeeDB-Live. Confocal (**a**) and two-photon (**b**) images (3D rendering) of acute hippocampal slices cleared with SeeDB-Live/ACSF (refractive index 1.363, 310 mOsm/kg in ACSF). Thy1-YFP-H mice (age P6) were used. (**c**) Fluorescence images of the acute hippocampal slices taken at different depths using confocal (left) and two-photon microscopy (right). (**d**) Fluorescence images of acute brain slices from the primary somatosensory (S1) cortex at different refractive indices. Thy1-YFP-H mice (age, P22) were used. SeeDB-Live/ACSF at different refractive indices (refractive index 1.338-1.369, 310 mOsm/kg) were tested. The optimal refractive index was 1.363. (**e**) Two-photon shadow images at the superficial region of an acute brain slice from S1. The slice (S1 L5) was perfused with ACSF containing 40 µM Calcein. Bright signals

are somata labeled with Calcein. The fluorescent dye was incorporated into the damaged cells at a depth of ~30 µm. Bright dots (neurites of dead neurons) were still visible at a depth of ~60 µm. (**f, g**) Fluorescence intensity from cell bodies in *x-y* fluorescence images of S1 L5ET neurons treated with 5% glycerol/ACSF. Acute brain slices of Thy1-EYFP-H mice (P19-21) were imaged before and after 5% glycerol/ACSF treatment (**f**). Mean fluorescence in ROIs are shown (**g**). For confocal imaging, n = 13, 32, and 26 cells from 3 mice for depths of 50, 100, and 150 µm, respectively. For two-photon imaging, n = 14, 33, and 15 cells from 3 mice for depths of 50, 100, and 200 µm, respectively. ***$p < 0.0001$ (two-tailed Wilcoxon signed-rank test). Images show representative samples out of 2-3 trials. See Supplementary Table 4 for detailed statistical data.

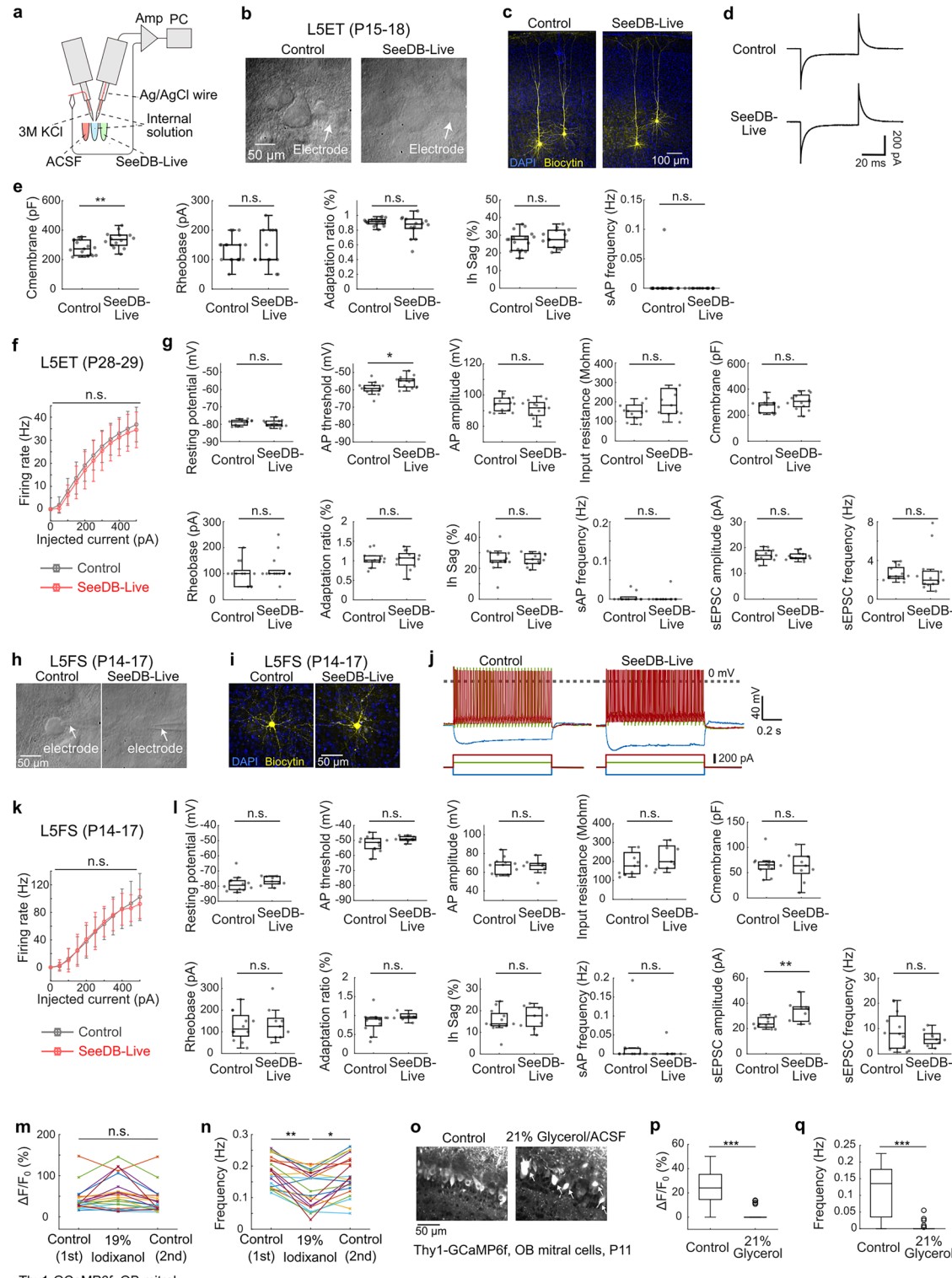

**Extended Data Fig. 6 | See next page for caption.**

**Extended Data Fig. 6 | Electrophysiological properties of neurons under SeeDB-Live.** (**a**) Measurement of liquid junction potential (LJP). We inserted recording electrode filled with internal solution and reference electrode filled with 3 M KCl into internal solution, ACSF, and SeeDB-Live/ACSF sequentially while recording the potential under current-clamp mode. LJPs between internal solution and control ACSF or SeeDB-Live/ACSF were determined by the differences in potentials. LJP was 13.04 ± 0.20 mV for control ACSF and 9.19 ± 0.09 mV for SeeDB-Live/ACSF (n = 12). (**b**) Infra-red differential interference contrast (IR-DIC) images of representative Layer 5 extratelencephalic-projecting (L5ET) neurons under electrophysiological recording. Age, P15-18. (**c**) Recorded neurons visualized by biocytin staining. They were all thick-tufted L5ET neurons. (**d**) Current responses to the test pulse (− 5 mV, 50 ms). (**e**) Additional electrophysiological properties of L5ET neurons at P15-18 recorded under ACSF and SeeDB-Live/ACSF. It should be noted that there may be biases in sampling step. For example, we could easily patch sufficient number of neurons under ACSF. However, it took much longer to patch neurons under SeeDB-Live/ACSF, because neurons are invisible and difficult to find. The box plots indicate median ± interquartile range (IQR). n = 17 neurons from 4 mice and 14 neurons from 3 mice for control and SeeDB-Live, respectively. **p < 0.01; n.s., not significant (p ≥ 0.05) (two-sided Wilcoxon rank sum test). (**f**) AP frequency was plotted against injected current amplitude for the more mature L5ET neurons (age, P28-29). (**g**) Electrophysiological properties of the more mature L5ET neurons (age, P28-29). Results were compared between ACSF and SeeDB-Live/ACSF. Recorded neurons were then confirmed by biocytin staining *post hoc*. Data are median ± IQR. *p < 0.05; n.s., non-significant (Wilcoxon

rank sum test). n = 13 cells from 2 mice per group (control and SeeDB-Live). (**h**) IR-DIC images of fast-spiking (FS) interneurons in L5 of S1. (**i**) Recorded neurons visualized by biocytin staining. They were all non-pyramidal shapes. (**j**) Changes in membrane potentials in response to square current pulses. Representative FS interneurons are shown. (**k**) AP frequency was plotted against injected current amplitude for the FS interneurons. (**l**) Electrophysiological properties of neurons for FS interneurons (age, P14-17). Results were compared between ACSF and SeeDB-Live/ACSF. FS interneurons were identified based on the firing properties ( ~ 100 Hz) upon current injection. Recorded neurons were then confirmed by biocytin staining *post hoc*. Data are median ± IQR. **p < 0.01; n.s., non-significant (two-tailed Wilcoxon rank sum test). n = 12 neurons from 4 mice and 9 neurons from 3 mice for the control and SeeDB-Live conditions, respectively. (**m, n**) Amplitude (**m**) and frequency (**n**) of spontaneous activity in the same set of mitral cells in ACSF and Iodixanol/ACSF (RI1.366). Olfactory bulb slices of Thy1-GCaMP6f mice were imaged. n = 21 cells from 3 mice. *p < 0.05; **p < 0.01; n.s., non-significant (multiple comparisons with Bonferroni correction). (**o-q**) (**o**) Spontaneous activity in the acute olfactory bulb slices was evaluated for glycerol-containing ACSF at a refractive index of 1.366. Thy1-GCaMP6f mice (age, P11) were used. Mean amplitude (**p**) and frequency (**q**) of spontaneous activity in individual mitral cells. Data are median ± IQR. n = 24, 33 cells from 3 mice for ACSF and 21% Glycerol/ACSF. ***p < 0.001 (two-tailed Wilcoxon rank sum test). Data with error bars indicate mean ± SD. Images show representative samples out of 2-4 independent trials. Box plots show median ± IQR and whiskers indicate 1.5× IQR. See Supplementary Table 4 for detailed statistical data.

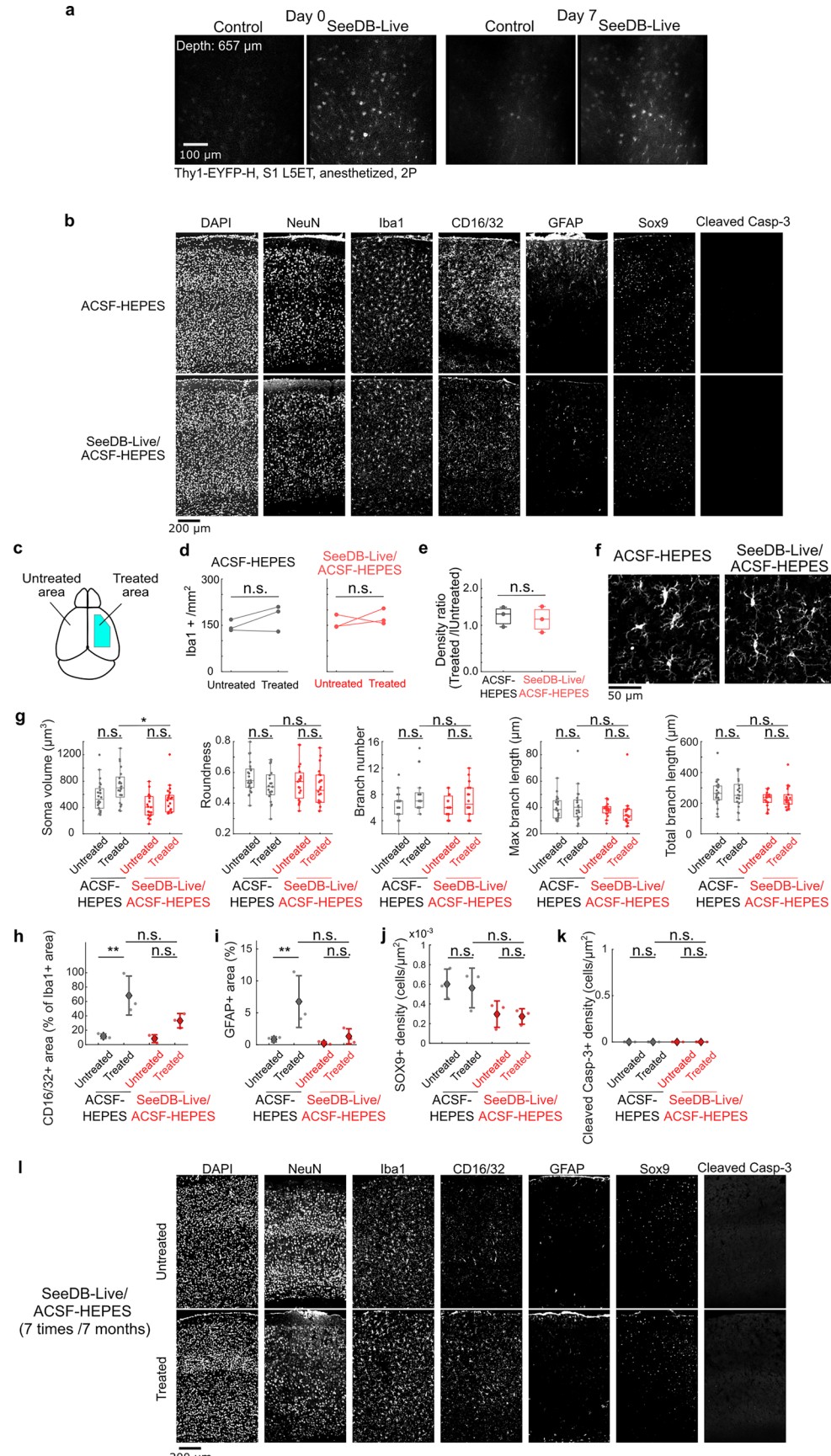

**Extended Data Fig. 7 | See next page for caption.**

**Extended Data Fig. 7 | Optical clearing in vivo and chronic imaging with SeeDB-Live. (a)** Layer 5 extratelencephalic-projecting (L5ET) neurons in the primary somatosensory cortex (S1) before and after clearing with SeeDB-Live on day 0 and 7. Preparation is illustrated in Fig. 5i. The S1 of a 4-month-old Thy1-EYFP-H mouse was imaged while the mouse was under anesthesia using two-photon microscopy. The imaging depth was 645 μm. **(b-k)** Evaluation of inflammatory responses in S1 region after SeeDB-Live treatment. A large cranial window was made 10 days prior to the SeeDB-Live treatment because open skull surgery alone is known to cause transient activation of microglia[82]. SeeDB-Live treatment was performed for 1 hour. Mice were sacrificed 1 day after the treatment. The brain sections were 16 μm thick. **(b)** Frozen sections of the cerebral cortex were stained with DAPI, anti-NeuN (neuron), anti-Iba1 (microglia), anti-CD16/32 (activated microglia, M1), anti-GFAP (activated astrocyte), anti-Sox9 (astrocyte nucleus), anti-cleaved caspase-3 (apoptosis). **(c)** Treated and untreated areas of ACSF-HEPES or SeeDB-Live/ACSF-HEPES in the brain. A large cranial window was made only over the treated area in the right cortex. **(d, e)** Density of Iba1-positive microglia in ACSF-HEPES- and SeeDB-Live/ACSF-HEPES-treated mice. Comparison of the microglial density in untreated and treated areas is shown in **(d)**, and the ratio (treated/untreated) is shown in **(e)**. n = 3 mice each for treatment with ACSF-HEPES and SeeDB-Live/ACSF-HEPES. n.s., not significant ($p \geq 0.05$) (two-sided Wilcoxon signed-rank test in **(d)**, two-sided Wilcoxon rank-sum test in **(e)**). **(f)** Morphology of microglia after treatment with

ACSF-HEPES or SeeDB-Live/ACSF-HEPES. **(g)** Evaluation of microglial morphology in ACSF-HEPES- and SeeDB-Live/ACSF-HEPES-treated mice. Microglial morphology was manually traced in 3D, and quantitative analyses (soma volume, soma roundness, branch number, maximum branch length, and total branch length) were performed. Cells were selected in the S1 L2/3 region in a blinded manner. The box plots indicate median ± interquartile range (IQR). *$p < 0.05$; n.s., not significant ($p \geq 0.05$) (two-sided Tukey-Kramer multiple comparison test). **(h–k)** Quantification of inflammatory and cytotoxic markers in untreated and treated areas of the cerebral cortex in ACSF-HEPES- and SeeDB-Live/ACSF-HEPES-treated mice. **(h)** Fraction of CD16/32-positive area within Iba1-positive regions (microglia activation). **(i)** Fraction of GFAP-positive area (astrocyte activation). **(j)** Density of SOX9-positive cells (all astrocytes). **(k)** Density of cleaved caspase-3-positive cells (apoptosis). Data indicate mean ± standard deviation. n = 3 sections from 3 mice each. **$p < 0.01$; n.s., not significant ($p \geq 0.05$) (two-sided Tukey–Kramer multiple comparison test). **(l)** Representative images of the cerebral cortex stained with DAPI, anti-NeuN, anti-Iba, anti-CD16/32, anti-GFAP, anti-SOX9, and anti-cleaved caspase-3 after repeated SeeDB-Live/ACSF-HEPES treatment (day 0, 3, 7, 80, 100, 120, and 218). Representative results from 3 sections of a single mouse are shown. Data with error bars indicate mean ± SD. Images show representative samples out of 2-4 independent trials. Box plots show median ± IQR and whiskers indicate 1.5× IQR. See Supplementary Table 4 for detailed statistical data.

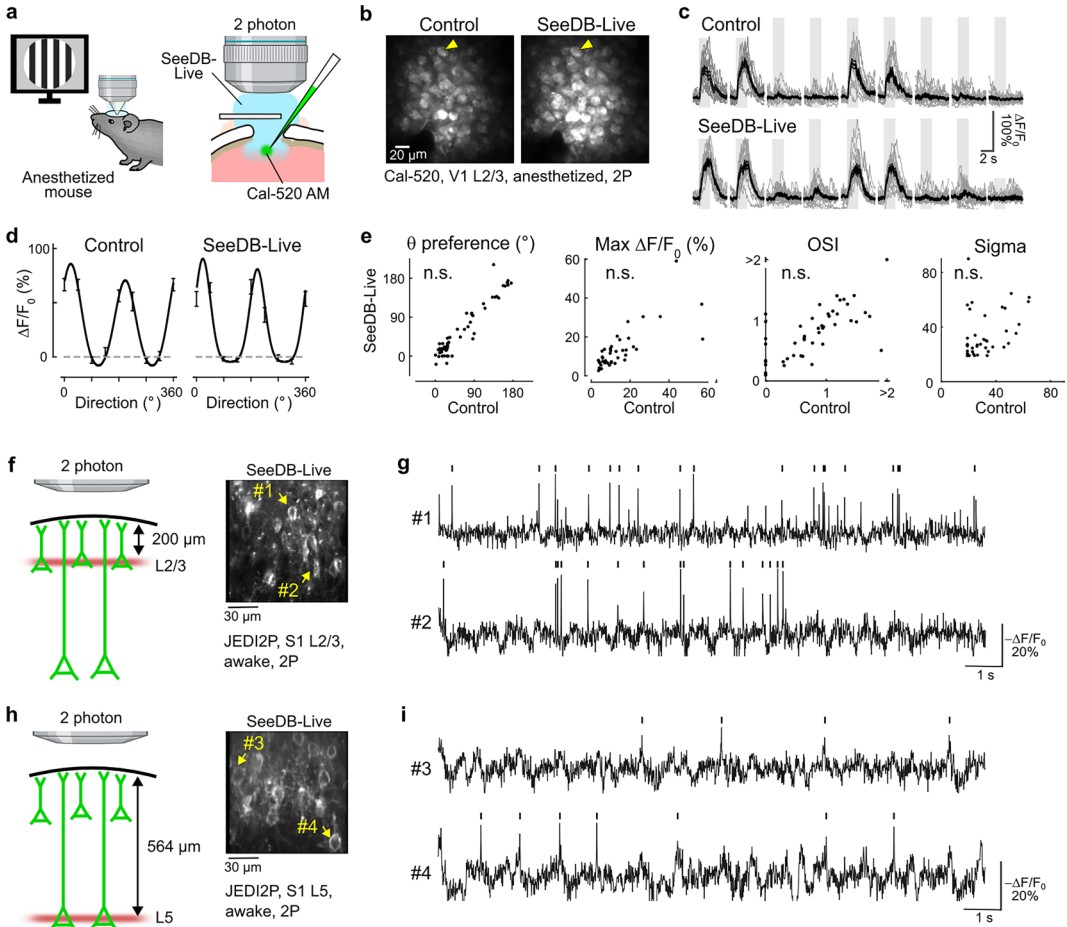

**Extended Data Fig. 8 | In vivo imaging of calcium and voltage in anesthetized and awake mice.** (**a-d**) Calcium imaging of L2/3 neurons in the primary visual cortex (V1) before and after clearing with SeeDB-Live/ACSF-HEPES. Anesthetized animals were imaged. (**a**) Experimental setup. Drifting gratings of various orientations were presented to anesthetized mice. Neurons were labeled with Cal-520-AM and imaged with two-photon microscopy. (**b**) Basal fluorescence of Cal-520 without visual stimulation. L2/3 neurons at a depth of ~420 μm. (**c, d**) Responses of a representative L2/3 neuron (indicated by arrowheads in **b**) to visual grating stimuli before and after clearing with SeeDB-Live. (**e**) Preferred orientation, maximum responses (ΔF/F$_0$), orientation selective index (OSI), and tuning width (Sigma) for the same set of L2/3 neurons (45-58 neurons in total) before (*x*-axis) and after (*y*-axis) clearing with SeeDB-Live. The comparison was performed as described previously[56]. n.s., non-significant (Wilcoxon signed-rank test). (**f-i**) Two-photon voltage imaging of JEDI-2P-labelled L2/3 (**f**) and L5 neurons (**h**) in S1 of awake mice. The fluorescence image of cell somata at 200 μm and 564 μm depth was shown on the right, respectively. The spikes were detected at the somata of L2/3 (**g**) and L5 neurons (**i**) indicated in (**f**) and (**h**), respectively. Ticks indicate detected action potentials. Data in (**c**) and (**d**) indicate mean ± SD. (**b**) shows a representative data out of 2 animals; (**f-i**) show data from single animals. See Supplementary Table 4 for detailed statistical data. Panels **a**, **f** and **h** created in BioRender. Imai, T. (2026) https://BioRender.com/gyynf4j.

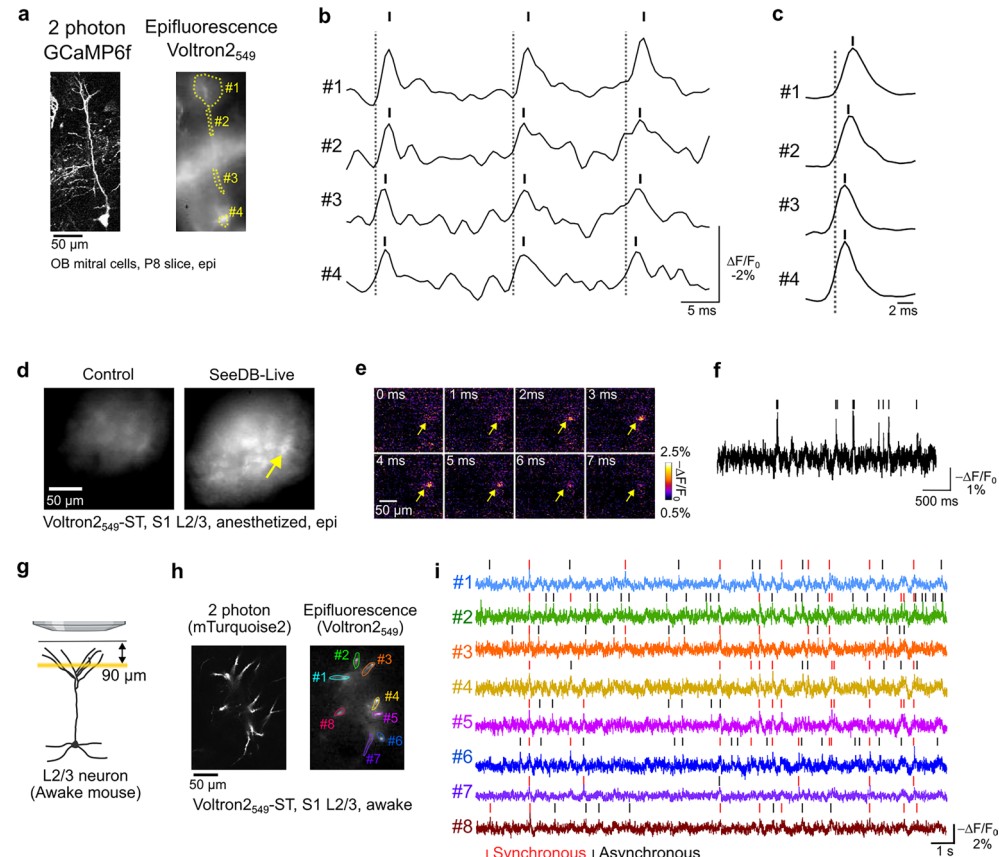

**Extended Data Fig. 9 | Epifluorescence imaging of subcellular voltage dynamics ex vivo and in vivo.** (a-c) Epifluorescence voltage imaging of the olfactory bulb slices. (a) Two-photon image identified a labeled mitral cell (z-stack; left) in acute olfactory bulb slices (P11) also shown in Fig. 6d-f. Epifluorescence of Voltron2$_{549}$ (temporal median) and ROIs are shown on the right. Cell bodies, shafts of a primary dendrite, and tufted structure within a glomerulus were analyzed. Voltron2 and GCaMP6f were introduced to mitral cells by *in utero* electroporation. (b, c) Time traces of backpropagating action potentials in each ROI indicated in (a). Traces from single shot images (b) and averaged images from 65 events (c) are shown. Ticks indicate the peaks of detected action potentials. Dotted lines indicate the half rise time of the peaks from ROI#4. (d-f) Epifluorescence voltage imaging of L2/3 neuron somata in anesthetized mice. (d) Epifluorescence images (temporal median) of L2/3 neurons in S1 labeled with Voltron2$_{549}$-ST in an anesthetized mouse (P17) before

and after clearing with SeeDB-Live (1 hour after clearing). (e, f) Action potentials were detected in a L2/3 neuron indicated by the arrow in (d). A temporal median filter with a 3-frame window was applied. Traces of the voltage changes in L2/3 neurons indicated in (f). Ticks indicate detected action potentials. (g-i) Epifluorescence voltage imaging of L2/3 neuron dendrites in awake mice. (g) Dendrites of a L2/3 neuron in S1 labeled with mTurquoise2 and Voltron2$_{549}$. We imaged an awake mouse (2 months old) after SeeDB-Live treatment. Imaging depth: 90 µm. Created in BioRender. Imai, T. (2026) https://BioRender.com/gyynf4j. (h) Two-photon fluorescence image of mTurquoise2 (left) and epifluorescence image of Voltron2$_{549}$ (right). A representative result is shown. (i) The spikes were detected at the dendrites of an L2/3 neuron in S1 of an awake mouse indicated in (h). The red ticks indicate the synchronous events that coincided among the dendrites. (a-c) is from a representative sample out of 2 slices; (d-f) is a representative sample out of 3 animals; (g-i) is from one sample.

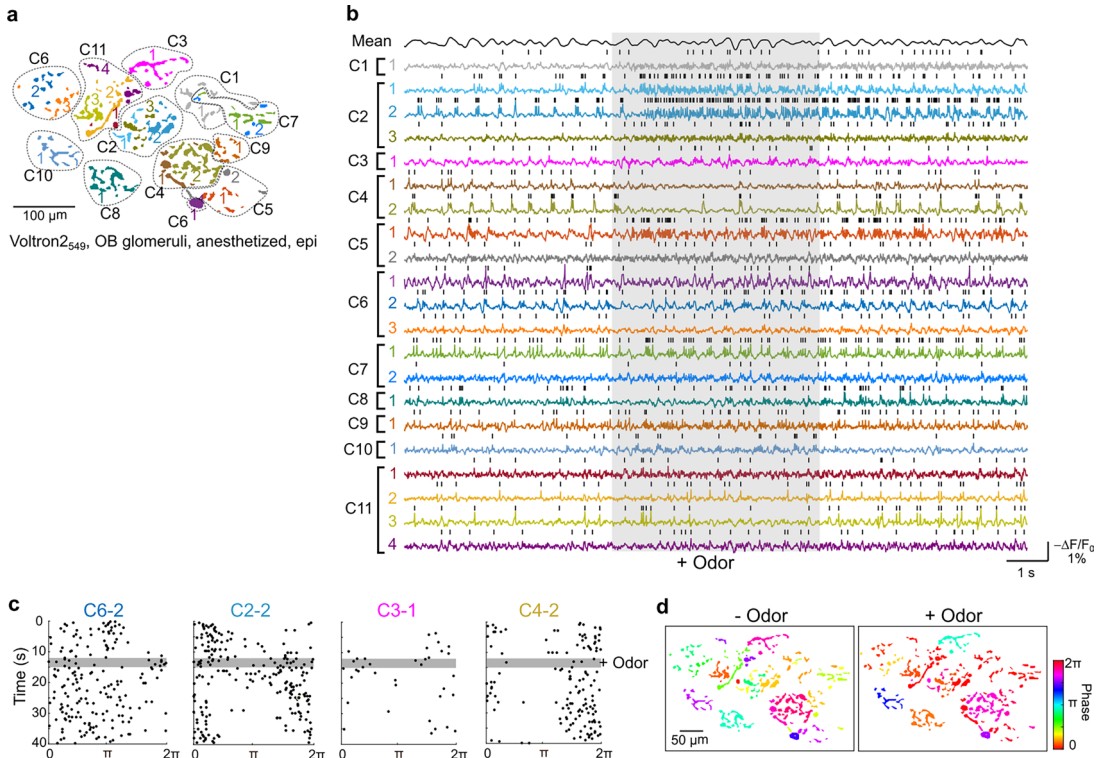

**Extended Data Fig. 10 | Epifluorescence voltage imaging of mitral/tufted cell dendrites in the olfactory bulb in vivo. (a)** Clusters (encircled by dotted lines) and subclusters (shown in different colors) of ROIs (mostly dendrites) of the mitral/tufted cells in the mouse olfactory bulb. The clusters and subclusters were defined based on voltage traces and spike synchronicity of ROIs, respectively. **(b)** Voltage traces of all subclusters ($-\Delta F/F_0$). The black trace on the top (Mean) shows the averaged $\Delta F/F_0$ of all glomeruli representing the theta wave in the olfactory bulb. Odor (1% amyl acetate) was delivered to the mice during the gray shaded period. Amyl acetate was diluted at 1% (v/v) in 1 mL mineral oil in a 50 mL

centrifuge tube. Saturated odor vapor in the centrifuge tube was delivered to a mouse nose for 5 second at 1 L/min. Ticks indicate the detected action potentials. **(c)** The spike timing in each sniff/theta cycle. The timing is shown by the phase of the theta cycle. Odor (1% amyl acetate) was delivered to the mice in the gray shaded time. The peaks of the wave (mean) were regarded as $0/2\pi$ phase. **(d)** Spike phase of each cluster against sniff-coupled theta wave before and during odor (1% amyl acetate) stimulation. The peaks of the wave were regarded as $0/2\pi$ phase. Data are from a representative sample out of two animals with similar results.

# Reporting Summary

## Statistics

For all statistical analyses, confirm that the following items are present in the figure legend, table legend, main text, or Methods section.

| n/a | Confirmed | |
|---|---|---|
| ☐ | ☒ | The exact sample size (*n*) for each experimental group/condition, given as a discrete number and unit of measurement |
| ☐ | ☒ | A statement on whether measurements were taken from distinct samples or whether the same sample was measured repeatedly |
| ☐ | ☒ | The statistical test(s) used AND whether they are one- or two-sided<br>*Only common tests should be described solely by name; describe more complex techniques in the Methods section.* |
| ☒ | ☐ | A description of all covariates tested |
| ☐ | ☒ | A description of any assumptions or corrections, such as tests of normality and adjustment for multiple comparisons |
| ☐ | ☒ | A full description of the statistical parameters including central tendency (e.g. means) or other basic estimates (e.g. regression coefficient) AND variation (e.g. standard deviation) or associated estimates of uncertainty (e.g. confidence intervals) |
| ☐ | ☒ | For null hypothesis testing, the test statistic (e.g. *F*, *t*, *r*) with confidence intervals, effect sizes, degrees of freedom and *P* value noted<br>*Give P values as exact values whenever suitable.* |
| ☒ | ☐ | For Bayesian analysis, information on the choice of priors and Markov chain Monte Carlo settings |
| ☒ | ☐ | For hierarchical and complex designs, identification of the appropriate level for tests and full reporting of outcomes |
| ☒ | ☐ | Estimates of effect sizes (e.g. Cohen's *d*, Pearson's *r*), indicating how they were calculated |

*Our web collection on statistics for biologists contains articles on many of the points above.*

## Software and code

Policy information about availability of computer code

| Data collection | Image data were obtained using the ThorImageLS, FV10-ASW, FLUOVIEW Smart, LAS X, LAS AF, Fusion, and Zen |
|---|---|
| Data analysis | All codes were written in MATLAB 2023b. They will be available upon request. |

For manuscripts utilizing custom algorithms or software that are central to the research but not yet described in published literature, software must be made available to editors and reviewers. We strongly encourage code deposition in a community repository (e.g. GitHub). See the Nature Portfolio guidelines for submitting code & software for further information.

## Data

Policy information about availability of data

All manuscripts must include a data availability statement. This statement should provide the following information, where applicable:
- Accession codes, unique identifiers, or web links for publicly available datasets
- A description of any restrictions on data availability
- For clinical datasets or third party data, please ensure that the statement adheres to our policy

Raw image data used in this study will be deposited to SSBD:repository (doi: 10.24631/ssbd.repos.2025.11.484).

# Research involving human participants, their data, or biological material

Policy information about studies with human participants or human data. See also policy information about sex, gender (identity/presentation), and sexual orientation and race, ethnicity and racism.

| | |
|---|---|
| Reporting on sex and gender | *Use the terms sex (biological attribute) and gender (shaped by social and cultural circumstances) carefully in order to avoid confusing both terms. Indicate if findings apply to only one sex or gender; describe whether sex and gender were considered in study design; whether sex and/or gender was determined based on self-reporting or assigned and methods used.*<br>*Provide in the source data disaggregated sex and gender data, where this information has been collected, and if consent has been obtained for sharing of individual-level data; provide overall numbers in this Reporting Summary. Please state if this information has not been collected.*<br>*Report sex- and gender-based analyses where performed, justify reasons for lack of sex- and gender-based analysis.* |
| Reporting on race, ethnicity, or other socially relevant groupings | *Please specify the socially constructed or socially relevant categorization variable(s) used in your manuscript and explain why they were used. Please note that such variables should not be used as proxies for other socially constructed/relevant variables (for example, race or ethnicity should not be used as a proxy for socioeconomic status).*<br>*Provide clear definitions of the relevant terms used, how they were provided (by the participants/respondents, the researchers, or third parties), and the method(s) used to classify people into the different categories (e.g. self-report, census or administrative data, social media data, etc.)*<br>*Please provide details about how you controlled for confounding variables in your analyses.* |
| Population characteristics | *Describe the covariate-relevant population characteristics of the human research participants (e.g. age, genotypic information, past and current diagnosis and treatment categories). If you filled out the behavioural & social sciences study design questions and have nothing to add here, write "See above."* |
| Recruitment | *Describe how participants were recruited. Outline any potential self-selection bias or other biases that may be present and how these are likely to impact results.* |
| Ethics oversight | *Identify the organization(s) that approved the study protocol.* |

Note that full information on the approval of the study protocol must also be provided in the manuscript.

# Field-specific reporting

Please select the one below that is the best fit for your research. If you are not sure, read the appropriate sections before making your selection.

☒ Life sciences    ☐ Behavioural & social sciences    ☐ Ecological, evolutionary & environmental sciences

For a reference copy of the document with all sections, see nature.com/documents/nr-reporting-summary-flat.pdf

# Life sciences study design

All studies must disclose on these points even when the disclosure is negative.

| | |
|---|---|
| Sample size | Sample size was not pre-determined. |
| Data exclusions | Data were not excluded unless the preparation failed. |
| Replication | We repeated the experiments to confirm their reproducibility. This information is included in Supplementary Table 4. |
| Randomization | We did not randomize samples. |
| Blinding | We did not any blind analysis except for Ext. Data Fig.7 d-g. |

# Reporting for specific materials, systems and methods

We require information from authors about some types of materials, experimental systems and methods used in many studies. Here, indicate whether each material, system or method listed is relevant to your study. If you are not sure if a list item applies to your research, read the appropriate section before selecting a response.

## Materials & experimental systems

| n/a | Involved in the study |
|-----|----------------------|
| ☐ | ☒ Antibodies |
| ☐ | ☒ Eukaryotic cell lines |
| ☒ | ☐ Palaeontology and archaeology |
| ☐ | ☒ Animals and other organisms |
| ☒ | ☐ Clinical data |
| ☒ | ☐ Dual use research of concern |
| ☒ | ☐ Plants |

## Methods

| n/a | Involved in the study |
|-----|----------------------|
| ☒ | ☐ ChIP-seq |
| ☒ | ☐ Flow cytometry |
| ☒ | ☐ MRI-based neuroimaging |

## Antibodies

| | |
|---|---|
| Antibodies used | Rabbit anti-Iba1 (Wako, #019-19741), mouse anti-NeuN (Millipore, MAB377), rabbit anti-Sox9 (MilliporeSigma, AB5535), rat anti-CD16/32 (BD Pharmingen, # 553142), mouse anti-Glial Fibrillary Acidic Protein (MilliporeSigma, G3893), and rabbit cleaved Caspase-3 (Cell Signaling Technology, #9664S) were used as primary antibodies. Alexa 488-conjugated donkey anti-mouse IgG (ThermoFisher, A21202), Alexa 647-conjugated donkey anti-rabbit IgG (ThermoFisher, A31573), and Alexa 647-conjugated donkey anti-rat IgG (1:500, ThermoFisher, A48272) |
| Validation | All antibodies were validated for immunohistochemistry in mouse tissue by the manufacturers or prior publications, and showed the expected cell-type-specific staining patterns in our experiments. |

## Eukaryotic cell lines

Policy information about cell lines and Sex and Gender in Research

| | |
|---|---|
| Cell line source(s) | AAVpro 293T (#632273, Takara), HeLa/Fucci2 cells (RCB2867, Riken BRC), HeLa S3 cells (JCRB9010, JCRB) |
| Authentication | The authentication of the cell line was guaranteed by the providers. |
| Mycoplasma contamination | No mycoplasma contamination was guaranteed by the providers. |
| Commonly misidentified lines (See ICLAC register) | *Name any commonly misidentified cell lines used in the study and provide a rationale for their use.* |

## Animals and other research organisms

Policy information about studies involving animals; ARRIVE guidelines recommended for reporting animal research, and Sex and Gender in Research

| | |
|---|---|
| Laboratory animals | ICR, C57BL/6N, Thy1-GCaMP6f Tg (line GP5.11; hemizygotes) (JAX #024339), Thy1-YFP-H (homozygotes) (JAX #003782), and Pcdh21-Cre (hemizygotes) (RIKEN BRC, RBRC02189). |
| Wild animals | *Provide details on animals observed in or captured in the field; report species and age where possible. Describe how animals were caught and transported and what happened to captive animals after the study (if killed, explain why and describe method; if released, say where and when) OR state that the study did not involve wild animals.* |
| Reporting on sex | No sex-based analysis was performed. We did not need to consider sex differences in this study. |
| Field-collected samples | *For laboratory work with field-collected samples, describe all relevant parameters such as housing, maintenance, temperature, photoperiod and end-of-experiment protocol OR state that the study did not involve samples collected from the field.* |
| Ethics oversight | All animal experiments were reviewed and approved by the Institutional Animal Care and Use Committees of Kyushu University, Kagoshima University, and Yamanashi University. |

Note that full information on the approval of the study protocol must also be provided in the manuscript.

# Plants

Seed stocks

*Report on the source of all seed stocks or other plant material used. If applicable, state the seed stock centre and catalogue number. If plant specimens were collected from the field, describe the collection location, date and sampling procedures.*

Novel plant genotypes

*Describe the methods by which all novel plant genotypes were produced. This includes those generated by transgenic approaches, gene editing, chemical/radiation-based mutagenesis and hybridization. For transgenic lines, describe the transformation method, the number of independent lines analyzed and the generation upon which experiments were performed. For gene-edited lines, describe the editor used, the endogenous sequence targeted for editing, the targeting guide RNA sequence (if applicable) and how the editor was applied.*

Authentication

*Describe any authentication procedures for each seed stock used or novel genotype generated. Describe any experiments used to assess the effect of a mutation and, where applicable, how potential secondary effects (e.g. second site T-DNA insertions, mosiacism, off-target gene editing) were examined.*

