## [Peer Review File · Nature Methods]

Isotonic and minimally invasive optical clearing media for live cell imaging ex vivo and in vivo

Corresponding Author: Professor Takeshi Imai

A version of this paper was originally rejected for publication by Nature Methods, however that decision was reconsidered after appeal by the authors.

Version 0:

Decision Letter:

10th Jan 2025

Dear Professor Imai,

Thank you for your patience. Your Article entitled "Isotonic and minimally invasive optical clearing media for live cell imaging ex vivo and in vivo" has now been seen by two reviewers, whose comments are attached. In the light of their advice we have decided that we cannot offer to publish your manuscript in Nature Methods.

You will see that, while they find your work of some potential interest, the reviewers raise concerns about the advance your methodological approach represents over available methods and about its broad applicability at this stage. Furthermore, they expressed concerns about the method's impact on cell physiology. We think that these criticisms are sufficiently important as to prevent publication of your work in Nature Methods.

Although we cannot publish your paper, it may be appropriate for another journal in the Nature Portfolio. If you wish to explore the journals and transfer your manuscript please use our manuscript transfer portal. You will not have to re-supply manuscript metadata and files, unless you wish to make modifications. For more information, please see our [manuscript transfer FAQ](http://www.nature.com/authors/author_resources/transfer_manuscripts.html?WT.mc_id=EMI_NPG_1511_AUTHORTRANSF&WT.ec_id=AUTHOR) page.

I am sorry that we cannot be more positive on this occasion but hope that you find the reviewers' comments helpful when preparing your paper for submission elsewhere.

Best regards,
Nina

Nina Vogt, PhD
Senior Editor
Nature Methods

Reviewer Comments:

Reviewer #1 (Remarks to the Author):

The study by Inagaki et al. introduces a new clearing method for live tissue termed SeeDB-Live. Here, light scattering was minimized by adding spherical polymers with low osmolarity to the extracellular medium, which enables minimally invasive optical clearing for fluorescence imaging of live tissue. The manuscript details the development of the SeeDB-Live clearing media, and using multiple microscopy techniques (confocal, two-photon, widefield), fluorescence was measured in multiple preparations including cell culture, organoids, acute brain slices and in vivo. The authors demonstrate how this clearing medium successfully increases the resolution of images and the depth of two-photon imaging. Overall, the study suggests a

step forward in live tissue clearing, however, I have some concerns regarding its use and benefit to fluorescence imaging.

Major:

1. At times, it is questionable how effective SeeDB-Live is at improving fluorescence imaging. The authors state various improvements in the different preparations, from 'up-to' two-fold (Figure 2) and four-fold (Figure 4). Although the images of the brain slices are impressive in Figure 2b and c, the measurements in Figure 2d are less conclusive as the values for control are not presented for similar depths. Please include data from the same imaging depths to allow for direct comparison. Was there a statistical difference in the normalized intensity with depth using two-photon?
Throughout the manuscript, important controls and comparisons are often lacking. For example, it is difficult to assess the improvement to morphology (Figure 2i) and voltage signals (Figure 5d-f and Figure 6) without comparable controls (fluorescent images do not necessarily equate to improved signal-to-noise responses). No controls were provided for the cortical organoid signals (Figure 1q and r), which makes it impossible to assess the benefit of using SeeDB-Live in this preparation.
2. SeeDB-Live appears to affect cellular properties, with significant changes in most cellular biophysiological properties. The authors state that the firing properties are 'largely' preserved, however, most other measurements (aside from the FI curve) is significantly different. This is to be expected due to the process but brings to question the physiological results obtained from cells within SeeDB-Live. Please alter the terminology to better reflect the influence of the media on all biophysiological properties to avoid misleading information. Also, please discuss the possible physiological implication of these changes on neuronal networks and neuronal transmission in the manuscript. Were there any changes in their electrophysiological properties of other cell types, such as interneurons?
3. SeeDB-Live's effect is transient and only lasts for ~1h in vivo and although the clearing reagent can be applied through removable cranial windows, this is still a significant limitation for chronic measurements. The examples of chronic imaging given in Figure 4 are of different neural populations and different imaging depths. Were the repeated applications of SeeDB-Live similarly effective for the same field of view to allow chronic imaging of neural morphological and activity changes over time? Furthermore, there does not appear to be an advantage of using SeeDB-Live in the in vivo V1 preparation as there was little difference compared to control. Although the data illustrates that neurons may be acting physiologically when bathed in SeeDB-Live in vivo, it does question what purpose it provides for functional imaging (in V1).
4. BSA is stated to infiltrate brain tissue to a depth of 800um. Was the infiltration of SeeDB-Live at different depths quantitatively checked to confirm clearing quality throughout the tissue? How was the imaging changed over the imaging session? Having a time-lapse measurement of the signal throughout one imaging session would be informative.
5. The concentration of Ca²⁺ and Mg²⁺ in SeeDB-Live is optimised. Potential artifacts may still arise for long-term in vivo imaging when this media has to be applied and washed multiple times. Did the functional signal-to-noise ratio change because of the application of the media? Did the media affect the firing properties of neurons in the long term in vivo? Did SeeDB-Live influence cell size over time, such as swelling or shrinking?
6. The use of SeeDB-Live was not tested against existing live clearing methods. Quantitative comparisons, such as signal-to-noise ratio and resolution improvement, would help demonstrate the strength of using this new media. Although direct experimental comparisons would be preferable, a detailed comparison in the discussion is required as a minimum.
7. Overall, the text and Figure captions require more information. Often information included in the Figures are not yet included in the text – for example, SeeDB-Live is illustrated in Figure 1d but not yet described in the corresponding text. Likewise, in Figure 1b, what are straight polymer, spherical polymer (PVA, Ficoll etc). Figure 1p – please expand to best represent and visualize the data. The yellow box in Figure 2g SeeDB-Live appears to be incorrectly placed. It is unclear in Figure 4 which data used jRCaMP8m and which used Cal-520. What does the arrow indicate in Figure 5f?
8. Why were young brain slices used to assess the electrophysiological properties of neurons cleared with SeeDB-Live? P14-18 is pre-weaning and considered immature – and developmentally younger than the in vivo measurements using Thy1 mice (P22 and 6 month old).

Reviewer #2 (Remarks to the Author):

The study of Shigenori Inagaki and colleagues proposes a clearing procedure containing bovine serum albumin (BSA) named SeeDB-Live, with refractive index of 1.363-1.366, osmolality 230-330 mOsm/kg, and Ca²⁺/Mg²⁺ concentrations of 2-4 mM, for structural and functional imaging of live tissues, such as spheroids, organoids, acute brain slices, and the mouse brain in vivo. The study presents interesting findings, however, there are some important aspects that the authors did not consider and need to be evaluated:

- A recent paper "Achieving optical transparency in live animals with absorbing molecules" published this summer by Science already introduced an approach based on Tartrazine to enable in vivo imaging of cleared samples. The authors didn't mention this approach nor compare their results with those already published. Tartrazine is a common food color approved by the US Food and Drug Administration that has the effect of reversibly making the skin, muscle, and connective tissues transparent in live rodents. Why should a user choose SeeDB-Live instead of tartrazine as Zihao Ou et al proposed?
- In the past years several articles presented different clearing in vivo, and the majority of them use RI between 1.4 and 1.5 (as mentioned in the review: "In-vivo and ex-vivo optical clearing methods for biological tissues"). On the contrary, the authors claim that "the optimal range of the refractive index for live cells was relatively narrow; the transparency of live cells became

lower at higher refractive indices (>1.38).". This is not in line with the findings already presented. The difference relies on the fact that impermeable chemicals can not enter the cells, therefore the approach used limited the finding. Therefore the general conclusion "These results indicate that the major source of light scattering in live cells is due to the index mismatch between the cytosol (~ 1.37) and the extracellular medium (typically 1.33-1.34), and thus index matching of the extracellular medium (~ 1.37) is the key to achieving maximum optical transparency of live mammalian tissues" is not completely true. Using a hypothetical chemical permeable to the membrane, with high RI, that matches the RI of all the cell components, the results could be better. The authors should clarify this point.

- The resting membrane potential of neurons cleared with SeeDB-Live results lower as well as the threshold membrane potential for the action potential. Can the authors explain this behavior? For how long does the behavior maintain this characteristic?
- For how long the spontaneous Calcium activity is constant with SeeDB-Live clearing? And for the functional imaging in vivo?
- Are there some toxic effects observed in the animal after the clearing? In the paper, there is a demonstration of the possibility of repeating the clearing after one day for functional recording however, to chronic imaging it is necessary to demonstrate the compatibility of the technique for a longer time. Have you monitored the animals for a long exposure (eg a month)?
- Have you noticed any alteration in the behavior of the animals during and/or after the clearing?

** For Nature Portfolio general information and news for authors, see <http://npg.nature.com/authors>.

Version 1:

Decision Letter:

28th Jan 2025

Dear Takeshi,

Thank you for your letter asking us to reconsider our decision on your Article, "Isotonic and minimally invasive optical clearing media for live cell imaging ex vivo and in vivo". After careful consideration we have decided that we are willing to consider a revised version of your manuscript. Importantly, please make sure to thoroughly address the reviewers' concerns about cell and animal health.

- * include a point-by-point response to our referees and to any editorial suggestions
- * please underline/highlight any additions to the text or areas with other significant changes to facilitate review of the revised manuscript
- * address the points listed described below to conform to our open science requirements
- * ensure it complies with our general format requirements as set out in our guide to authors at www.nature.com/naturemethods
- * resubmit all the necessary files electronically by using the link below to access your home page

Link Redacted

We hope to receive your revised paper within two months. If you cannot send it within this time, please let us know. In this event, we will still be happy to reconsider your paper at a later date so long as nothing similar has been accepted for publication at Nature Methods or published elsewhere.

OPEN SCIENCE REQUIREMENTS

REPORTING SUMMARY AND EDITORIAL POLICY CHECKLISTS

When revising your manuscript, please submit reporting summary and editorial policy checklists.

DATA AVAILABILITY

MATERIALS AVAILABILITY

SUPPLEMENTARY PROTOCOL

To help facilitate reproducibility and uptake of your method, we ask you to prepare a step-by-step Supplementary Protocol for the method described in this paper. We [encourage authors to share their step-by-step experimental protocols](https://www.nature.com/nature-research/editorial-policies/reporting-standards#protocols) on a protocol sharing platform of their choice and report the protocol DOI in the reference list. Nature Portfolio's protocols.io is a free-to-use and open resource for protocols; protocols deposited onto protocols.io are citable and can be linked from the published article. More details can found at [protocols.io](https://www.protocols.io/help/publish-articles).

ORCID

Best regards,
Nina

Nina Vogt, PhD
Senior Editor
Nature Methods

Version 2:

Decision Letter:

12th May 2025

Dear Takeshi,

Thank you for your letter detailing how you would respond to the reviewer concerns regarding your Article, "Isotonic and minimally invasive optical clearing media for live cell imaging ex vivo and in vivo". We have decided to invite you to revise your manuscript as you have outlined, before we reach a final decision on publication.

We do recommend to strengthen the electrophysiology data in addition to discussing potential limitations. We also think it would be helpful to include the data on the absence of longterm effects. Since these appear to be anecdotal data, we think it would be best to present them in the supplement rather than in any of the main figures. Finally, while we think that adding immunostaining data on the absence of inflammation would further strengthen the manuscript, we do think that such data are not absolutely essential.

Link Redacted

We hope to receive your revised paper within 2-3 months. If you cannot send it within this time, please let us know. In this event, we will still be happy to reconsider your paper at a later date so long as nothing similar has been accepted for publication at Nature Methods or published elsewhere.

OPEN SCIENCE REQUIREMENTS

REPORTING SUMMARY AND EDITORIAL POLICY CHECKLISTS

EXTENDED DATA FIGURES

When re-submitting your manuscript, please ensure that any supplementary figures and tables that are crucial to the

manuscript's conclusions are converted into Extended Data figures and tables to increase visibility of these data. Extended Data figures and tables are online-only (present in the online PDF and full-text HTML versions of the paper), peer-reviewed display items that provide essential background to the article but are not included in the main article due to space constraints. A maximum of ten Extended Data display items (figures and tables) is permitted.

DATA AVAILABILITY

MATERIALS AVAILABILITY

SUPPLEMENTARY PROTOCOL

To help facilitate reproducibility and uptake of your method, we ask you to prepare a step-by-step Supplementary Protocol for the method described in this paper. We [encourage authors to share their step-by-step experimental protocols](https://www.nature.com/nature-research/editorial-policies/reporting-standards#protocols) on a protocol sharing platform of their choice and report the protocol DOI in the reference list. Nature Portfolio's protocols.io is a free-to-use and open resource for protocols; protocols deposited onto protocols.io are citable and can be linked from the published article. More details can found at [protocols.io](https://www.protocols.io/help/publish-articles).

ORCID

Best regards,
Nina

Nina Vogt, PhD
Senior Editor
Nature Methods

Reviewers' Comments:

Reviewer #1 (Remarks to the Author):

The manuscript by Inagaki et al has improved from the initial submission. New experiments and analysis have been performed which is appreciated. I still have concerns to my previous comments, and the addition of new data/analysis has raised further questions.

A major change in the revised manuscript is the measurement of the liquid junction potential, which was found to alter the reporting of the electrophysiological recordings. I am very glad this was realized as otherwise the wrong results would be reported. However, I am still concerned about the electrophysiological results as there are some results which need greater

attention. For example, in Figure R3 (Figure 3, Extended Data Fig. 6), how can there be no effect on AP frequency (Figure 3c and R3d) even though there is a significant difference in input resistance and rheobase? Why is there a difference in EPSP frequency or amplitude in some preparations, and not others? Although the authors state that there is variability in slices, this is not an adequate explanation (especially with only 5/6 data points). Also, the influence on the electrophysiological properties differed in the different ages of the neurons tested which makes for great confusion. This data needs to be highlighted and discussed, and simply referring to the firing rate being 'largely preserved' does not correctly represent the data. Overall, it appears as though SeeDB-Live influences cellular properties, and the authors should make it extremely clear what is influenced to alert future users of any limitations.

The manuscript still lacks important details. Were the experiments illustrated in Figure 4a-j performed in the anaesthetized state? If so, please state this (the schematic in 4a is different than Extended Data Figure 8a). If not, then I can't see how you could perfuse the surface of the brain with SeeDB-Live for 1 hour and then perform imaging (without a coverslip to dampen movement). Along the same lines, please provide more information than 'we performed a durotomy' (line 312) for the data illustrated in Figure 5a-e. Please clearly state in the main text, and Figure caption that the visual recordings were performed in anaesthetized mice.

Furthermore, there is little information provided in the text for the values presented. Please always, where possible, include the absolute values and the exact p values (not just $p > 0.05$). This would be helpful to understand some of the data – for example, Figure 5n illustrates there is no significant change in the amplitude of the calcium responses, however, there really appears to be a strong trend for most cells to decrease on day 3. Please include more information about the odor presented (valeraldehyde) in the main text.

Although the revised manuscript has improved in clarity, the manuscript is still disjointed. For example, Figures 4 and 5 jump between awake and anaesthetized (presumably) recordings, and there are a mis-match of imaging locations/cell types. It is unclear which imaging was performed during the locomotion behavior and the positioning of the locomotion data in Figure 4 is potentially misleading if prior data (Figure 4a-j) are not performed during this behavior. What was the preparation for the behavioral assessment?

Please always note how long after SeeDB-Live solution was applied were the electrophysiological recordings were performed (were control recordings performed at similar timepoints?).

Extended Data Figure 7 illustrates important information which supports the statement (line 328) "Optical clearing with SeeDB-Live is transient in vivo (~ 1 hour) as BSA is gradually washed out." Please include this, at least in graphical form, in a main Figure.

Considering the similarities of the approach, more consideration must be taken of the Ou et al, Science, 2024 study in the Introduction. Understandably it has different applications, but only referring to this study as a previous immersion-based clearing agent is not adequate. Please consider moving the paragraph (line 189) to the Introduction.

The statement (line 285) is misleading "Thus, SeeDB-Live enables calcium imaging of healthy neuronal activity in acute brain slices using conventional confocal microscopy, without using expensive two-photon microscopy systems." Confocal microscopy can be used to image fluorescence in shallow cells in slices (up to 50um or so depth) – please alter the sentence.

Reviewer #2 (Remarks to the Author):

The updated version of the paper from Inagaki and colleagues addresses most of the concerns raised during the first revision and drastically improves the results.

A point that wasn't entirely covered is the tolerability of the clearing.

The authors demonstrated that SeeDB-Live can be used up to 7 days, but a longitudinal study can last weeks, even months. Therefore, it would be important to verify the tolerance of treatment for a longer period, adding some measurements after 2 - 3 and 4 weeks.

Moreover, the craniotomy was performed on the primary somatosensory cortex and not on the motor cortex; therefore, the observation that the locomotor activity on a treadmill is intact doesn't necessarily mean that there aren't any other alterations. I suggest evaluating the cytoarchitecture integrity of the brain, also by verifying the presence of inflammatory processes, with a post-mortem structural analysis of the animal's brain.

Finally, there is curiosity about the applicability of the method to other tissues. Do the authors expect that See-Live will work on other animals or types of tissue, e.g., the heart, or will the concentration of the clearing solution need to be adjusted? It would be interesting to include a discussion of this in the future perspectives.

Version 3:

Decision Letter:

Our ref: NMETH-A58498C

7th Oct 2025

Dear Takeshi,

Thank you for submitting your revised manuscript "Isotonic and minimally invasive optical clearing media for live cell imaging ex vivo and in vivo" (NMETH-A58498C). As I mentioned before, it has been seen by the original referees and their comments are below. The reviewers find that the paper has improved in revision, and therefore we'll be happy in principle to publish it in Nature Methods, pending minor revisions to satisfy the referees' final requests and to comply with our editorial and formatting guidelines. Please make sure to include explanations as necessary to address the remaining concerns.

TRANSPARENT PEER REVIEW

Nature Methods offers a transparent peer review option for new original research manuscripts. We encourage increased transparency in peer review by publishing the reviewer comments, author rebuttal letters and editorial decision letters if the authors agree. Such peer review material is made available as a supplementary peer review file. **Please state in the cover letter 'I wish to participate in transparent peer review' if you want to opt in, or 'I do not wish to participate in transparent peer review' if you don't.** Failure to state your preference will result in delays in accepting your manuscript for publication.

ORCID

Author names using non-Roman characters

Nature Portfolio journals can support presentation of author names using non-Roman characters in the HTML version of the article. If you wish to, please include author names in parentheses after the Roman-character spelling; [see example online here](https://www.nature.com/articles/s44222-024-00258-2). Currently supported scripts are: Arabic, Chinese, Cyrillic, Devanagari, Greek, Hebrew, Hangul, Japanese and Persian. You will be asked to verify the rendering is correct at proof stage.

Best regards,
Nina

Nina Vogt, PhD
Senior Editor
Nature Methods

Reviewer #1 (Remarks to the Author):

The authors have adequately responded to my questions with appropriate and informative experiments and I have no further concerns.

Reviewer #2 (Remarks to the Author):

The authors present a comprehensive set of experiments demonstrating that SeeDB-Live enables high-resolution functional imaging across a range of live tissues and organoids. The manuscript is well-structured, clearly written, and the results are compelling, largely supporting the authors' claims. However, several revisions and additional data would strengthen the conclusions, particularly regarding claims of non-toxicity and long-term compatibility.

1. Although the authors note that BSA "is structurally stable and minimally induces immune reactions," it remains a large, xenogenic protein. Inflammation was assessed only 24 hours after a single, 1-hour application of SeeDB-Live in thin sections.

However, many key applications, including the chronic imaging experiments (Fig. 5i–p), involve repeated use. To exclude long-term immune responses, it would be valuable to examine repeated applications over the four-month imaging period in which no changes in soma size or sensory responses were observed. Because Iba1 labels both resting and activated microglia, additional co-labeling with additional markers (e.g. CD68, Cb11, HLA-DR, GFAP, TUNEL, and activated Caspase-3) would strengthen the conclusion of “undetected toxicity.” Furthermore, performing immunohistochemical and morphological analyses on thicker sections (50–100 µm) rather than 16 µm sections would enable 3D reconstruction of the cytoarchitecture and provide more accurate quantification.

2. The authors show that intermittent 4-hour clearing with SeeDB-Live does not affect intestinal organoid growth over a few days. To validate the use of SeeDB-Live in long-term organoid studies (which often run for weeks), it would be ideal to also assess viability and apoptosis markers (e.g., live/dead staining, cleaved caspase-3) after repeated exposures. Longitudinal functional assays, such as repeated Ca²⁺ imaging over several days, would also help rule out cumulative toxicity.

3. In Figure 5n–p, please clarify whether the data in panel p comes from a single animal or is representative of a cohort. Demonstrating consistent results from multiple animals (n = 2–3) undergoing the same repeated clearing and imaging protocol would make this already strong conclusion unassailable.

4. The vivo clearing is shown to be transient, lasting approximately one hour before washing out. While this reversibility is presented as a positive feature, it also imposes a practical limitation by constraining the duration of imaging sessions. Including a brief sentence in the Discussion acknowledging this trade-off would be helpful for readers planning experiments.

5. The authors describe the lower optimal refractive index (RI) for live cells (~1.37) compared to fixed cells (~1.42) as “paradoxical.” The Discussion would be strengthened by elaborating on the biophysical basis for this difference. Specifically, the manuscript could clarify that for live cells, the goal is matching the extracellular medium to the aqueous cytosol, whereas for fixed/permeabilized cells, the higher RI is needed to match dehydrated intracellular biomolecules.

Version 4:

Decision Letter:

3rd Feb 2026

Dear Takeshi,

I am pleased to inform you that your Article, "Isotonic and minimally invasive optical clearing media for live cell imaging ex vivo and in vivo", has now been accepted for publication in Nature Methods. The received and accepted dates will be November 1st, 2024 and February 3rd, 2026. This note is intended to let you know what to expect from us over the next month or so, and to let you know where to address any further questions.

Over the next few weeks, your paper will be copyedited to ensure that it conforms to Nature Methods style. Once your paper is typeset, you will receive an email with a link to choose the appropriate publishing options for your paper and our Author Services team will be in touch regarding any additional information that may be required. It is extremely important that you let us know now whether you will be difficult to contact over the next month. If this is the case, we ask that you send us the contact information (email, phone and fax) of someone who will be able to check the proofs and deal with any last-minute problems.

Authors may need to take specific actions to achieve compliance with funder and institutional open access mandates.

If your research is supported by a funder that requires immediate open access (e.g. according to [a Plan S principles](https://www.springernature.com/gp/open-science/plan-s-compliance) or the [NIH public access policy](https://www.springernature.com/gp/open-science/us-federal-agency-compliance)) then you should select the gold OA route, and we will direct you to the compliant route where possible. Because authors warrant under our subscription licensing terms that they haven't committed to licensing any version of their article under a licence inconsistent with the terms of our agreement – including the applicable embargo period – publication under the subscription model isn't suitable for authors whose funders require no embargo.

You may wish to make your media relations office aware of your accepted publication, in case they consider it appropriate to organize some internal or external publicity. Once your paper has been scheduled you will receive an email confirming the publication details. This is normally 3-4 working days in advance of publication. If you need additional notice of the date and time of publication, please let the production team know when you receive the proof of your article to ensure there is sufficient time to coordinate. Further information on our embargo policies can be found here:

<https://www.nature.com/authors/policies/embargo.html>

If you are active on Twitter/X or Bluesky, please e-mail me your and your coauthors' handles so that we may tag you when the paper is published.

Best regards,
Nina

Nina Vogt, PhD
Senior Editor
Nature Methods

** Visit the Springer Nature Editorial and Publishing website at http://editorial-jobs.springernature.com?utm_source=ejP_NMeth_email&utm_medium=ejP_NMeth_email&utm_campaign=ejp_Nmeth for more information about our career opportunities. If you have any questions please click [here](mailto:editorial.publishing.jobs@springernature.com).

Summary of Revision

We thank all the reviewers for their critical and constructive comments. We have addressed all the points as described below. The data are much strengthened, and we believe that the revision is satisfactory to both reviewers. Here is the summary of the revision.

Fig. 2d, e; Extended Data Fig. 4d, 5f, g. Improved brightness of the fluorescence signals is now quantified for the same sets of neurons in each depth with statistical analyses. (Reviewer #1, comment #1-1)

Fig. 6b, c. We added control data for calcium and voltage imaging to better show the effects of SeeDB-Live. (Reviewer #1, comment #1-2)

Fig. 3a-c; Extended Data Fig. 6a-o. Electrophysiology data were updated. After calibration of the reference (Liquid junction potential), we found that there is no difference in membrane potentials between ACSF and SeeDB-Live. We have also recorded from interneurons and older neurons. (Reviewer #1, comment #2, #5, #8; Reviewer #2, comment #3)

Fig. 2b, c (Supplementary Video 3); Fig. 3j-l (Supplementary Video 10); Fig. 4g, h (Supplementary Video 13). Time-lapse movies of the clearing process are now shown for acute brain slice and the live mouse brain *in vivo* (Reviewer #1, comment 4)

Fig. 5j-n; Extended Data Fig. 7b. We performed long-term chronic imaging for the same sets of neurons *in vivo* (Reviewer #1, comment #3, #5)

Fig. 4k-p; Fig. 5m, n. We evaluated long-term cellular toxicity and animal behavior with control experiments. We found no toxic effects of SeeDB-Live treatment *in vivo*. (Reviewer #1, comment #5; Reviewer #2, comment #5, #6)

Extended Data Fig. 3h-l. Comparison with other clearing methods are shown. (Reviewer #1, comment #6; Reviewer #2, comment 1)

Extended Data Fig. 1e. We characterized the residual salts in commercialized BSA products to better formulate the ionic compositions of SeeDB-Live. (Reviewer #1, comment #5)

In addition, we have taken all the minor comments into account to revise the manuscript.

We have updated some of the figures and movies with more trials. We have also incorporated all the necessary details to the figures and figure legends to improve the readability (Reviewer #1, comment #7). To better show the chronic *in vivo* imaging data, the **old Fig. 4** has been split into **new Fig. 4 and 5**. Instead, the voltage imaging data have been merged into a **new Fig. 6**.

In this rebuttal letter, the reviewers' comments are *italicized*, and our responses are provided for each comment. Changes to **bold texts** have been made by the authors. Figure legends are shown in Arial. Newly added figures and text in this revision are **highlighted** in both the manuscript file and the rebuttal letter.

Reviewer #1 (Remarks to the Author):

*The study by Inagaki et al. introduces a new clearing method for live tissue termed SeeDB-Live. Here, light scattering was minimized by adding spherical polymers with low osmolarity to the extracellular medium, which enables minimally invasive optical clearing for fluorescence imaging of live tissue. The manuscript details the development of the SeeDB-Live clearing media, and using multiple microscopy techniques (confocal, two-photon, widefield), fluorescence was measured in multiple preparations including cell culture, organoids, acute brain slices and in vivo. The authors demonstrate how this clearing medium successfully increases the resolution of images and the depth of two-photon imaging. Overall, **the study suggests a step forward in live tissue clearing**, however, I have some concerns regarding its use and benefit to fluorescence imaging.*

Thank you for your positive comments on our work. We believe that our new strategy will revolutionize live cell imaging. Some of the tough comments were extremely helpful in improving the quality and readability of the manuscript. We believe that our rebuttal letter and the revised manuscript with additional data will fully address all concerns.

Major:

*1. At times, it is questionable how effective SeeDB-Live is at improving fluorescence imaging. The authors state various improvements in the different preparations, from 'up-to' two-fold (Figure 2) and four-fold (Figure 4). Although the images of the brain slices are impressive in Figure 2b and c, the measurements in Figure 2d are less conclusive as the values for control are **not presented for similar depths**. Please include data from the same imaging depths to allow for direct comparison. Was there a **statistical difference** in the normalized intensity with depth using two-photon?*

In the superficial depths (up to 150 μm), we compared control vs. SeeDB-Live. However, in deeper areas (at $> 150 \mu\text{m}$ depth), we cannot find enough numbers of neurons for the control. The brightness is just below the noise level.

We agree that we should show the statistical data. We compared the fluorescence intensity at 50, 100 and 150/200 μm depths (**Fig. R1, also in new Fig. 2d, e**). The brightness was comparable at 50 μm , but significantly higher for SeeDB-Live in the deeper areas. We

hope that these new data are convincing.

Figure R1. Fluorescence intensity in acute brain slices before and after clearing (new Fig. 2d, e).

(a) X-y fluorescence images of an acute brain slice from a Thy1-YFP-H mouse (P22). Layer 5 neurons are labeled with YFP in the cortex. Two-photon images are shown. (b) Comparison of fluorescence intensity in cell bodies. In the superficial areas, the fluorescence signals were comparable. However, fluorescence intensity was significantly higher in deeper regions (>100 μm) following SeeDB-Live treatment, suggesting reduced light scattering in the brain slice. Intensities were normalized to the mean intensity in the control at 50 μm depth. n.s., not significant ($p \geq 0.05$), *** $p < 0.0001$ (Wilcoxon signed-rank test).

Throughout the manuscript, important controls and comparisons are often lacking. For example, it is difficult to assess the improvement to morphology (Figure 2i) and voltage signals (Figure 5d-f and Figure 6) without comparable controls (fluorescent images do not necessarily equate to improved signal-to-noise responses). No controls were provided for the cortical organoid signals (Figure 1q and r), which makes it impossible to assess the benefit of using SeeDB-Live in this preparation.

We respectfully disagree. We did perform control experiments to demonstrate the

improved brightness for all the imaging modalities and samples.

Fig. 2i: This is just a representative data to demonstrate the usefulness of SeeDB-Live. The quantitative comparison for the improved brightness under the same imaging conditions is **already shown in Fig. 2h**.

Old Fig. 5d-f (new Fig. 6e and new Extended Data Fig. 9b,c): The comparison of the brightness with the control is **already shown in old Fig. 5b (new Fig. 6b)**. As you can see, the neurons are almost invisible under the control condition. To fully address your comment, here we show the traces of the Voltron2 signals before and after SeeDB-Live treatment (**Fig. R2**; also in **new Fig. 6c**). Due to the low image contrast, it was extremely difficult to detect spike events under the control. As explained for **Fig. 3j (new Fig. 3m)** and **Extended Data Fig. 5e**, neurons in the superficial part are severely damaged; healthy neurons are located in deeper areas in acute brain slices. In deeper areas, however, we cannot fully visualize them under one-photon imaging under the control ACSF. Therefore, **Fig. 5d-f (new Fig. 6e and Extended Data Fig. 9b,c) demonstrate the application only possible with SeeDB-Live**.

Old Fig. 6 (new Fig. 6g-p): The comparison of the brightness with the control is **already shown in old Fig. 6b-c (new Fig. h, i)**. In the control conditions, it is very difficult to recognize dendrites. It was almost impossible to define ROIs. Therefore, **Fig. 6 demonstrates the unique application only possible with SeeDB-Live**, which would be required for publication in a high impact methods journal.

Old Fig. 1q, r (new Fig. o, p): We have shown the comparison data for the organoids in **Extended Data Fig. 4g-j**, although a quantitative assessment is difficult due to their complicated morphology and its variations. Again, we did not see any meaningful signals for the control for **Fig. 1q and r**.

Figure R2. Epifluorescence voltage imaging in acute brain slices before and after clearing (new Fig. 6c).

(a) Mitral cells in the olfactory bulb slices (P11, 150 μm depth) were imaged before and after SeeDB-Live treatment. **(b)** Voltron2 (labeled with Janelia Fluor HaloTag Ligand549) was introduced to mitral cells by *in utero* electroporation. Ticks indicate the timing of detected action potentials. Due to lower signal-to-noise ratio, it was difficult to reliably detect action potentials for the control.

2. SeeDB-Live appears to affect cellular properties, with significant changes in most cellular biophysiological properties. The authors state that the firing properties are 'largely' preserved, however; most other measurements (aside from the FI curve) is significantly different. This is to be expected due to the process but brings to question the physiological results obtained from cells within SeeDB-Live. Please alter the terminology to better reflect the influence of the media on all biophysiological properties to avoid misleading information. Also, please discuss the possible physiological implication of these changes on neuronal networks and neuronal transmission in the manuscript.

We took this criticism seriously and tried to determine the possible origin of the differences in cellular biophysical properties for the past months. For example, we determined the contaminated ions in BSA product and estimated the exact amount Na^+ as well as free Ca^{2+} and Mg^{2+} as explained below. The recorded membrane potential was different from theoretical values based on Goldman-Hodgkin-Katz (GHK) equation. Eventually, we found that the difference originated from the Liquid Junction Potential

(LJP), the potential of the references under different buffers. It is not so common to determine LJP experimentally, because it is easy to estimate LJP with a simulator. However, experimentally determined LJP was different from prediction with a simulator. We found significant difference between control ACSF and SeeDB-Live/ACSF (13.04 mV for ACSF and 9.19 mV for SeeDB-Live/ACSF). Possibly, this is due to difference in ionic mobility under ACSF and SeeDB-Live/ACSF. **After calibration for LJP, we found no significant differences for the resting membrane potentials** (-77.8 ± 2.2 mV for the ACSF and -78.6 ± 2.9 mV for SeeDB-Live/ACSF) (**Fig. R3a-f**). This important issue is now described in the main text (**Lines 257-264**), **new Fig. 3a**, and **new Extended Data Fig. 6a**.

As for other electrophysiological parameters shown in the **old Extended Data Fig. 6**, we checked original data and noted that there were considerable variations among slices/animals. We, therefore, re-examined the mean values for the neurons from the same sets of animals between ACSF and SeeDB-Live/ACSF (paired analyses). As a result, we did not find statistically significant differences between ACSF vs SeeDB-Live/ACSF (**Fig. R3f**; **new Extended Data Fig. 6i**). We, therefore, believe that the statistical differences found for some parameters (input resistance, membrane capacitance, rheobase, AP half width) in our initial analyses were **derived from inter-sample and/or inter-individual variations**. In the revised manuscript, we show data from all neurons (**new Fig. 3a-c**, **new Extended Data Fig. 6h**) and mean data for each animal (**new Extended Data Fig. 6d, i**).

It should also be noted that we cannot exclude **possible biases in sampling** between ACSF and SeeDB-Live/ACSF. For example, we can easily patch sufficient number of neurons under ACSF. However, it took much longer to patch neurons under SeeDB-Live/ACSF, because neurons are invisible and difficult to find. This can be particularly problematic for sEPSC frequency analysis, where the spontaneous network activity dynamically changes over time during recovery and recording. This point is now described in the legend to **new Extended Data Fig. 6h**. It should also be noted that the spontaneous Ca^{2+} activity data was unchanged by SeeDB-Live treatment (**new Fig. 3d-f**).

We performed additional analyses for fast-spiking interneurons and for older animals, and found no clear difference, as explained later. We apologize for the confusion. We also thank this reviewer for giving us the opportunity to re-evaluate our electrophysiology experiments.

Figure R3. Liquid junction potentials and electrophysiological recording of L5ET neurons at P14-18 (new Fig. 3a-c, new Extended Data Fig. 6a-i).

(a) Recording of liquid junction potentials (LJPs) in ACSF and SeeDB-Live/ACSF. (b) LJP was 13.04 ± 0.20 mV for ACSF and 9.19 ± 0.09 mV for SeeDB-Live/ACSF ($n = 12$). (c) Changes of membrane potentials upon current injection in L5ET neurons. (d) F-I curve. (e) Additional electrophysiological properties in ACSF vs SeeDB-Live/ACSF. Data are from all neurons. Note that some of the parameters were highly variable among animals/samples as shown in (f). (f) Additional electrophysiological properties are compared for the same sets of animals. Data are from mean values for each animal. There were no statistical difference between ACSF and SeeDB-Live/ACSF.

Were there any changes in their electrophysiological properties of other cell types, such as interneurons?

Slice electrophysiology is challenging because neurons are invisible after clearing with

SeeDB-Live under IR-DIC. To address this comment, we focused on fast-spiking interneurons, which are relatively abundant and easy to identify based on their characteristic firing properties. Like L5ET neurons, electrophysiological properties were highly variable among neurons and samples but were largely the same between ACSF and SeeDB-Live/ACSF after calibration for LJP.

Figure R4. Electrophysiological recording of fast-spiking (FS) interneurons in S1 at P14-17 (new Extended Data Fig. 6j-n).

(a) IR-DIC image. (b) Biocytin images. (c) Changes in membrane potentials in response to square current pulses for representative FS interneurons. Characteristic high-frequency APs (~100 Hz) were found. (d) F-I curve. (e) Current responses to the test pulses (-5 mV, 50 ms). (f) Additional electrophysiological parameters recorded under ACSF and SeeDB-Live/ACSF.

3. SeeDB-Live's effect is transient and only lasts for ~1h in vivo and although the clearing reagent can be applied through removable cranial windows, this is still a significant limitation for chronic measurements. The examples of chronic imaging given in Figure 4 are of different neural populations and different imaging depths. Were the repeated applications of SeeDB-Live similarly effective for the same field of view to allow chronic imaging of neural morphological and activity changes over time?

Thank you for raising this issue. We completely agree that we should show the same field of view. We have performed repeated applications of SeeDB-Live on consecutive days and show the same field of view of L5ET neurons in S1 (Fig. R5a-c). The old Fig. 4p-

q have been replaced to the new one (new Fig. 5j-l).

We performed similar experiment for L2/3 (new Fig. 5m, n) and Thy1-YFP-H+ L5ET neurons (new Extended Data Fig. 7b). We demonstrate chronic imaging up to 7 days, which would be sufficient to show the long-term stability of the SeeDB-Live treated neurons *in vivo*. (Fig. R5d)

Figure R5. Chronic imaging with SeeDB-Live for the same field of view (new Fig. 5j-n, Extended Data Fig. 7b).

(a) Basal fluorescence (temporal median) of jGCaMP8m-expressing L5 neurons before and after SeeDB-Live treatment on the first day (day 1). The depth was 658 μm. (b) Basal fluorescence of jGCaMP8m-expressing L5 neurons on the next day (day 2). The depth was 650 μm. (c) Representative Ca²⁺ responses of L5 neurons under SeeDB-Live indicated in (a, b) during repeated whisker stimuli (2 s air puff every 15 s). (d) L5ET neurons before and after clearing with SeeDB-Live on day 0 and 7.

*Furthermore, there does not appear to be an advantage of using SeeDB-Live in the *in vivo* V1 preparation as there was little difference compared to control. Although the data illustrates that neurons may be acting physiologically when bathed in SeeDB-Live *in vivo*, it does question what purpose it provides for functional imaging (in V1).*

It should be noted that the old Fig. 4g-k were performed to make sure that there was

no change in the physiological responses of neurons in V1 when cleared with SeeDB-Live. For this purpose, we have intentionally chosen a relatively superficial layer where the neurons would be fully superfused with SeeDB-Live and the brightness is adequate for both control and SeeDB-Live. Therefore, this is a positive data to demonstrate the usefulness of our method. As is shown in **Fig. R1** (also in **new Fig. 2d,e**), the improvement is not evident when imaging at a relatively superficial layer and normalizing ΔF signals by F_0 . Please note that $\Delta F/F_0$ rather than F_0 is shown for Fig. 4i (new Fig. 5c).

The improved fluorescence and signal-to-noise ratio were particularly evident in deeper areas of the brain (e.g., L5), as shown in **new Fig. 4d-j**, **new Fig. 5f-l**, and **new Extended Fig. 7**.

Unique applications only possible with SeeDB-Live are shown in **new Fig. 4i, j** (spine imaging in L5), **new Fig. 6** (one-photon voltage imaging), and **new Extended Data Fig. 8h, i** (voltage imaging in L5).

4. BSA is stated to infiltrate brain tissue to a depth of 800um. Was the infiltration of SeeDB-Live at different depths quantitatively checked to confirm clearing quality throughout the tissue?

The quantitative data are already in **old Extended Data Fig. 8b**. We now have obtained data from multiple animals (**Fig. R6**). Data are now shown in the main figure (**new Fig. 4b, c**).

Figure R6. Infiltration of BSA into the brain of live animals (new Fig. 4b, c).

(a) Penetration of BSA into the brain of live animals. After the craniotomy and durotomy,

SeeDB-Live with 1% BSA-CF594 (refractive index 1.366, 310 mOsm/kg) was applied on the surface of S1. 1.5 hours later, the brain was isolated without perfusion and immediately frozen. Frozen sections were cut and imaged. **(b)** Graph showing BSA-CF594 fluorescence intensity at different depths of the S1 (n=3 mice). The BSA-CF594 signal penetrated to a depth of ~800 μm .

*How was the imaging changed over the imaging session? **Having a time-lapse measurement of the signal throughout one imaging session would be informative.***

Thank you very much for the very interesting idea of demonstrating a time-lapse measurement (and movie). It should be informative and impressive. Here we performed a time-lapse measurement of acute brain slices (**Fig. R7**). As you can see in **Video R1** (**new Supplementary Video 10**), the brightness of the neurons in deep areas dramatically improved over time. See also time-lapse movie of bright field images (**new Supplementary Video 3**). Very convincing quantitative data are now shown in **Fig. R7** (**new Fig. 3j-l**).

We also performed time-lapse imaging of L5ET neurons in live animals *in vivo* (**Video R2**; also in **new Supplementary Video 13**). Quantitative data are shown in **Fig. R8** (also in **new Fig. 4g, h**).

Video R1. Time-lapse imaging of the GCaMP6f fluorescence during the clearing of acute brain slices with SeeDB-Live (**new Supplementary Video 10**).

See legends to **Fig. R7**.

Figure R7. Time-lapse measurement of the GCaMP6f fluorescence during clearing of acute brain slices with SeeDB-Live (new Fig. 3j-I).

(a) Acute olfactory bulb slices (P9) were imaged with two-photon microscopy at a depth of 150 μm . Basal fluorescence (temporal median) of GCaMP6f and $\Delta F/F_0$ images are shown for different time points. Quantification of the fluorescence signals at cell bodies are shown at the bottom. (b) Basal fluorescence intensity of mitral cell somata during clearing with SeeDB-Live. *** $p < 0.001$; n.s., not significant ($p \geq 0.05$) (Tukey-Kramer multiple comparison test).

Video R2. Time-lapse imaging of the Thy1-YFP-H⁺ L5ET neurons during clearing with SeeDB-Live *in vivo* (new Supplementary Video 13).

See legends to **Fig. R8**.

Figure R8. Time-lapse measurement of Thy1-YFP-H+ L5ET neurons during clearing with SeeDB-Live in vivo (new Supplementary Fig. 4g, h).

(a) Time-lapse images of L5ET neurons in S1 during in vivo clearing with SeeDB-Live. (b) Quantification of fluorescence for the same sets of neurons.

5. The concentration of Ca²⁺ and Mg²⁺ in SeeDB-Live is optimised. Potential artifacts may still arise for long-term in vivo imaging when this media has to be applied and washed multiple times. Did the functional signal-to-noise ratio change because of the application of the media? Did the media affect the firing properties of neurons in the long term in vivo? Did SeeDB-Live influence cell size over time, such as swelling or shrinking?

While we did not observe any major issues for many of the examples shown in the paper, we fully understand these additional concerns. The signal-to-noise ratios are much better with SeeDB-Live as explained earlier (see also, **Figs. R2, R5, and R7**), since the tissues are more transparent.

To address the concern about the long-term effects, we performed long-term imaging combined with *in vivo* clearing with SeeDB-Live (up to 7 days). To examine potential toxicity, we focused on L2/3 neurons in S1, where SeeDB-Live fully infiltrate (**Fig. R6**). We examined soma size, $\Delta F/F_0$ of the responses to whisker stimuli, and half-rise time of the responses. We did not find any changes in morphology as well as functional responses over time (**Fig. R9; new Fig. 5m, n**). We, therefore, conclude that there is no clear toxicity to neurons in vivo even after repeated clearing with SeeDB-Live.

Figure R9. Long-term morphological and functional imaging *in vivo* (new Fig. 5m, n).

(a) Long-term monitoring of neuronal morphology and physiology. L2/3 neurons in S1 were visualized with AAV-jGCaMP8m-P2A-CyRFP1. Maximum projection images of the basal fluorescence (temporal median) of jGCaMP8m-expressing L2/3 neurons after SeeDB-Live treatment on Day 0, 3 and 7. The imaging depth was 238 to 358 μm . **(b)** Somata size, $\Delta F/F_0$ and half-rise time of jGCaMP8m-expressing L2/3 neurons to repeated whisker stimulation after SeeDB-Live treatment on Day 0, 3, and 7.

6. The use of SeeDB-Live was not tested against existing live clearing methods. Quantitative comparisons, such as signal-to-noise ratio and resolution improvement, would help demonstrate the strength of using this new media. Although direct experimental comparisons would be preferable, a detailed comparison in the discussion is required as a minimum.

Some chemicals have been used to clear skin under unhealthy conditions; however, **NO tissue clearing agents has ever been successfully used for live cell imaging.** As discussed in detail for **Fig. 1**, cell-permeable chemicals such as glycerol were highly toxic to cultured cells (**old Fig. 1e; new Fig. 1c**) and brain slices (**old Extended Data Fig. 7c-e; new Extended Data Fig. 6r-t**). A non-cell-permeable chemical, iodixanol, also demonstrated some toxicity to cultured cells (**old Fig. 1i, j; new Fig. 1g, h**) and brain slices (**old Extended Data Fig. 7a, b; new Extended Data Fig. 6p, q**). Therefore, it is impossible compare the signal-to-noise ratio and resolution in functional imaging (e.g., Ca^{2+} imaging).

Very recently, two other groups have reported tissue clearing agents that claim to clear live tissues (Ou et al., *Science* 2024, <https://doi.org/10.1126/science.adm6869>; Franzesi et al., *bioRxiv* 2024, <https://doi.org/10.1101/2024.09.05.611421>), and this reviewer may perhaps refers to these papers. **We have now performed a quantitative comparison** as shown in **Fig. R10 and R11**.

Ou et al. (*Science* 2004) reported tissue clearing with **0.6 M tartrazine** solution. However, this solution is **extremely hypertonic and toxic (Fig. R10)**. Therefore, this solution cannot be used for live cell imaging. We then considered a lower concentration of tartrazine. However, **tartrazine only had a refractive index of 1.351 when adjusted to isotonic conditions (300 mOsm/kg)**, which was not effective at all in reducing light scattering of live cells (**Fig. R11a-c**). Therefore, **tartrazine cannot be applied for live cell imaging**. It should be noted that Ou et al. only cleared skin to better visualize the underlying organs, and **did not demonstrate functional imaging of cleared tissue itself**.

Franzesi et al (*bioRxiv* 2024) is based on a similar concept as our current study. However, according to our analysis (**old Fig. 1f; new Fig. 1d and Fig. R10a**), iodixanol, dextran, and PEG (straight polymer) have higher osmolality at the ideal refractive index, and therefore we excluded them from our candidates in our initial screening. Moreover, **the refractive indices of the clearing media reported by Franzesi et al. were much lower than the ideal range** (RI 1.344 for 6.2% iodixanol, 1.340 for 4% dextran, and 1.343 for 6% PEG, respectively), resulting in **much lower clearing efficiency**. Direct comparison of the light transmittance demonstrated that SeeDB-Live was most effective in clearing live cells (**Fig. R11c**). Under the isotonic condition, SeeDB-Live best cleared acute brain slices (**Fig. R11d, e**).

Together, our results demonstrate that **SeeDB-Live is the best in terms of tissue transparency and osmolality**. We hope that these new data are convincing to this reviewer.

Figure R10. Osmolality of SeeDB-Live and other clearing agents (new Fig.1d and Extended Data Fig. 3i).

(a) The osmolality of candidate chemicals in water (refractive index 1.365, in ddH₂O, n = 3 each). The concentration was adjusted so that the refractive index of the solution becomes 1.365. Sucrose was used as a control. BSA#1 and BSA#2 represent two examples of different BSA products (BSA#1, crystal; BSA#2, low salt). The osmolality of the low-salt BSA (#2) was 2.7 mOsm/kg. Tartrazine and PEG 10K solution were prohibitively hypertonic (528 mOsm/kg) at RI 1.365. Iodixanol and dextran showed moderate osmolality (95 and 67 mOsm/kg) at RI 1.365. **(b)** Phase contrast and fluorescence images of GFP-expressing HEK293T cells. The cells were immersed in SeeDB-Live (RI 1.363, 296 mOsm/kg) followed by 0.6 M tartrazine solution (RI 1.43, 1407 mOsm/kg). The cells shrank, indicating that tartrazine cannot permeate into live cells and that 0.6 M tartrazine is highly hypertonic. Scale bar, 50 μ m. **(c, d)** Primary culture of mouse cardiomyocytes soaked in 0.6 M tartrazine solution. The muscle fibers shrank, indicating that 0.6 M tartrazine is hypertonic **(c)**. In addition, the rhythmic contraction of the cardiomyocyte quickly ceased under 0.6 M tartrazine **(d)**. Thus, 0.6M tartrazine is extremely toxic to live cells. Scale bar, 30 μ m.

Figure R11. Comparison of SeeDB-Live and other clearing agents tested *in vivo* (new Fig. 2b, c and Extended Data Fig. 3h, j, k).

(a) HeLa cells were immersed in 0.1 M tartrazine solution (RI 1.351, adjusted to isotonic condition of 310 mOsm/kg) or SeeDB-Live (RI1.363, 300 mOsm/kg). Isotonic tartrazine solution did not optically clear HeLa cells. SeeDB-Live efficiently cleared the cells. Scale bar, 50 μ m. **(b)** Transmittance of HeLa cell suspension (4×10^6 cells/mL) at different wavelengths. Due to high absorption of tartrazine, we could not obtain transmission below 550 nm for the tartrazine solution. **(c)** Refractive index, osmolality, and transmittance (at 600 nm) were compared among clearing agents reported for live tissues. Red indicates ideal ranges for optical transparency and minimal toxicity for live cells. Osmolality was within acceptable range except for tartrazine solution. However, only SeeDB-Live was within the ideal range of refractive index. SeeDB-Live showed the highest clearing efficiency of live cells under the physiological osmolality condition. **(d)** An acute brain slice (300 μ m thick, age P3) was cleared with SeeDB-Live. **(e)** Acute brain slices (300 μ m thick, age P4) cleared with 4% Dextran/ACSF and SeeDB-Live are compared. The highest transparency was also found for SeeDB-Live with acute brain slices. Magnified images for the yellow boxes (cerebral cortex) are shown on the bottom. Note that the transmission image of 4% Dextran/ACSF is omitted in the manuscript, because this is not yet published in a journal. See also Supplementary Video 3 for the time-lapse movie of the clearing process with SeeDB-Live.

7. Overall, the text and Figure captions require more information. Often information included in the Figures are not yet included in the text – for example, SeeDB-Live is illustrated in Figure 1d but not yet described in the corresponding text. Likewise, in Figure 1b, what are straight polymer; spherical polymer (PVA, Ficoll etc). Figure 1p – please expand to best represent and visualize the data. The yellow box in Figure 2g SeeDB-Live appears to be incorrectly placed. It is unclear in Figure 4 which data used jGCaMP8m and which used Cal-520. What does the arrow indicate in Figure 5f?

We apologize for the poor readability and layout of the figures.

The old Fig. 1d has been moved to the end of Fig. 1, after the introduction of SeeDB-Live formula in the main text.

The definition of spherical and straight polymers is now described in the main text (Line 142-143) and figure legends (Line 618-619).

The yellow box in the bottom left of Fig. 2g was incorrectly placed, and we thank you for

pointing this out.

The type of the sensors is now added to each panel of **new Fig. 5**. The **new Fig. 5a-e** only uses jGCaMP8m. Cal520 data have been moved to **Extended Data Fig. 8a-e**.

The arrow in **old Fig. 5f (new Fig. 6e)** indicate the initiation of AP and this is now mentioned in the figure legends (**Line 837-828**).

8. Why were young brain slices used to assess the electrophysiological properties of neurons cleared with SeeDB-Live? P14-18 is pre-weaning and considered immature – and developmentally younger than the in vivo measurements using Thy1 mice (P22 and 6 month old).

We have chosen P14-18 for two reasons. Firstly, it is considered that membrane properties of L5 pyramidal neurons mature by ~P14 (<https://doi.org/10.1152/jn.00855.2003>). More specifically, the resting membrane potential become stable by this period (<https://doi.org/10.1152/jn.00855.2003>). Secondly, it is well recognized that adult mouse brain slices are very difficult to prepare; adult neurons are easily damaged during slice preparation under the classical ACSF. Therefore, many electrophysiologists traditionally preferred to use young brain slices. For the same reasons, Ca²⁺ imaging of acute brain slices is most commonly performed for younger animals (up to ~4W). We therefore used younger animals in our initial submission.

However, we agree that it is important to see older animals. Also, as we discussed earlier, younger animals are sometimes problematic there are variations in maturation among individuals. Here we performed electrophysiological recording from P28-29 animals using high-sucrose cutting solution developed for older animals (Sun et al., 2020). The L5ET data from P28-29 are shown in **Fig. R12 (new Extended Data Fig. 6o)**. Overall, the results were consistent with P14-18 data.

Figure R12. Electrophysiological recording of L5ET neurons in S1 from P28-29 mice (new Extended Data Fig. 6o).

(a) IR-DIC image. **(b)** Biocytin images. **(c)** Changes in membrane potentials in response to square current pulses for representative L5ET neurons. **(d)** Current responses to the test pulses (-5 mV, 50 ms). **(e)** Electrophysiological parameters recorded under ACSF and SeeDB-Live/ACSF.

Reviewer #2 (Remarks to the Author):

The study of Shigenori Inagaki and colleagues proposes a clearing procedure containing bovine serum albumin (BSA) named SeeDB-Live, with refractive index of 1.363-1.366, osmolality 230-330 mOsm/kg, and Ca²⁺/Mg²⁺ concentrations of 2-4 mM, for structural and functional imaging of live tissues, such a spheroids, organoids, acute brain slices, and the mouse brain in vivo.

*The study presents interesting findings, however, there are some important aspects that the authors did not consider and **need to be evaluated**:*

Thank you for recognizing the exciting aspects of our work. We are also grateful that you have raised some important questions that the general audience would also have. We believe that in this rebuttal you will be fully convinced of the major advance of our new method, especially the conceptual difference from the recent tartrazine paper in *Science*.

*1) A recent paper “Achieving optical transparency in live animals with absorbing molecules” published this summer by Science already introduced an approach based on Tartrazine to enable invivo imaging of cleared samples. The authors didn’t mention this approach nor compare their results with those already published. Tartrazine is a common food color approved by the US Food and Drug Administration that has the effect of reversibly making the skin, muscle, and connective tissues transparent in live rodents. Why should a user choose SeeDB-Live **instead of tartrazine** as Zihao Ou et al proposed?*

Thank you for raising this issue. We have cited this tartrazine paper as one of the high osmolarity clearing agents; however, we agree that more direct comparison with additional data is informative and helpful for the general audience.

As shown in **Fig. R10** (also shown in our response to Reviewer #1, comment #6), **the osmolality of the tartrazine solution reported in the Ou et al. is prohibitively high and toxic to the cells. When tartrazine solution was made isotonic (300 mOsm/kg), the refractive index was too low (1.351) to make live cells transparent (Fig. R11a).** Therefore, we believe that **tartrazine is not useful at all for live cell clearing and functional imaging.** It should be noted that Ou et al. only achieved optical clearing of juvenile mouse skin, and not of any other tissues. Moreover, as the skin is under extremely

toxic condition, they applied tartrazine only transiently to minimize the damage to the skin. They also used tartrazine to image organs under the skin after skin clearing, but they did not clear the organs themselves.

In our opinion, the tartrazine paper was extremely misleading. It does make the skin transparent, but under extremely high osmolality conditions, which prevents its use for live cell imaging. **This point is now explicitly mentioned in the main text of the revised manuscript (Line 189-194).**

We also compared the performance of the clearing agents reported in another recent study by Franzesi et al (*bioRxiv* 2024, <https://doi.org/10.1101/2024.09.05.611421>) (Fig. R10-11). Franzesi et al. mainly used 6% PEG and 4% dextran for *in vivo* clearing. However, we found that the refractive index was not fully optimized in their study. **The refractive index was only 1.340-1.344**, in contrast to the optimal range of 1.360-1.370. As a result, the transparency of the live cells was much lower than with SeeDB-Live.

We therefore conclude that **SeeDB-Live best achieves optical transparency under physiological osmolality and ionic conditions**, paving the way for its application to live cell imaging of physiological responses.

Figure R10. Osmolality of SeeDB-Live and other clearing agents (new Fig.1d and Extended Data Fig. 3i).

(a) The osmolality of candidate chemicals in water (refractive index 1.365, in ddH₂O, n = 3 each). The concentration was adjusted so that the refractive index of the solution becomes 1.365. Sucrose was used as a control. BSA#1 and BSA#2 represent two examples of different BSA products (BSA#1, crystal; BSA#2, low salt). The osmolality of the low-salt BSA (#2) was 2.7 mOsm/kg. Tartrazine and PEG 10K solution were prohibitively hypertonic (528 mOsm/kg) at RI 1.365. Iodixanol and dextran showed moderate osmolality (95 and 67 mOsm/kg) at RI 1.365. **(b)** Phase contrast and fluorescence images of GFP-expressing HEK293T cells. The cells were immersed in SeeDB-Live (RI 1.363, 296 mOsm/kg) followed by 0.6 M tartrazine solution (RI 1.43, 1407 mOsm/kg). The cells shrank, indicating that tartrazine cannot permeate into live cells and that 0.6 M tartrazine is highly hypertonic. Scale bar, 50 μ m. **(c, d)** Primary culture of mouse cardiomyocytes soaked in 0.6 M tartrazine solution. The muscle fibers shrank, indicating that 0.6 M tartrazine is hypertonic **(c)**. In addition, the rhythmic contraction of the cardiomyocyte quickly ceased under 0.6 M tartrazine **(d)**. Thus, 0.6M tartrazine is extremely toxic to live cells. Scale bar, 30 μ m.

Figure R11. Comparison of SeeDB-Live and other clearing agents tested in vivo (**new** Fig. 2b, c and Extended Data Fig. 3h, j, k).

(a) HeLa cells were immersed in 0.1 M tartrazine solution (RI 1.351, adjusted to isotonic condition of 310 mOsm/kg) or SeeDB-Live (RI1.363, 300 mOsm/kg). Isotonic tartrazine solution did not optically clear HeLa cells. SeeDB-Live efficiently cleared the cells. Scale bar, 50 μ m. (b) Transmittance of HeLa cell suspension (4×10^6 cells/mL) at different wavelengths. Due to high absorption of tartrazine, we could not obtain transmission below 550 nm for the tartrazine solution. (c) Refractive index, osmolality, and transmittance (at 600 nm) were compared among clearing agents reported for live tissues. Red indicates ideal ranges for optical transparency and minimal toxicity for live cells. Osmolality was within acceptable range except for tartrazine solution. However, only SeeDB-Live was within the ideal range of refractive index. SeeDB-Live showed the highest clearing efficiency of live cells under the physiological osmolality condition. (d) An acute brain slice (300 μ m thick, age P3) was cleared with SeeDB-Live. (e) Acute brain slices (300 μ m thick, age P4) cleared with 4% Dextran/ACSF and SeeDB-Live are compared. The highest transparency was also found for SeeDB-Live with acute brain slices. Magnified images for the yellow boxes (cerebral cortex) are shown on the bottom. See also Supplementary Video 3 for the time-lapse movie of the clearing process with SeeDB-Live.

Please note that the transmission image of 4% dextran/ACSF (Franzesi et al., *bioRxiv*) is not included in the current version of the manuscript, because Franzesi et al. has not yet been published in a journal at this moment. We would be happy to follow the advice of editors and reviewers.

2) In the past years several articles presented different clearing in vivo, and the majority of them use RI between 1.4 and 1.5 (as mentioned in the review: “In-vivo and ex-vivo optical clearing methods for biological tissues”). On the contrary, the authors claim that “the optimal range of the refractive index for live cells was relatively narrow; the transparency of live cells became lower at higher refractive indices (>1.38).”. This is not in line with the findings already presented. The difference relies on the fact that impermeable chemicals can not enter the cells, therefore the approach used limited the finding. Therefore the general conclusion “These results indicate that the major source of light scattering in live cells is due to the index mismatch between the cytosol (~1.37) and the extracellular medium (typically 1.33-1.34), and thus index matching of the extracellular medium (~1.37) is the key to achieving maximum optical transparency of live mammalian tissues” is not completely true. Using a hypothetical chemical

permeable to the membrane, with high RI, that matches the RI of all the cell components, the results could be better. The authors should clarify this point.

We agree. In the revised manuscript, we will mention in the revised text that a hypothetical cell-permeable chemical may achieve optical clearing at RI 1.4-1.5. In fact, this is exactly what we have been doing for fixed cells.

However, in reality, all the chemicals tested to date (listed in the review article) are toxic. Therefore, live imaging has been impossible with currently known cell-permeable clearing media. In our opinion, it would be highly unlikely that cell-permeable chemicals at RI 1.4-1.5 (in the 5-10M concentration range) would not interfere with intracellular functions at all.

It was indeed a big surprise to us that the highest optical transparency can be achieved at a much lower refractive index when using non-cell-permeable chemicals. This unexpected finding was the basis for our current study.

All these suggestions are now taken into account. We have revised the description as follows (Lines 220-225).

“Together, our results indicate that the light scattering in live cells can be greatly reduced by index matching between the cytosol (~1.363) and the extracellular medium (typically 1.33-1.34) without the use of cell-permeable chemicals that would easily interfere with cellular functions. Index matching of the extracellular medium with isotonic medium with BSA (SeeDB-Live) is minimally invasive but powerful for optical clearing of live mammalian tissue (Fig. 1q).”

We also understand that it would be helpful to show **a table comparing some key aspects of the previous and current *in vivo* clearing methods**. We have newly prepared **Table R1**.

RI reagent	Transparency of live cells	Membrane permeable	Refractive index	Osmolality (mOsm/kg)	physiological ionic conditions	Ca ²⁺ responses in cultured cells	Cell growth in cultured cells	Functional imaging in tissues	Applications	Reference
15-17% w/v BSA (SeeDB-Live)	+++	NO	1.363-1.366	280-320	YES	YES	YES	YES	live cells	This study
20% w/v Ficoll70	+++	NO	1.363-1.366	280-320	YES	YES	Suboptimal	Suboptimal	live cells	This study
6% w/v PEG10K	+	NO	1.343	330-340	YES	NO	NO	-	live cells	Franzesi et al., 2024
4% w/v Dextran	+	NO	1.34	290-300	YES	-	-	YES	live cells	Franzesi et al., 2024
6.2% w/v Iodixanol	+	NO	1.344	320-335	YES	YES	-	YES	live cells	Franzesi et al., 2024
20% w/v Iodixanol	+++	NO	1.363-1.365	280-300	YES	YES	Suboptimal	NO	live cells	Boothe et al., 2017
0.6 M Tartrazine	++	NO	1.42	1743	NO	NO	NO	NO	Skin	Ou et al., 2024
38% w/v Ampyrone	-	NO	1.43	846	NO	NO	NO	NO	Skin	Keck et al., 2025
40% Glucose	-	NO	1.39	3158	NO	NO	NO	NO	Skin etc	Bashkatov et al., 2001
88% v/v Glycerol	-	YES	1.46	-	NO	NO	NO	NO	Skin etc	Bashkatov et al., 2000
64.7% Iohexol	-	NO	1.43	465	NO	NO	NO	NO	Skin etc	Sdobnov et al., 2017

Table R1. Comparison with other *in vivo* clearing methods (new Extended Data Fig. 3I).

The live tissue clearing agents should be isotonic and minimally invasive to live cells. We compared several key factors required for live cell clearing. Transparency of live cells were evaluated based on the transmittance of cell suspension (4×10^6 cells/mL) at 600 nm. +++; >40% transmittance, ++; >20% transmittance, +; 0-20%, -; not measured. 0.6M tartrazine, 38% ampyrone, 40% glucose, and 88% glycerol are tolerated in live animals but are toxic to cellular functions. 20% Ficoll70 and 20% iodixanol was suboptimal in terms of cellular and/or neuronal functions. 6% PEG10K, 4% dextran, and 6.2% iodixanol are suboptimal in terms of transparency.

3) The resting membrane potential of neurons cleared with SeeDB-Live results lower as well as the threshold membrane potential for the action potential. Can the authors explain this behavior? For how long does the behavior maintain this characteristic?

We thank this reviewer for raising this important issue. We agree that the slightly lower membrane potential can be a significant concern for the users. We took this issue seriously and tried to determine the possible origin of the differences.

We initially considered the possibility that the lower resting membrane potential is due to lower ionic concentration of the medium based on the prediction from Goldman-Hodgkin-Katz (GHK) equation. We, therefore, determined the exact amounts of salts remained in the BSA product used in this study (BSA #1) using Inductively Coupled Plasma Mass Spectrometry (ICP-MS) (Table R2; new Extended Data Fig. 1e). Based

on this information, we estimated the membrane potential but the predicted value was inconsistent with our initial result.

We, therefore, considered the possibility that the recorded membrane potential was wrong. Eventually, we found that the difference originated from the Liquid Junction Potential (LJP), the potential of the references under different buffers. It is not so common to determine LJP experimentally, because it is easy to estimate LJP with a simulator. However, experimentally determined LJP was different from prediction with a simulator. We found significant difference between control ACSF and SeeDB-Live/ACSF (13.04 mV for ACSF and 9.18 mV for SeeDB-Live/ACSF). Possibly, this is due to difference in ionic mobility under ACSF and SeeDB-Live/ACSF. **After calibration for LJP, we found no significant differences for the resting membrane potentials** (-77.8 ± 2.2 mV for the ACSF and -77.6 ± 2.9 mV for SeeDB-Live/ACSF; mean \pm SD) (**Fig. R3**). This important issue is now described in the main text (**Lines 257-264**), **new Fig. 3a-c**, and **new Extended Data Fig. 6a**.

The reliability of our electrophysiology data was much improved, and we thank this reviewer for asking this point.

	Na ⁺ (mM)	K ⁺ (mM)	Ca ²⁺ (mM)	Mg ²⁺ (mM)
BSA#1 Lot 1	33.3	0.1	1.2	0
BSA#1 Lot 2	35.4	0.1	1.3	0
BSA#1 Lot 3	34.9	0.1	1.3	0
BSA#2	3.0	0	0.1	0

Table R2. Residual salts contained in BSA products #1 and #2 (new Extended Data Fig. 1e).

Salts contained in BSA powder were analyzed using Inductively Coupled Plasma Mass Spectrometry (ICP-MS). Ionic concentrations of BSA solution at a refractive index of 1.365 (17% w/v) are shown.

Figure R3. Liquid junction potentials and electrophysiological recording of L5ET neurons at P14-18 (new Fig. 3a-c, new Extended Data Fig. 6a-i).

(a) Recording of liquid junction potentials (LJPs) in ACSF and SeeDB-Live/ACSF. (b) LJP was 13.04 ± 0.20 mV for ACSF and 9.19 ± 0.09 mV for SeeDB-Live/ACSF ($n = 12$). (c) Changes of membrane potentials upon current injection in L5ET neurons. (d) F-I curve. (e) Additional electrophysiological properties in ACSF vs SeeDB-Live/ACSF. Data are from all neurons. Note that some of the parameters were highly variable among animals/samples as shown in (f). (f) Additional electrophysiological properties are compared for the same sets of animals. Data are from mean values for each animal. There was no statistical difference between ACSF and SeeDB-Live/ACSF.

4) For *how long* the spontaneous Calcium activity is constant with SeeDB-Live clearing?
And for the functional imaging *in vivo*?

For *ex vivo* imaging, we were able to image up to ~5 hours, similar to the control

conditions. This is now explained in the Methods section (Line 1196).

As for the *in vivo* application, the transparency will gradually decrease as the SeeDB-Live is washed out by the endogenous circulation system. Transparency was maintained at least up to 1 hour. Transparency is lost once SeeDB-Live is cleared by the CSF circulation or by active washout (Fig. R13; new Extended Data Fig. 7a). This technical issue is now mentioned in the main text (Line 227-228).

Figure R13. Washout of SeeDB-Live (new Extended Data Fig. 7a).

The primary somatosensory cortex (S1) of a 4-month-old Thy1-EYFP-H mouse was imaged using two-photon microscopy before SeeDB-Live treatment, after 1 hour of clearing with SeeDB-Live, and after 3 hours of washout with ACSF. L5ET neurons were labeled with EYFP in Thy1-YFP-H mice.

5) Are there some toxic effects observed in the animal after the clearing? In the paper, there is a demonstration of the possibility of repeating the clearing after one day for functional recording however; to chronic imaging it is necessary to demonstrate the compatibility of the technique for a longer time. Have you monitored the animals for a long exposure (eg a month)?

Since the SeeDB-Live treatment is only transient (1-2 hours) and for just a part of the brain (cortex for our examples), we do not expect any toxicity. It should also be noted that albumin is the most abundant protein in the CSF. The concentration of BSA used in SeeDB-Live is only twice higher than the total concentration of the proteins in the serum (7-8%) as described in the main text (Lines 187-188).

Nonetheless, we understand the concern about possible toxicity. To address the concern about the long-term effects, we performed long-term imaging combined with *in vivo* clearing with SeeDB-Live (up to 7 days). To examine potential toxicity, we focused on L2/3 neurons in S1, where SeeDB-Live fully infiltrate (Fig. R6). We examined soma size, $\Delta F/F_0$ of the responses to whisker stimuli, and $\tau_{on1/2}$ of the responses. We did not find any changes in morphology as well as functional responses over time (Fig. R9; new Fig. 5m, n). We, therefore, conclude that there is no clear toxicity to neurons *in vivo* even after repeated clearing with SeeDB-Live.

Figure R9. Long-term morphological and functional imaging *in vivo* (new Fig. 5m, n).

(a) Long-term monitoring of neuronal morphology and physiology. L2/3 neurons in S1 were visualized with AAV-jGCaMP8m-P2A-CyRFP1. Maximum projection images of the basal fluorescence (temporal median) of jGCaMP8m-expressing L2/3 neurons after SeeDB-Live treatment on Day 0, 3 and 7. The imaging depth was 238 to 358 μm . **(b)** Somata size, $\Delta F/F_0$ and half-rise time of jGCaMP8m-expressing L2/3 neurons to repeated whisker stimulation after SeeDB-Live treatment on Day 0, 3, and 7.

6) Have you noticed any alteration in the behavior of the animals during and/or after the clearing?

Not for now. BSA is removed by the circulation system and should be drained into the lymphatic system. The serum already contains ~5% of albumin and it is unlikely that additional albumin will affect physiology of the animals.

Nonetheless, we agree that it is important to ensure the SeeDB-Live does not affect animal behavior. Here we tested both acute and long-term effects.

As for the acute effects, we cleared the right hemisphere of the cortex (including motor cortex) with SeeDB-Live in awake animals and examined locomotor activity using a treadmill. We did not find obvious changes by SeeDB-Live treatment compared to ACSF (Fig. R14a, b; new Fig. 4k, l).

We also tested chronic effects on behavior to examine possible long-term toxicity. Locomotor activity in open chamber was unchanged after the SeeDB-Live treatment (Day 1, 4, 7) (Fig. R14c; new Fig. 4m).

In another experiment, we performed wire hang test, which requires motor cortex. In the control experiment with ischemia on the right hemisphere. The performance was reduced. However, in the SeeDB-Live group, we did not find any reduction in performance, suggesting that motor cortex is intact by SeeDB-Live treatment (Fig. R14d, e; new Fig. 4n, o).

Food intake was also unchanged suggesting that animals were overall healthy after SeeDB-Live treatment at least up to 1 week (Fig. R14f; new Fig. 4p).

In summary, we found no evidence of toxicity to neurons, cortex, or animals in either the short or long time window.

Figure R14. Long-term effects on animal behavior (new Fig. 4k-p).

(a, b) Mouse locomotor activity on a treadmill was measured for 10 min during clearing with SeeDB-Live in head-fixed awake animals (a). The total distance traveled and the maximum

speed of mice treated with ACSF and SeeDB-Live were compared **(b)**. **(c)** Locomotion assay. Total distances traveled by mice in an open chamber at 1, 4, and 7 days after treatment with ACSF and SeeDB-Live are shown. n.s., not significant ($p \geq 0.05$) (Wilcoxon rank-sum test). **(d, e)** Motor function was examined with the wire hang test 54. As a control, ischemia was induced by injection of a photosensitive dye (Rose Bengal) and photostimulation of the right cortical surface **(d)**. Fall time of mice in the wire hanging test at 1, 4, and 7 days after treatment with ACSF, Rose Bengal, and SeeDB-Live **(e)**. $**p < 0.001$; n.s., not significant ($p \geq 0.05$) (Tukey-Kramer multiple comparison test). **(f)** Food intake of mice treated with ACSF, ischemia, and SeeDB-Live. $***p < 0.001$; $**p < 0.01$; n.s., not significant ($p \geq 0.05$) (Tukey-Kramer multiple comparison test).

We thank both reviewers for asking tough but critical questions. We have responded to all concerns with a lot of new data. We would like to thank the reviewers for giving us the opportunity to correct the critical electrophysiology data with LJP calibration. We believe that all concerns, especially the advantages of our method over previous methods and the potential toxicity, have been fully addressed in this revision. We hope that this revision is now satisfactory to both reviewers.

Summary of Revision

Figure 3a-e, Extended Data Figure 6. The electrophysiology data has been updated. Now, we performed recording of spontaneous EPSC at the same timing after sample preparation between the control and SeeDB-Live. We now find no statistical difference for sEPSC frequency between the control and SeeDB-Live (**new Fig. 3e**). (Reviewer #1, comment #1, 5)

Description of electrophysiology data. Extended Data Fig. 2c-e. We mentioned some differences in electrophysiology data between the control and SeeDB-Live and discussed possible reasons for the difference. (Reviewer #1, comment #1)

Figure 5n-p. We performed chronic imaging up to four months with no obvious abnormality in neuronal responses. (Reviewer #2, comment #1)

Extended Data Figure 7b-g. We evaluated inflammatory responses by SeeDB-Live treatment. There was no obvious sign of microglial activation by SeeDB-Live treatment. (Reviewer #2, comment #3)

We have also included as much experimental detail as possible in the text, figure legends, and figures, according to the reviewers' suggestion. We have also improved the terminology to avoid any confusion (e.g., "widefield imaging" was changed to "epifluorescence imaging").

In the rebuttal letter, the reviewers' comments are *italicized*, and our responses are provided for each of them. Newly added figures and revisions in the text are **highlighted** in both the manuscript file and the rebuttal letter.

Reviewer #1:

Remarks to the Author:

The manuscript by Inagaki et al has improved from the initial submission. New experiments and analysis have been performed which is appreciated. I still have concerns to my previous comments, and the addition of new data/analysis has raised further questions.

1) A major change in the revised manuscript is the measurement of the liquid junction potential, which was found to alter the reporting of the electrophysiological recordings. I am very glad this was realized as otherwise the wrong results would be reported. However, I am still concerned about the electrophysiological results as there are some results which need greater attention. For example, in Figure R3 (Figure 3, Extended Data Fig. 6), how can there be no effect on AP frequency (Figure 3c and R3d) even though there is a significant difference in input resistance and rheobase? Why is there a difference in EPSP frequency or amplitude in some preparations, and not others? Although the authors state that there is variability in slices, this is not an adequate explanation (especially with only 5/6 data points). Also, the influence on the electrophysiological properties differed in the different ages of the neurons tested which makes for great confusion. This data needs to be highlighted and discussed, and simply referring to the firing rate being 'largely preserved' does not correctly represent the data.

Overall, it appears as though SeeDB-Live influences cellular properties, and the authors should make it extremely clear what is influenced to alert future users of any limitations.

We are glad to find that this reviewer was satisfied with most of our previous responses. We totally understand and appreciate the remaining concerns by this reviewer.

We agree that some of the differences should be highlighted and discussed. However, it is equally important to consider the possibility that some of the differences between ACSF and SeeDB-Live/ACSF found in the previous manuscript are due to sampling bias, as discussed below.

1) It should be noted that **it is extremely difficult to patch neurons under SeeDB-Live,**

because neurons are almost invisible even under IR-DIC (because neurons are transparent). Therefore, we ejected the internal solution with the lower refractive index from the pipette to better visualize target neurons (now in Lines 1366-1368). Therefore, it usually takes much longer to record from neurons under SeeDB-Live.

2) Related to this point, healthy neurons are highly transparent under SeeDB-Live. However, it is known that the refractive index of live cells changes as they become unhealthy (e.g., Rais et al., 2024). As a result, unhealthy neurons become slightly more visible than healthy cells under SeeDB-Live. Therefore, it is possible that recorded neurons are more biased towards unhealthy conditions in SeeDB-Live (even though we tried our best to avoid these neurons).

3) For P14-18 L5ET neurons in the previous revision, we recorded slices from the same sets of animals for ACSF and SeeDB-Live/ACSF. Admittedly, we first recorded under ACSF and then in SeeDB-Live/ACSF, because recording is much easier under ACSF. Therefore, **there were the differences in the recording timepoint**. This may explain the differences in the electrophysiology data, especially the spontaneous EPSC (sEPSC).

To minimize the potential bias due to the point 3), **we have newly performed the electrophysiology experiments for P15-18 S1 L5ET neurons under the same conditions and timepoint**. Specifically, both the control and SeeDB-Live samples were recorded within 4 hours after clearing. To minimize the potential sampling bias, recording was first performed under the control ACSF for the half of the experiments; for the remaining half, the recording was first performed under the SeeDB-Live/ACSF. In this way, now the differences in sampling conditions were minimized. As a result, we found smaller differences between the control and SeeDB-Live, compared to the previous datasets. For example, we now found no statistical differences in sEPSC frequency and amplitude between the control and SeeDB-Live (**Fig. R15**). It should also be noted that the results are now more consistent across ages (P15-18 vs. P28-29 L5ET) and cell types (L5ET vs L5 fast-spiking).

We have also updated **new Fig. 3a-e** and **new Extended Data Fig. 6a-e**. We thank this reviewer for pointing out this important issue.

Figure R15. New slice electrophysiology experiments (new Fig. 3a-e and new Extended Data Fig. 6a-e).

Electrophysiological recordings were performed within 4 hours of sample preparations. The recording timepoint was equalized between the control and SeeDB-Live groups. L5ET neurons in S1 (P15-18) were recorded. **(a)** IR-DIC images. **(b)** Biocytin staining of recorded neurons. **(c)** Current responses of the test pulses. **(d)** Responses to square current pulses. **(e)** Current-frequency curve. **(f)** Spontaneous currents at a holding potential of -60 mV. **(g)** Electrophysiological parameters under the control and SeeDB-Live conditions. Data from different animals are indicated in different colors, indicating the minimal variations among animals, unlike the previous experiments (see previous revision). Some parameters were different between the two conditions (e.g., AP threshold, input resistance, and membrane capacitance). However, the firing properties in the frequency-current curve were not significantly affected, possibly because differences in some factors counteracted each other. For example,

an increase in input resistance would lead to a higher firing rate, while an increase in AP threshold and an increase in membrane capacitance would lead to a lower firing rate.

We tried our best to perform the very difficult patch clamp recording under the same conditions as much as possible. Nevertheless, there were still considerable variations among slices. Therefore, we cannot be 100% confident that statistical significance seen for neuronal comparison data are derived from the true differences between ACSF and SeeDB-Live. Therefore, **we discussed the limitations of our electrophysiology experiments in the main text (Lines 273-280).**

Based on our repeated trials, we do agree that some of the electrophysiological parameters may reflect the genuine differences between control and SeeDB-Live. This may not be very surprising given that ionic conditions are not exactly the same. We increased Ca^{2+} and Mg^{2+} concentrations for SeeDB-Live, assuming that they are buffered by BSA. It is possible that Ca^{2+} and Mg^{2+} concentrations were not fully optimized and had affected some parameters, such as AP threshold (J Neurochem. 2024 Aug 20;169(1):e16209). It should also be noted that BSA in SeeDB-Live has a substantial amount of negative charge, and as a result, the concentration of Cl^- was adjusted lower than the control ACSF. Moreover, BSA is known to buffer Na^+ , known as the Donnan effect, potentially affecting effective Na^+ concentration. It should also be noted that SeeDB-Live was optimized based on the overall firing properties of brain slices, not individual electrophysiological parameters. According to the reviewer's suggestion, **these potential issues and mechanisms are now discussed in the main text (Lines 268-280).**

In relation to this issue, we also evaluated spontaneous firing under different Na^+ and Cl^- conditions. In one condition, the Na^+ concentration was adjusted to ACSF, while the Cl^- concentration was lower. In another condition, the Cl^- concentration was adjusted to ACSF, while the osmolality was higher. There were no statistically significant differences between the conditions (**Fig. R16**). However, abnormal firing (cortical spreading depression) was occasionally found at higher concentrations. Moreover, osmolality will be slightly higher in this condition. Therefore, we have chosen lower NaCl condition (151 mM Na^+ and 111 mM Cl^-). **Fig. R16** is now included in the **new Extended Data Fig. 2c-e**.

Thy1-GCaMP6f, OB mitral cells, P9-11, 2P

Figure R16. Optimization of sodium chloride in SeeDB-Live (new Extended Data Fig. 2c, d).

(a, b) Spontaneous activity of mitral cells in the olfactory bulb (P9-11) was measured under different concentrations of sodium and chloride. Amplitude (a) and frequency (b) were analyzed. Thy1-GCaMP6f mice were used for two-photon imaging of acute olfactory bulb slices.

In summary, it is extremely difficult to perform patch clamp recordings for the control and SeeDB-Live under the same conditions. We did our best to make a fair comparison. However, we cannot fully exclude possible sampling biases. We also acknowledge that some of the parameters may genuinely differ between the control and SeeDB-Live conditions, because the ionic conditions are not identical. **We have now highlighted these issues in the revised text (Lines 268-280).** We agree that this clarification will be informative for future users.

2) The manuscript still lacks important details. Were the experiments illustrated in Figure 4a-j performed in the anaesthetized state? If so, please state this (the schematic in 4a is different than Extended Data Figure 8a). If not, then I cant see how you could perfuse the surface of the brain with SeeDB-Live for 1 hour and then perform imaging (without a

coverslip to dampen movement). Along the same lines, please provide more information than 'we performed a durotomy' (line 312) for the data illustrated in Figure 5a-e. Please clearly state in the main text, and Figure caption that the visual recordings were performed in anaesthetized mice.

Thank you for raising this issue. We agree and have updated the schematic illustrations, figure captions, and the main text. These changes are **highlighted** in the revised manuscript.

3) Furthermore, there is little information provided in the text for the values presented. Please always, where possible, include the absolute values and the exact p values (not just $p > 0.05$). This would be helpful to understand some of the data – for example, Figure 5n illustrates there is no significant change in the amplitude of the calcium responses, however, there really appears to be a strong trend for most cells to decrease on day 3. Please include more information about the odor presented (valeraldehyde) in the main text.

We agree, but some of the figures (e.g., multiple comparison) would be too busy if we include p values for all combinations. Instead, we included p values in the raw data in Supplementary Excel file (**Table S3 and S4**).

4) Although the revised manuscript has improved in clarity, the manuscript is still disjointed. For example, Figures 4 and 5 jump between awake and anaesthetized (presumably) recordings, and there are a mis-match of imaging locations/cell types. It is unclear which imaging was performed during the locomotion behavior and the positioning of the locomotion data in Figure 4 is potentially misleading if prior data (Figure 4a-j) are not performed during this behavior. What was the preparation for the behavioral assessment?

We agree that old Fig. 4a-j and 4k-p are separate sets of experiments. Similarly, old Fig. 6a-f and 6g-n are separate sets of experiments. We have now rearranged the figure layout. **We included spaces between different sets of experiments in the figures.** We also improved the descriptions of the figure legends to avoid the confusion (**highlighted** in the figure legends).

5) *Please always note how long after SeeDB-Live solution was applied were the electrophysiological recordings were performed (were control recordings performed at similar timepoints?).*

Thank you for raising this issue. Indeed, we found that sEPSC frequency is strongly affected by the time after slice preparations. In the new sets of electrophysiology experiments, we have performed recording under the same timepoint for the control and ACSF. As a result, we found no significant differences in sEPSC frequency, as explained above (**Fig. R15**).

This point is now mentioned in the methods section (**Lines 1348-1356**). Thank you again for raising this important issue.

6) *Extended Data Figure 7 illustrates important information which supports the statement (line 328) “Optical clearing with SeeDB-Live is transient in vivo (~ 1 hour) as BSA is gradually washed out.” Please include this, at least in graphical form, in a main Figure.*

We agree. This figure has been moved to the main figure as **new Fig. 4k**. **New Fig 5j** also explains that we need to repeat SeeDB-Live treatment for chronic imaging.

7) *Considering the similarities of the approach, more consideration must be taken of the Ou et al, Science, 2024 study in the Introduction. Understandably it has different applications, but only referring to this study as a previous immersion-based clearing agent is not adequate. Please consider moving the paragraph (line 189) to the Introduction.*

We totally agree. Ou et al. has been too influential but is misunderstood by most people. The osmolarity issue was completely overlooked in the Ou et al., (*Science* 2024), and it is important to highlight what is necessary for the optical clearing of live tissue (i.e., optimal refractive index, osmolarity, physiological ionic conditions, toxicity, etc.). We have now fully explained in the introduction that the method reported by Ou et al, *Science* 2024 cannot make live and healthy tissue transparent (**Lines 67-71**).

8) *The statement (line 285) is misleading “Thus, SeeDB-Live enables calcium imaging of healthy neuronal activity in acute brain slices using conventional confocal microscopy,*

without using expensive two-photon microscopy systems.”. Confocal microscopy can be used to image fluorescence in shallow cells in slices (up to 50um or so depth) – please alter the sentence.

We did demonstrate Ca²⁺ imaging with confocal in **old Fig. 3m (new Fig. 3o)**. We have split this part into separate paragraphs to avoid confusion.

Reviewer #2:

Remarks to the Author:

The updated version of the paper from Inagaki and colleagues addresses most of the concerns raised during the first revision and drastically improves the results.

Thank you for valuable suggestions.

1) A point that wasn't entirely covered is the tolerability of the clearing.

The authors demonstrated that SeeDB-Live can be used up to 7 days, but a longitudinal study can last weeks, even months. Therefore, it would be important to verify the tolerance of treatment for a longer period, adding some measurements after 2 - 3 and 4 weeks.

We agree. This is simply a matter of time. After submission of our previous revision, we continued the chronic imaging experiments. **We have now performed longitudinal imaging of the same sets of neurons for up to 4 months**, without noticeable changes in neuronal morphology and Ca^{2+} responses (**Fig. R17**). This is now included in **new Fig. 5n-p**.

Figure R17. Longitudinal imaging cortical pyramidal neurons in awake animals using repeated clearing with SeeDB-Live (new Fig. 5n-p).

(a) Morphology of cortical pyramidal neurons visualized with CyRFP1 fluorescence. (b) Ca^{2+} responses to whisker stimulation for the same sets of neurons. jRCaMP1m was imaged to measure $\Delta F/F_0$. (c) Quantification of soma size and Ca^{2+} responses, showing no sign of toxicity.

2) Moreover, the craniotomy was performed on the primary somatosensory cortex and not on the motor cortex; therefore, the observation that the locomotor activity on a treadmill is intact doesn't necessarily mean that there aren't any other alterations.

We apologize for the confusion. The craniotomy made in old Fig. 4n include the motor cortex, of course. The **new Fig. 4l (left)** now shows the area of the craniotomy.

3) I suggest evaluating the cytoarchitecture integrity of the brain, also by verifying the presence of inflammatory processes, with a post-mortem structural analysis of the animal's brain.

We totally agree. Indeed, we were often asked whether microglial activation occurs after SeeDB-Live treatment. To address this issue, we performed immunohistochemistry for the post-mortem brain sections 1 day after SeeDB-Live treatment (10 days after durotomy). We have chosen the timepoint based on standard protocol in this field (Trachtenberg et al., 2002). We found no difference in the density of neurons (NeuN^+) and microglia after SeeDB-Live treatment. Microglial density was comparable to the contralateral (non-treated) cortex for both ACSF and SeeDB-Live/ACSF groups. Moreover, dendritic morphology of microglia was comparable between control and SeeDB-Live, suggesting no obvious sign of inflammatory responses (**Fig. R18**). These data are now included in the **new Extended Data Fig. 7b-e**.

We also show the morphology and Ca^{2+} response data for longitudinal imaging experiments (**new Fig. 5n-p**).

Figure R18. Evaluation of inflammatory responses after SeeDB-Live treatment (new Extended Data Fig. 7b-e).

(a) DAPI, anti-NeuN (neuron), and anti-Iba1 (microglia) staining of the cerebral cortex after ACSF-HEPES or SeeDB-Live/ACSF-HEPES treatment. The cytoarchitecture in S1 was preserved after SeeDB-Live/ACSF-HEPES treatment.

(b) Morphology of microglia after treatment with ACSF-HEPES or SeeDB-Live/ACSF-HEPES.

(c) Treated and untreated areas of ACSF-HEPES or SeeDB-Live/ACSF-HEPES in the brain. A

large cranial window was made only over the treated area in the right cortex.

(d, e) Density of Iba1-positive microglia in ACSF-HEPES- and SeeDB-Live/ACSF-HEPES-treated mice. Comparison of the microglial density in untreated and treated areas is shown in **(d)**, and the ratio (treated/untreated) is shown in **(e)**.

(f) Evaluation of microglial morphology in ACSF-HEPES- and SeeDB-Live/ACSF-HEPES-treated mice. Microglial morphology was manually traced in 3D, and quantitative analyses (soma volume, soma roundness, branch number, maximum branch length, and total branch length) were performed. Cells were selected in the S1 L2/3 region in a blinded manner. The box plots indicate median \pm interquartile range (IQR). $n = 24$ and 21 for untreated (contralateral) and treated area of ACSF-HEPES, 19 and 22 for untreated (contralateral) and treated area of SeeDB-Live/ACSF-HEPES, respectively. There was no obvious sign of inflammatory responses induced by SeeDB-Live. Indeed, it is known that BSA is structurally stable and minimally induces immune reactions. Moreover, albumin can bind to some toxic and/or inflammatory chemicals.

4) Finally, there is curiosity about the applicability of the method to other tissues. Do the authors expect that See-Live will work on other animals or types of tissue, e.g., the heart, or will the concentration of the clearing solution need to be adjusted? It would be interesting to include a discussion of this in the future perspectives.

We totally agree. SeeDB-Live would be useful for other species with minor modifications (e.g., primates and invertebrate brains). However, we don't think that SeeDB-Live can be easily applied to many other organs, as was briefly discussed in Lines 438-445. BSA would not easily penetrate the highly packed tissues (e.g., heart) and/or barrier structures (e.g., vascular endothelium) in other organs.

Reviewer's comments are *italicized*. Newly added parts are **highlighted**.

Reviewer #2:

Remarks to the Author:

The authors present a comprehensive set of experiments demonstrating that SeeDB-Live enables high-resolution functional imaging across a range of live tissues and organoids. The manuscript is well-structured, clearly written, and the results are compelling, largely supporting the authors' claims.

However, several revisions and additional data would strengthen the conclusions, particularly regarding claims of non-toxicity and long-term compatibility.

We are pleased to see that this reviewer was mostly satisfied with our previous revision. We also appreciate the additional comments to further improve the quality of the work.

*1. Although the authors note that BSA “is structurally stable and minimally induces immune reactions,” it remains **a large, xenogenic protein**. Inflammation was assessed only 24 hours after a single, 1-hour application of SeeDB-Live in thin sections. However, many key applications, including the chronic imaging experiments (Fig. 5i–p), involve repeated use. To exclude long-term immune responses, **it would be valuable to examine repeated applications over the four-month imaging period** in which no changes in soma size or sensory responses were observed. Because Iba1 labels both resting and activated microglia, **additional co-labeling with additional markers** (e.g. CD68, Cb11, HLA-DR, GFAP, TUNEL, and activated Caspase-3) would strengthen the conclusion of “undetectable toxicity.” Furthermore, performing immunohistochemical and morphological analyses on **thicker sections (50–100 μm) rather than 16 μm sections** would enable 3D reconstruction of the cytoarchitecture and provide more accurate quantification.*

It is true that BSA is a large xenogenic protein. However, it has already been widely used in the field of drug delivery studies *in vivo* (Bhushan et al., 2017; Fasano et al., 2005). Moreover, isotope- or fluorescently labeled BSA has been used extensively for 50 years to study CSF dynamics *in vivo* without serious side effects (e.g., Brady et al., *Fluids Barriers CNS*, 2020, PMID: 33256800).

We agree that it will be informative to include staining for activated microglia and astrocytes. As a marker for activated microglia for inflammatory responses (M1), we used anti-CD16/32. We also stained for activated astrocytes (anti-GFAP) and all astrocytes (anti-Sox9). The staining for CD16/32 and GFAP in SeeDB-Live treated brains was comparable to or even lower in the control (contralateral side). Moreover, we detected no signals for cleaved Caspase 3 in both control and SeeDB-Live samples (of course, we confirmed that this antibody works with positive controls). These additional results (**Fig. R19**) further strengthen the claim that there is no detectable toxicity by SeeDB-Live. The additional data are now included in the **new Extended Data Fig. 7**.

Figure R19. Evaluation of inflammatory responses after SeeDB-Live treatment (new Extended Data Fig. 7b-k).

(a-j) Evaluation of inflammatory responses in S1 after SeeDB-Live treatment *in vivo*. A large cranial window was made 10 days prior to the SeeDB-Live treatment. SeeDB-Live treatment was performed for 1 hour. Mice were sacrificed 1 day after the treatment. S1 is shown.

(a) DAPI (nuclei), anti-NeuN (neurons), anti-Iba1 (microglia), anti-CD16/32 (activated microglia), anti-GFAP (activated astrocytes), anti-SOX9 (astrocyte nuclei), and anti-cleaved caspase-3 (apoptotic cells) staining of the cerebral cortex after ACSF-HEPES or SeeDB-Live/ACSF-HEPES treatment. The cytoarchitecture in S1 was preserved after SeeDB-Live/ACSF-HEPES treatment. Representative results from three mice per condition are shown.

(b) Treated and untreated (control) areas by ACSF-HEPES or SeeDB-Live/ACSF-HEPES. A large cranial window was made only over the right cortex.

(c, d) Density of Iba1-positive microglia in ACSF-HEPES- and SeeDB-Live/ACSF-HEPES-treated mice. Comparison of the microglial density in untreated and treated areas is shown in **(c)**, and the ratio (treated/untreated) is shown in **(d)**. $n = 3$ mice each for treatment with ACSF-HEPES (control) and SeeDB-Live/ACSF-HEPES. n.s., not significant ($p \geq 0.05$) (Wilcoxon signed-rank test in **(c)**, Wilcoxon rank-sum test in **(d)**).

(e) Morphology of microglia after treatment with ACSF-HEPES or SeeDB-Live/ACSF-HEPES.

(f) Evaluation of microglial morphology in ACSF-HEPES- and SeeDB-Live/ACSF-HEPES-treated mice. Microglial morphology was manually traced in 3D, and quantitative analyses (soma volume, soma roundness, branch number, maximum branch length, and total branch length) were performed. Cells were selected in the S1 L2/3 region in a blinded manner. The box plots indicate median \pm interquartile range (IQR). $n = 24$ and 21 for untreated (contralateral) and treated area of ACSF-HEPES, 19 and 22 for untreated (contralateral) and treated area of SeeDB-Live/ACSF-HEPES, respectively. $*p < 0.05$; n.s., not significant ($p \geq 0.05$) (Tukey-Kramer multiple comparison test).

(g-j) Quantification of inflammatory and cytotoxic markers in untreated and treated areas of the cerebral cortex in ACSF-HEPES- and SeeDB-Live/ACSF-HEPES-treated mice.

(g) Fraction of CD16/32-positive area within Iba1-positive regions (microglia activation).

(h) Fraction of GFAP-positive area (astrocyte activation).

(i) Density of SOX9-positive cells (astrocyte nuclei).

(j) Density of cleaved caspase-3-positive cells (neuronal apoptosis).

$n = 3$ mice each for treatment with ACSF-HEPES and SeeDB-Live/ACSF-HEPES. $**p < 0.01$; n.s., not significant ($p \geq 0.05$) (Tukey-Kramer multiple comparison test).

We also agree that it is important to examine the possible inflammatory responses after repeated SeeDB-Live treatment. Using the same set of antibodies, we stained the brain sections treated with SeeDB-Live 7 times over the 7 months. One day after durotomy, we detected activated microglia (anti-CD16/32) and astrocytes (anti-GFAP) as has been reported previously (e.g., Liu et al., *Nat Commun*, 16: 7584, 2025). On the other hand, we did not observe any obvious signs of inflammatory responses after repeated treatment with SeeDB-Live (**Fig. R20**). Due to space limitations, a part of **Fig. R20** is now included in the **new Extended Data Fig. 7**.

Figure R20. Assessment of inflammatory responses after repeated SeeDB-Live treatment (new Extended Data Fig. 7).

(a) A large cranial window was made only over the right cortex. **(b)** Experimental timeline showing the durotomy and repeated SeeDB-Live/ACSF-HEPES treatments. For the chronic SeeDB-Live/ACSF-HEPES treatment, mice underwent durotomy on day -7, followed by seven SeeDB-Live treatments between day 0 and day 218, and were sacrificed on days 219. **(c)** Representative images of the cerebral cortex stained with DAPI (nuclei), anti-NeuN (neurons), anti-Iba1 (microglia), anti-CD16/32 (activated microglia), anti-GFAP (activated astrocytes), anti-SOX9 (astrocyte nuclei), and anti-cleaved caspase-3 (apoptotic cells) after durotomy or repeated SeeDB-Live/ACSF-HEPES treatment (day 0, 3, 7, 80, 100, 120, and 218). Representative images from 3 slices of 3 mice (durotomy) and of a single mouse (SeeDB-Live) are shown.

We agree that analyzing thicker brain sections can reveal more detailed morphology of microglia. However, it is difficult to stain thick tissue (>50 μm) with anti-Iba1 antibodies. Therefore, our colleagues who specialize in microglia have advised that immunostaining is best performed on thin, frozen sections. Indeed, most of previous studies in the field have used 10-20 μm thick frozen sections to detect very subtle morphological changes in microglia (e.g., Wang et al., *Nature* 644, 759-, 2025). Our analysis was performed in an unbiased and blind manner to detect morphological changes. It should also be noted that a similar approach was used to examine the inflammation caused by the optical window implantation (Xu et al., *Nat Neurosci*, 10: 549-, 2007; Liu et al., *Nat Commun*, 16: 7584, 2025). Therefore, the requested experiments exceed the current standard of this field and the purpose of our analysis. We believe that our quantification was accurate enough to detect any faint changes in microglial morphology.

2. The authors show that intermittent 4-hour clearing with SeeDB-Live does not affect intestinal organoid growth over a few days. To validate the use of SeeDB-Live in long-term organoid studies (which often run for weeks), it would be ideal to also assess viability and apoptosis markers (e.g., live/dead staining, cleaved caspase-3) after repeated exposures. Longitudinal functional assays, such as repeated Ca^{2+} imaging over several days, would also help rule out cumulative toxicity.

As shown in **Figs. R19** and **R20**, apoptotic cells were extremely rare in normal tissues. Moreover, we do not believe that we can detect differences in apoptosis when cell growth is comparable to the control. In general, the growth assay is a more sensitive method than

apoptotic markers, which only capture a snapshot of cell death. Therefore, we do not believe that the requested experiment will yield meaningful results.

More importantly, we do not want to claim that SeeDB-Live is very useful for the longitudinal functional assay of organoids. As discussed in the current manuscript, cell growth decreased when spheroids were cultured continuously in SeeDB-Live (**Extended Data Fig. 4a**). This is why we treated spheroids and intestinal organoids for only 4 hours per day. This is most likely due to limited diffusion of oxygen or nutrients, as discussed in Lines 207-208, because monolayer culture was totally healthy. Similarly, we failed to differentiate neural organoids from ES cells under SeeDB-Live medium (data not shown). Therefore, we believe that it is more informative to discuss this potential limitation of SeeDB-Live for those interested in its application to organoids. Currently, SeeDB-Live is useful for acute functional assays, but not yet for long-term culture or induction experiments. This point is now added to the manuscript (**Lines 436-437**). A microfluidics-based circulation system may overcome this problem as discussed in Lines 437-438, but that is beyond the scope of this study.

*3. In Figure 5n–p, please clarify whether the data in panel p comes from a single animal or is representative of a cohort. **Demonstrating consistent results from multiple animals (n = 2–3) undergoing the same repeated clearing and imaging protocol would make this already strong conclusion unassailable.***

The chronic imaging with SeeDB-Live treatment is a highly demanding experiment, and it was unrealistic to obtain multiple animals in parallel under the same timeline. We, of course, performed chronic imaging on additional animals, but in a different timeline. We performed chronic imaging (≥ 5 weeks) with multiple SeeDB-Live treatments (≥ 4 times) for four animals and obtained consistent results. The information regarding reproducibility is now included in the legend to **Fig. 5**. An additional example with similar results is shown in **Fig. R21** (not included in the manuscript, due to the space limitation).

Figure R21. Longitudinal imaging of cortical pyramidal neurons in awake animals using repeated clearing with SeeDB-Live (additional example).

(a) L2/3 neurons in S1 were visualized with AAV-jGCaMP8m-P2A-CyRFP1. Z-stacked images (imaging depth: 177-237 μm) of the CyRFP1 fluorescence of L2/3 neurons after SeeDB-Live treatment at weeks 1, 2, 3 and 5. (b) Mean calcium responses of jGCaMP8m-expressing L2/3 neurons to whisker stimulations with air puffs. Mean responses to five whisker stimulations are shown. (c) Soma size, $\Delta F/F_0$, and half-rise time of jGCaMP8m-expressing L2/3 neurons after SeeDB-Live treatment at weeks 1, 2, 3 and 5. (p) $\Delta F/F_0$ and half-rise time were calculated from the mean responses to five whisker stimulations. Half-rise time was analyzed only for cells whose maximum $\Delta F/F_0$ was greater than the mean + 5 SD of F_0 on all time points (1, 2, 3 and 5). $n = 16$ cells for soma size and $\Delta F/F_0$; $n = 6$ cells for half-rise time. n.s., not significant ($p \geq 0.05$) (Paired ANOVA). Representative data from four mice with different imaging durations (one mouse for 7 months, two mice for 5 weeks, and one mouse for 3 weeks).

4. The vivo clearing is shown to be transient, lasting approximately one hour before washing out. While this reversibility is presented as a positive feature, it also imposes a practical limitation by constraining the duration of imaging sessions. Including a brief sentence in the Discussion acknowledging this trade-off would be helpful for readers planning experiments.

We agree, and this point is now briefly mentioned in the Discussion (pages 417-418).

5. The authors describe the lower optimal refractive index (RI) for live cells (~1.37) compared to fixed cells (~1.42) as "paradoxical." The Discussion would be strengthened by elaborating on the biophysical basis for this difference. Specifically, the manuscript could clarify that for live cells, the goal is matching the extracellular medium to the aqueous cytosol, whereas for fixed/permeabilized cells, the higher RI is needed to match dehydrated intracellular biomolecules.

We agree, and we have now discussed this point (pages 414-417).